# The primate Major Histocompatibility Complex as a case study of gene family evolution

**Alyssa Lyn Fortier[1]\*, Jonathan K Pritchard[1,2]**

[1]Department of Biology, Stanford University, Stanford, United States; [2]Department of Genetics, Stanford University, Stanford, United States

## eLife Assessment

This **important** manuscript presents a thorough analysis of the evolution of Major Histocompatibility Complex gene families across Primates. A key strength of this analysis is the use of state-of-the-art phylogenetic methods to estimate rates of gene gain and loss, accounting for the notorious difficulty to properly assemble MHC genomic regions. Overall the evidence for the authors' conclusions - that there is considerable diversity in how MHC diversity is deployed across species - **compelling**.

**\*For correspondence:**
afortier@stanford.edu

**Abstract** Gene families are groups of evolutionarily related genes. One large gene family that has experienced rapid evolution lies within the Major Histocompatibility Complex (MHC), whose proteins serve critical roles in innate and adaptive immunity. Across the ~60 million year history of the primates, some MHC genes have turned over completely, some have changed function, some have converged in function, and others have remained essentially unchanged. Past work has typically focused on identifying MHC alleles within particular species or comparing gene content, but more work is needed to understand the overall evolution of the gene family across species. Thus, despite the immunologic importance of the MHC and its peculiar evolutionary history, we lack a complete picture of MHC evolution in the primates. We readdress this question using sequences from dozens of MHC genes and pseudogenes spanning the entire primate order, building a comprehensive set of gene and allele trees with modern methods. Overall, we find that the Class I gene subfamily is evolving much more quickly than the Class II gene subfamily, with the exception of the Class II MHC-DRB genes. We also pay special attention to the often-ignored pseudogenes, which we use to reconstruct different events in the evolution of the Class I region. We find that despite the shared function of the MHC across species, different species employ different genes, haplotypes, and patterns of variation to achieve a successful immune response. Our trees and extensive literature review represent the most comprehensive look into primate MHC evolution to date.

## Introduction

Gene families are groups of related genes categorized by functional similarity or presumed evolutionary relatedness. Based on clustering of their proteins' sequences, human genes fall into hundreds to thousands of distinct families (*Gu et al., 2002*; *Li et al., 2001*; *Demuth et al., 2006*; *Friedman and Hughes, 2003*). Families originate from successive gene duplications, although particular gene copies or entire families can also be lost (*Nei et al., 1997*; *Demuth et al., 2006*). For example, there are hundreds of genes that are specific to human or chimpanzee and have no orthologs in the other species (*Demuth et al., 2006*). This birth-and-death evolution is distinct from evolution at the nucleotide or protein level (*Thornton and DeSalle, 2000*; *Hahn et al., 2005*). However, phylogenetics can

still be applied to understand the relationships within families of genes, providing insight into speciation and specialization (*Thornton and DeSalle, 2000*).

One large gene family is united by a common protein structure called the 'MHC fold'. Having originated in the jawed vertebrates, this group of genes is now involved in diverse functions including lipid metabolism, iron uptake regulation, and immune system function (proteins such as zinc-α2-glycoprotein [ZAG], human hemochromatosis protein [HFE], MHC class I chain–related proteins [MICA, MICB], and the CD1 family; *Hansen et al., 2007*; *Kupfermann et al., 1999*; *Kaufman, 2022*; *Adams and Luoma, 2013*). However, here we focus on the Class I and Class II MHC genes whose protein products present peptides to T-cells ('classical' genes) and/or interact with other immune cell receptors like killer cell immunoglobulin-like receptors (KIRs) or leukocyte immunoglobulin-like receptors (LILRs; both 'classical' and 'non-classical' genes). The classical genes are conventionally known to be highly polymorphic, have an excess of missense variants, and even share alleles across species, all indicative of balancing selection at the allele level (*Maccari et al., 2017*; *Maccari et al., 2020*; *Robinson et al., 2024*; *Hughes and Nei, 1988*; *Hughes and Nei, 1989b*; *Arden and Klein, 1982*; *Mayer et al., 1988*). In addition to variation within individual genes, the region is also significantly structurally divergent across the primates (*Mao et al., 2024*). Balancing selection is evident at the haplotype level as well, where haplotypes with drastically different functional gene content are retained in various primate populations (*Hans et al., 2017*; *de Groot et al., 2017b*; *de Groot et al., 2009*; *Gleimer et al., 2011*; *Heijmans et al., 2020*). This motivates the need to study the MHC holistically as a gene family. Even though species may retain different sets of genes and haplotypes, related genes likely function similarly, facilitating comparisons across species. Thus, by treating the genes as a related set, our understanding improves significantly compared to considering single genes in isolation. Because gene family birth-and-death is important to speciation and the MHC itself is highly relevant to organismal health, this family is an excellent case study for gene family evolutionary dynamics. Here, we focus on the primates, spanning approximately 60 million years within the over 500 million-year evolution of the family (*Flajnik and Kasahara, 2010*).

There are two classes of MHC genes within the greater family (Class I and Class II), and each class contains two functionally distinct types of genes: 'classical' and 'non-classical'. 'Classical' MHC molecules perform antigen presentation to T cells with variable $\alpha\beta$ TCRs—a key part of adaptive immunity—while 'non-classical' molecules have niche immune roles. The classical Class I molecules are generally highly polymorphic, ubiquitously expressed, and present short, intracellularly derived peptides to T cells. Many of them also serve as ligands for other types of immune cell receptors and influence innate immunity (see Appendix 1; General roles of MHC and MHC-like genes; for an overview; *Anderson et al., 2023*; *Parham and Moffett, 2013*; *Guethlein et al., 2015*; *Hans et al., 2017*; *Wroblewski et al., 2019*). The non-classical Class I molecules have limited polymorphism, restricted expression, and perform specific tasks such as mediating maternal-fetal interaction and monitoring levels of MHC synthesis. In humans, the classical Class I genes are HLA-A, -B, and -C, and the non-classical Class I genes are HLA-E, -F, and -G (*Heijmans et al., 2020*). In contrast, the classical Class II molecules are expressed only on professional antigen-presenting cell types and present longer, extracellularly-derived peptides to T cells (*Gfeller and Bassani-Sternberg, 2018*; *Heijmans et al., 2020*; *Neefjes et al., 2011*). The non-classical Class II molecules assist with loading peptides onto the classical Class II molecules before their transport to the cell surface (*Dijkstra and Yamaguchi, 2019*; *Neefjes et al., 2011*). In humans, HLA-DP, -DQ, and -DR are the classical Class II molecules, and HLA-DM and -DO are the non-classical molecules (see Appendix 3 for more detail on all of these genes).

However, the landscape of MHC genes differs even across closely related species. Over evolutionary time, the Class I gene subfamily has been extraordinarily plastic, having undergone repeated expansions, neofunctionalizations, and losses (*Hans et al., 2017*; *Wilming et al., 2013*; *Otting et al., 2020*; *Heijmans et al., 2020*). Convergent evolution has also occurred; in different primate lineages, the same gene may be inactivated, acquire a new function, or even evolve similar splice variants (*Hans et al., 2017*; *Wilming et al., 2013*; *Otting et al., 2020*; *Heijmans et al., 2020*; *Walter, 2020*). As a result, it is often difficult to detect orthologous relationships in Class I even within the primates (*Hughes and Nei, 1989a*; *Piontkivska and Nei, 2003*; *Go et al., 2003*; *Flügge et al., 2002*; *de Groot et al., 2020*). Studies that focus only on the highly polymorphic binding-site-encoding exons are complicated by these phenomena, necessitating a more comprehensive look into MHC evolution across exons and species groups.

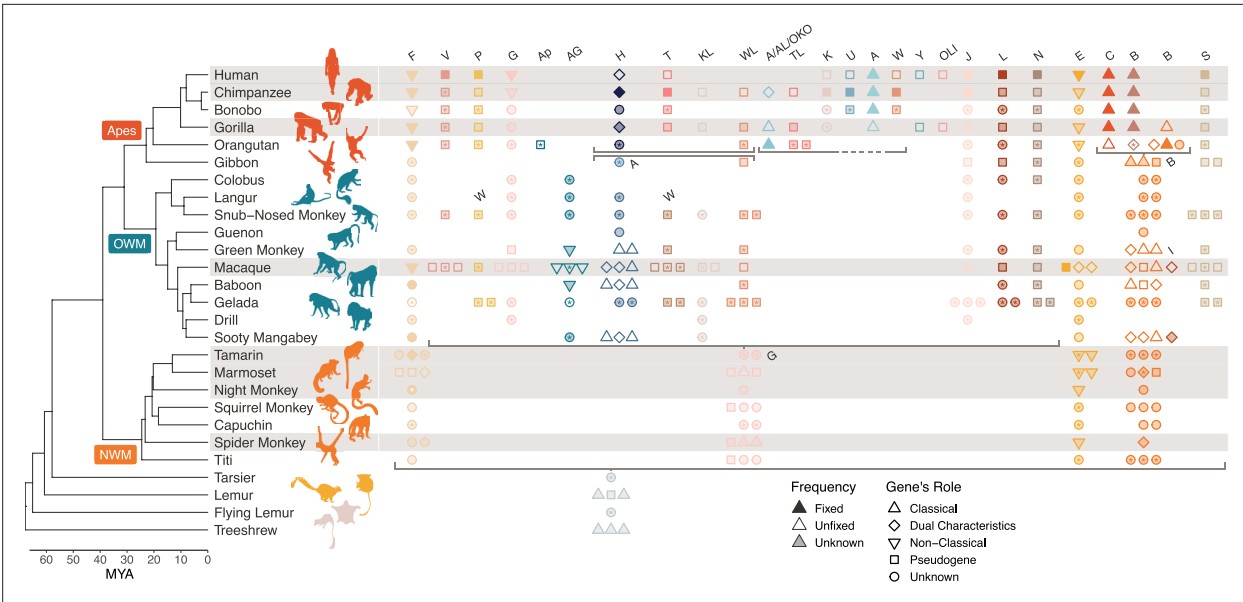

**Figure 1.** Class I MHC genes present in different species. The primate evolutionary tree (*Kuderna et al., 2023*) is shown on the left-hand side (nonprimate icons are shown in beige). The MHC region has been well characterized in only a handful of species; the rows corresponding to these species are highlighted in gray. Species that are not highlighted have partially characterized or completely uncharacterized MHC regions. Asterisks indicate new information provided by the present study, typically discovery of a gene's presence in a species. Each column/color indicates an orthologous group of genes, labeled at the top and ordered as they are in the human genome (note that not all genes appear on every haplotype). A symbol indicates that a given gene is present in a given species; when a species has three or more paralogs of a given gene, only three symbols are shown for visualization purposes. Filled symbols indicate that the gene is fixed in that species, outlined symbols indicate that the gene is unfixed, and semi-transparent symbols indicate that the gene's fixedness is not known. The shape of the symbol indicates the gene's role, either a pseudogene, classical MHC gene, non-classical MHC gene, a gene that shares both features ('dual characteristics'), or unknown. The horizontal gray brackets indicate a breakdown of 1:1 orthology, where genes below the bracket are orthologous to two or more separate loci above the bracket. The set of two adjacent gray brackets in the top center of the figure shows a block duplication. Gene labels in the middle of the plot ('W', 'A', 'G', 'B', and 'I') clarify genes that are named differently in different species. OWM, old-world monkeys; NWM, new-world monkeys.

The online version of this article includes the following source data and figure supplement(s) for figure 1:

**Source data 1.** References for *Figure 1*.

**Figure supplement 1.** Class I MHC genes present in different species (without asterisks).

**Figure supplement 2.** MHC Class I genes in reference genomes.

In contrast to Class I, the Class II region has been largely stable across the primates, but gene content still varies in other species. For example, the pig has lost the MHC-DP genes while expanding the number of MHC-DR genes, and the cat has lost both the MHC-DQ and -DP genes, relying entirely on MHC-DR (*Hammer et al., 2020*; *Okano et al., 2020*). The use of the different Class II molecules appears to be fluid, at least over longer timescales, motivating the need to fill in the gaps in knowledge in the primate tree.

Due to the large volume of existing MHC literature, results are scattered across hundreds of papers, each presenting findings from a limited number of species or genes. Thus, we first performed an extensive literature review to identify the genes and haplotypes known to be present in different primate species. We present a detailed summary of these genes and their functions in Appendix 3. We also performed a *BLAST* search using a custom IPD-based MHC allele database against several available reference genomes to discover which genes were present on various primate reference haplotypes (*Figure 1—figure supplement 1*, *Figure 2—figure supplement 2*). Our *BLAST* search and our search of NCBI RefSeq confirmed the presence of various genes in several species for the first time. *Figures 1 and 2* show the landscape of MHC genes present in different primate species for Class I and Class II, respectively. The inclusion of sequences from dozens of new species across all genes and the often-ignored pseudogenes helps us paint a more detailed picture of MHC evolution in the primates.

In this work, we present a large set of densely sampled Bayesian phylogenetic trees using sequences from a comprehensive set of MHC genes across dozens of primate species. These trees permit us to

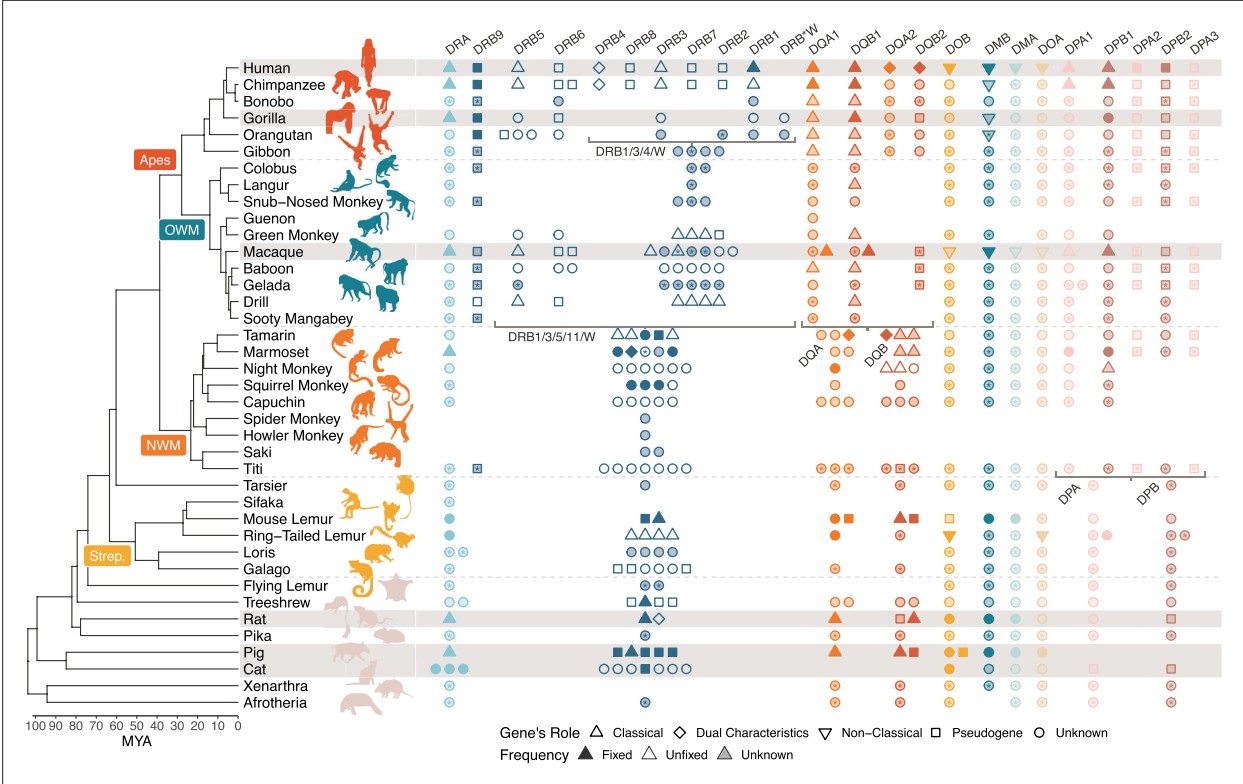

**Figure 2.** Class II MHC genes present in different species. The mammal evolutionary tree is shown on the left-hand side, with an emphasis on the primates (**Foley et al., 2023**; **Kuderna et al., 2023**). The rest of the figure design follows that of **Figure 1**, except that we did not need to limit the number of symbols shown per locus/species due to space constraints. OWM, old-world monkeys; NWM, new world monkeys; Strep., *Strepsirrhini*.

The online version of this article includes the following source data and figure supplement(s) for figure 2:

**Source data 1.** References for **Figure 2**.

**Figure supplement 1.** Class II MHC genes present in different species (without asterisks).

**Figure supplement 2.** MHC Class II genes in reference genomes.

explore the overall evolution of the gene family and relationships between genes, as well as trace particular allelic lineages over time. Across the trees, we see examples of rapid gene turnover over just a few million years, evidence for long-term balancing selection retaining allelic lineages, and slowly evolving genes where orthology is retained for long time periods. In this paper, we describe broad-scale differences between classes and discuss some specific results about the relationships between genes. In a companion paper (**Fortier and Pritchard, 2025**), we explore the patterns of polymorphism within individual genes, finding evidence for deep trans-species polymorphism at multiple genes.

## Results

### Data

We collected MHC nucleotide sequences for all genes from the IPD-MHC/HLA database, a large repository for MHC alleles from humans, non-human primates, and other vertebrates (**Maccari et al., 2017**; **Maccari et al., 2020**; **Robinson et al., 2024**). Although extensive, this database includes few or no sequences from several key lineages including the gibbon, tarsier, and lemur. Thus, we supplemented our set of alleles using sequences from NCBI RefSeq (see asterisks in **Figures 1 and 2**). Because the MHC genes make up an evolutionarily related family, they can all be aligned. Using MUSCLE (**Edgar, 2004**), we aligned all Class I sequences together, all Class IIA sequences together, and all Class IIB sequences together. We then constructed trees for various subsets of these sequences using *BEAST2*, a Bayesian MCMC phylogenetic inference method (see Materials and methods; Bayesian phylogenetic analysis for more detail; **Bouckaert et al., 2014**; **Bouckaert et al., 2019**). One major advantage

of *BEAST2* over less tunable methods is that it can allow evolutionary rates to vary across sites, which is important for genes such as these which experience rapid evolution in functional regions (*Wu et al., 2013*). We also considered each exon separately to minimize the impact of recombination as well as to compare and contrast the binding-site-encoding exons with non-binding-site-encoding exons. We did not analyze the introns due to a lack of available data and difficulty aligning intron sequences across genes.

Here, we present these densely sampled Bayesian phylogenetic trees which include sequences from 106 species and dozens of MHC genes. In this paper, we focus on the Class I, Class IIA, and Class IIB multi-gene trees and discuss overall relationships between genes. Our companion paper (*Fortier and Pritchard, 2025*) explores individual clades/gene groups within these multi-gene trees to understand allele relationships and assess support for trans-species polymorphism.

## The MHC across the primates

The MHC is a particularly dynamic example of a gene family due to intense selective pressure driven by host-pathogen co-evolution (*Ebert and Fields, 2020*; *Radwan et al., 2020*). Within the family, genes have duplicated, changed function, and been lost many times in different lineages. As a result, even closely related species can have different sets of MHC genes. Thus, while the MHC has been extensively studied in humans, there is a limit to how much we can learn from a single species. Leveraging information from other species helps us understand the evolution of the entire family and provides key context as to how it currently operates in humans (*Adams and Parham, 2001b*; *Thornton and DeSalle, 2000*). In *Figures 1 and 2*, we compare the genes present in different species. In both, each column represents an orthologous gene, while the left-hand side shows the evolutionary tree for primates and our closest non-primate relatives (*Foley et al., 2023*; *Kuderna et al., 2023*). Humans are part of the ape clade (red label), which is most closely related to the old-world monkeys (OWM; blue label). Next, the ape/OWM clade is most closely related to the new-world monkeys (NWM; orange label), and the ape/OWM/NWM clade is collectively known as the *Simiiformes*. Only species with rows highlighted in gray have had their MHC regions extensively studied (and thus only for these rows is the absence of a gene symbol meaningful). Gene presence in each species is indicated by symbols in each column, and the symbols also indicate the function of the gene and whether it is fixed in the species. Symbols with an asterisk indicate contributions from this work.

*Figure 1* shows that not all Class I genes are shared by apes and OWM, and much fewer are shared between apes/OWM and NWM. Genes have also been differently expanded in different lineages. While humans and most other apes have a single copy of each gene, the OWM and NWM have multiple copies of nearly all genes. Additionally, many genes exhibit functional plasticity; for example, MHC-G is a non-classical gene in the apes and a pseudogene in the OWM (it is not 1:1 orthologous to NWM MHC-G). The differences between even closely related primate groups indicate that the Class I region is evolving very rapidly.

In contrast, the Class II genes are more stable, as the same genes can be found in even distantly related mammals (*Figure 2*). The notable exception to this pattern is the MHC-DRB group of genes, indicated by dark blue symbols in the middle of *Figure 2*. While some of the individual MHC-DRB genes are orthologous between apes and OWM, indicated by symbols in the same column (e.g. MHC-DRB5), others are limited to the apes alone (e.g. MHC-DRB2). Furthermore, no individual MHC-DRB genes (with the possible exception of MHC-DRB9) are shared between apes/OWM and NWM, pointing to their extremely rapid evolution. Aside from MHC-DRB, the other genes have been relatively stable, although there have been expansions in certain lineages—such as separate duplications of the MHC-DQA and -DQB genes in apes/OWM, NWM, and mouse lemur. Thus, both of the MHC Class I and Class II gene subfamilies appear to be subject to birth-and-death evolution, with Class I and MHC-DRB undergoing the process more rapidly than non-DRB Class II.

## Evolution of a gene family

Now that we had a better picture of the landscape of MHC genes present in different primates, we wanted to understand the genes' relationships. Treating Class I, Class IIA, and Class IIB separately, we performed phylogenetic inference using *BEAST2* on our aligned MHC allele sequences collected from NCBI RefSeq and the IPD-MHC database. *BEAST*2 is a Bayesian method, meaning the set of trees it produces represents the posterior space of trees (*Bouckaert et al., 2019*). For visualization purposes,

we collapsed the space of trees into a single summary tree that maximizes the product of posterior clade probabilities (*BEA, 2024*). In each tree, the tips represent sequences, named either with their RefSeq identifier or with standard allele nomenclature (see Appendix 2). The summary tree for Class I is shown in *Figure 3* while the summary trees for Class IIA and Class IIB are shown in *Figure 4*. Because the tree space cannot be collapsed onto a bifurcating tree with perfect accuracy, we encourage the interested reader to explore the full set of posterior trees (available on Dryad).

We focus first on the Class I genes. *Figure 3A* shows the Class I multi-gene tree using sequences from exon 4, a non-peptide-binding-region-encoding (non-PBR) exon equal in size to each of the peptide-binding-region-encoding (PBR) exons 2 and 3. This exon is the least likely to be affected by convergent evolution, making its tree's structure easier to interpret. This tree—which contains hundreds of tips—has been further simplified for visualization purposes by collapsing clades of related tips, although two fully-expanded clades are shown in panels B and C. We find that sequences do not always assort by locus, as would be expected for a typical gene. For example, ape MHC-J is separated from OWM MHC-J, which is more closely related to ape/OWM MHC-G. Meanwhile, NWM MHC-G does not group with ape/OWM MHC-G, instead falling outside of the clade containing ape/OWM MHC-A, -G, -J, and -K. This supports the fact that the NWM MHC-G genes are broadly orthologous to a large group of genes which expanded within the ape/OWM lineage, rather than being directly orthologous to the ape/OWM MHC-G genes. Appendix 3 explains each of these genes in detail, including previous work and findings from this study.

However, some clades/genes do behave in the expected fashion; that is, with their subtrees matching the overall species tree. One such gene is non-classical MHC-F, shown in *Figure 3B*. Although the gene has duplicated in the common marmoset (Caja-F), this subtree closely matches the species tree shown in the upper right. This indicates that MHC-F is truly 1:1 orthologous across apes, OWM, and NWM. Orthology between apes and OWM (but not with NWM) is also observed for pseudogenes MHC-L, -K, -J, and -V and non-classical MHC-E and -G (*Figure 3—figure supplement 2*, *Figure 5—figure supplement 1*). For the other NWM genes, orthology with apes/OWM is less clear.

While genes such as MHC-F have trees which closely match the overall species tree, other genes show markedly different patterns, such as NWM MHC-G. This gene group is broadly orthologous to a large set of ape/OWM genes and pseudogenes, as its ancestor expanded independently in both lineages. In NWM, the many functional MHC-G genes are classical, and there are also a large number of MHC-G-related pseudogenes. Shown in *Figure 3C*, NWM MHC-G sequences do not always group by species (colored box with abbreviation), instead forming mixed-species clades. Thus, while some MHC-G duplications appear to have occurred prior to speciation events within the NWM, others are species-specific. Similar patterns of expansion are seen among the MHC-A and -B genes of the OWM and the MHC-B genes of the NWM, indicating rapid evolution of many of the Class I genes (*Figure 3—figure supplements 2–4*).

Now turning to the Class II genes, *Figure 4* shows summary trees for exon 3 (non-PBR) for the Class IIA and IIB sequence sets. In the Class II genes, exon 3 does not encode the binding site and is thus less likely to be affected by convergent evolution. In contrast to Class I (*Figure 3*), Class II sequences group entirely and unambiguously by gene, shown by the collapsed trees in *Figure 4A*. However, the subtrees for each gene exhibit varying patterns. As with Class I, non-classical genes tend to evolve in a 'typical' fashion with sequences assorting according to the species tree. This is clearly the case for non-classical MHC-DMA, -DMB, -DOA, and -DOB (*Figure 4—figure supplements 1 and 2*). MHC-DRA and -DPA—although classical—also follow this pattern (*Figure 4B*, *Figure 4—figure supplement 1*). However, the other classical genes' subtrees look very different from the species tree.

There are several reasons why particular MHC gene trees can differ from the overall species tree. Incomplete lineage sorting can happen purely by chance, especially if species have recently diverged. However, balancing selection can cause alleles to be longer-lived, resulting in incomplete lineage sorting even among deeply diverged species; this is called trans-species polymorphism (TSP). *Figure 4C* illustrates this phenomenon for MHC-DPB. Within the OWM clade (shades of green), sequences group by allelic lineage (see Appendix 2 for details on allele nomenclature) rather than by species. For example, crab-eating macaque allele Mafa-DPB1*09:02:01:01 groups with green monkey allele Chsa-DPB1*09:01 (both members of the DPB1*09 lineage) rather than with the other macaque alleles (Mane-, Mamu-, Math-, and Malo-DPB1), despite the fact that these species are 15 million years separated from each other (*Kuderna et al., 2023*). We see this pattern in many Class II genes and

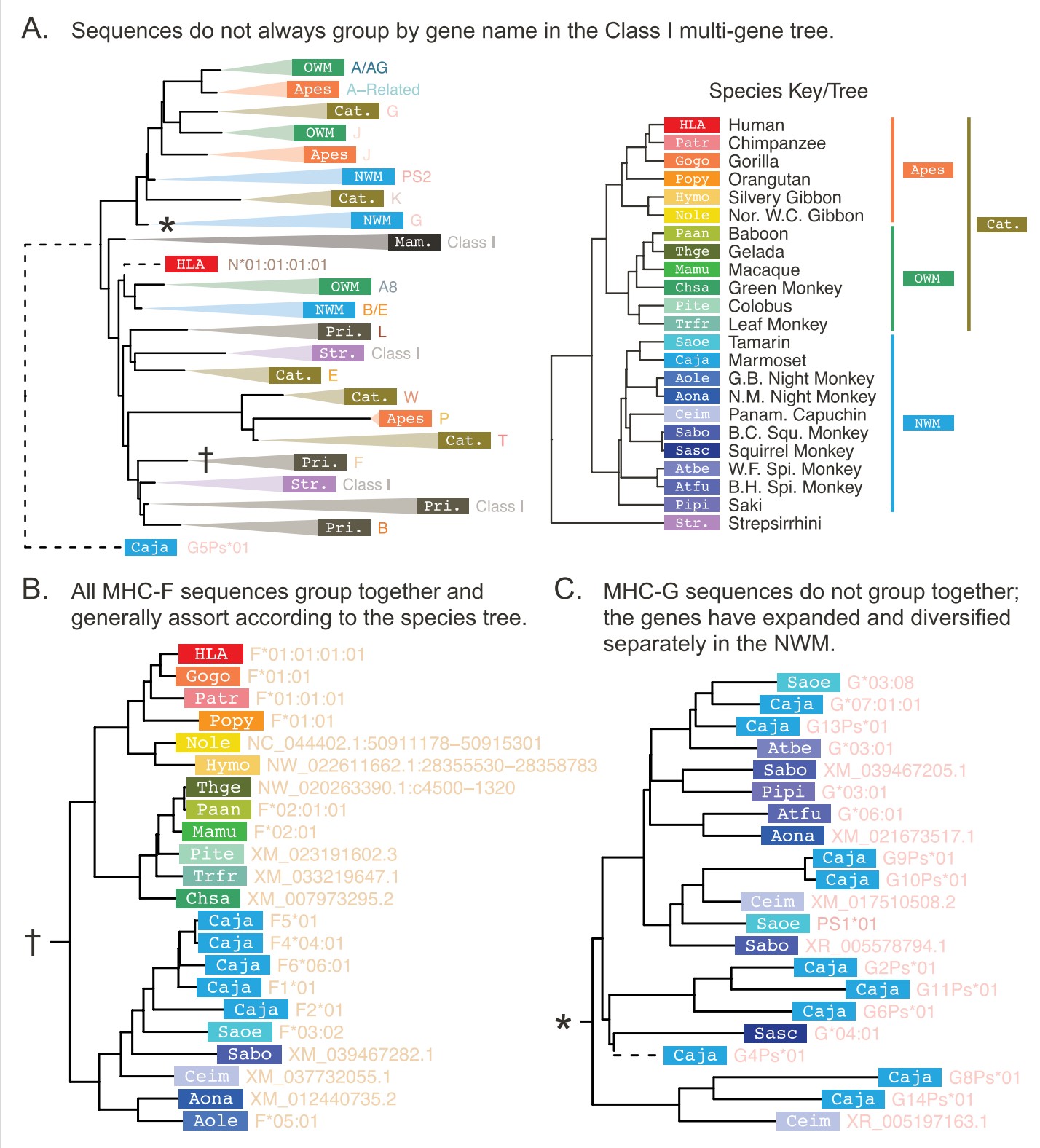

**Figure 3.** The Class I exon 4 multi-gene *BEAST2* tree. The Class I multi-gene tree was constructed using exon 4 (non-PBR) sequences from Class I genes spanning the primates. (**A**) For the purposes of visualization, each clade in the multi-gene tree is collapsed and labeled according to the main species group and gene content of the clade. The white labels on colored rectangles indicate the species group of origin, while the colored text to the right of each rectangle indicates the gene name. The abbreviations are defined in the species key to the right. (**B**) The expanded MHC-F clade (corresponding to the clade in panel A marked by a †). (**C**) The expanded NWM MHC-G clade (marked by a * in panel **A**). In panels B and C, each tip represents a

*Figure 3 continued on next page*

sequence and is labeled with the species of origin (white label on colored rectangle) and the sequence ID or allele name (colored text to the right of each rectangle; see Appendix 2). The species key is on the right-hand side of panel A. Dashed branches have been shrunk to 10% of their original length (to clarify detail in the rest of the tree at this scale). OWM: old-world monkeys; NWM: new-world monkeys; Cat.: Catarrhini—apes and OWM; Pri.: Primates—apes, OWM, and NWM; Mam.: mammals—primates and other outgroup mammals.

The online version of this article includes the following source data and figure supplement(s) for figure 3:

**Source data 1.** GENECONV results for the Class I focused alignments.

**Figure supplement 1.** Species Key.

**Figure supplement 2.** Class I multi-gene *BEAST2* trees.

**Figure supplement 3.** MHC-A-related multi-gene *BEAST2* trees.

**Figure supplement 4.** MHC-B-related multi-gene *BEAST2* trees.

**Figure supplement 5.** MHC-C-related multi-gene *BEAST2* trees.

**Figure supplement 6.** MHC-E-related multi-gene *BEAST2* trees.

**Figure supplement 7.** MHC-F-related multi-gene *BEAST2* trees.

**Figure supplement 8.** MHC-G-related multi-gene *BEAST2* trees.

some Class I genes (*Figure 3—figure supplements 3–5*, *Figure 4—figure supplements 8–10*). In our companion paper, we explore each of these genes further and evaluate the strength of support for TSP in each gene (*Fortier and Pritchard, 2025*).

Another way to obtain discordant trees is in the case of recent expansions of genes. Such expansions make it difficult to assign sequences to loci, resulting in clades where sequences (ostensibly from the same locus) do not group by species. An example of this is shown in *Figure 3C* for the NWM Class I gene MHC-G; long-read sequencing of more NWM haplotypes will help to identify individual genes. The Class II MHC-DRB genes have also expanded, although locus assignments are somewhat clearer. *Figure 4D* shows the Class II subtree for MHC-DRB, where ape sequences (red/orange boxes) are interspersed with OWM sequences (green boxes). The MHC-DRB genes have specific named loci (e.g. MHC-DRB1 or -DRB2), but in this tree only MHC-DRB5 sequences group by named locus (the collapsed ape/OWM MHC-DRB5 clade can be found about 1/3 from the bottom of the tree). The failure of the other named loci to group together indicates a lack of 1:1 orthology between apes, OWM, and NWM for these genes, meaning their names reflect previously observed functional similarity more than evolutionary relatedness. This rapid evolution makes the MHC-DRB genes unique among the Class II genes. Therefore, we created a 'focused' tree with more MHC-DRB sequences in order to explore the evolution of this subgroup further, which is presented in a later section (*Figure 4—figure supplement 8*).

Gene conversion is a third way that gene trees might differ from the overall species tree. Gene conversion is the unidirectional copying of a sequence onto a similar sequence (usually another allele or a related locus), which results in two sequences being unusually similar even if they are not related by descent. We consider this possibility in the next section.

## Detection of gene conversion

Because the MHC contains many related genes in close proximity, gene conversion—the unidirectional exchange of sequence between two similar sequences—can occur (*Chen et al., 2007*). We used the program GENECONV (*Sawyer, 1999*) to infer pairs of sequences of which one has likely been converted by the other (*Figure 3—source data 1*, *Figure 4—source data 1*). We recovered known gene conversion events, such as between human allelic lineages HLA-B*38 and HLA-B*67:02, as well as novel events, such as between gorilla allelic lineages Gogo-B*01 and Gogo-B*03 and ape/OWM lineages MHC-DQA1*01 and MHC-DQA1*05.

However, most of the GENECONV tracts implicated the same pair of loci but in many different groups of species. We interpreted these as gene conversion events that must have happened a long time ago in the early history of the two genes, and they are likely to blame for the topological differences from exon to exon among the trees. For example, in exon 2, the Class I pseudogene MHC-K groups with MHC-G, while in exon 3, it groups with MHC-F, and in exon 4, it groups outside of

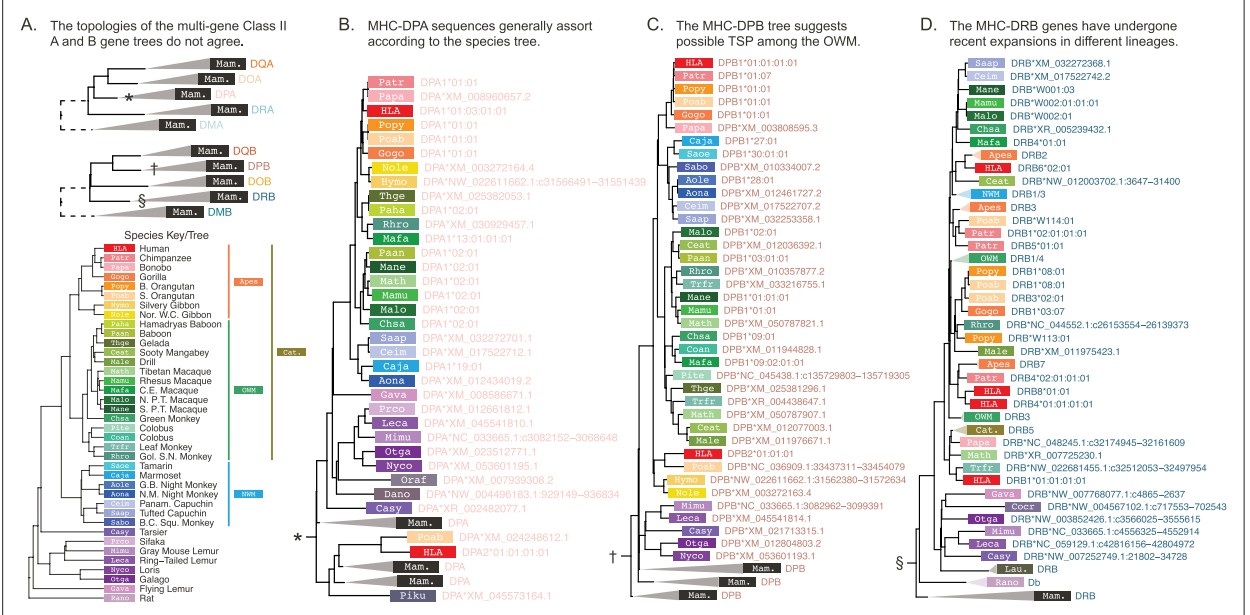

**Figure 4.** The Class II exon 3 multi-gene *BEAST2* trees. The trees were constructed using all Class IIA and all Class IIB exon 3 (non-PBR) sequences across all available species. The design of this figure follows *Figure 3*. (**A**) The top tree shows the collapsed Class IIA gene tree, while the bottom tree shows the collapsed Class IIB gene tree. In this case, all collapsed clades are labeled with 'Mam.' for mammals, because sequences from primates and mammal outgroups assort together by gene. (**B**) The expanded MHC-DPA clade (corresponding to the clade in panel A marked by a *). (**C**) The expanded MHC-DPB clade (marked by a † in panel A). (**D**) The expanded MHC-DRB clade (marked by a § in panel A). OWM: old-world monkeys; NWM: new-world monkeys; Cat.: Catarrhini—apes and OWM; Mam.: mammals—primates and other outgroup mammals.

The online version of this article includes the following source data and figure supplement(s) for figure 4:

**Source data 1.** GENECONV results for the Class II focused alignments.

**Figure supplement 1.** Class IIA multi-gene *BEAST2* trees.

**Figure supplement 2.** Class IIB multi-gene *BEAST2* trees.

**Figure supplement 3.** MHC-DRA multi-gene *BEAST2* trees.

**Figure supplement 4.** MHC-DQA multi-gene *BEAST2* trees.

**Figure supplement 5.** MHC-DPA multi-gene *BEAST2* trees.

**Figure supplement 6.** MHC-DMA multi-gene *BEAST2* trees.

**Figure supplement 7.** MHC-DOA multi-gene *BEAST2* trees.

**Figure supplement 8.** MHC-DRB multi-gene *BEAST2* trees.

**Figure supplement 9.** MHC-DQB multi-gene *BEAST2* trees.

**Figure supplement 10.** MHC-DPB multi-gene *BEAST2* trees.

**Figure supplement 11.** MHC-DMB multi-gene *BEAST2* trees.

**Figure supplement 12.** MHC-DOB multi-gene *BEAST2* trees.

MHC-G, -J, and -A (*Figure 3*). The uncertain early branching structure we observe in our trees may be due to these ancient gene conversion events.

## The importance of the pseudogenization process

Gene birth-and-death drives the evolution of a gene family as a whole. The 'death' can include the deletion of all or part of a gene from the genome or pseudogenization by means of inactivating mutations, which can leave gene remnants behind. In Class I, we find many pseudogenes that have been produced in this process; while countless more have undoubtedly already been deleted from primate genomes, many full-length and fragment pseudogenes still remain. Although non-functional, these sequences provide insight into the granular process of birth-and-death as well as improve tree inference.

Full Class I haplotypes including the pseudogenes are known only for human, chimpanzee, gorilla, and macaque, and even so we do not have sequences for *all* the balanced haplotypes in each species (*Anzai et al., 2003*; *Wilming et al., 2013*; *Shiina et al., 2017*; *Karl et al., 2023*). From these studies, we know that few functional Class I genes are shared by apes/OWMs and NWMs, and so far, no shared pseudogenes have been found (*Lugo and Cadavid, 2015*; *Kono et al., 2014*; *Cadavid et al., 1996*; *Maccari et al., 2017*; *Maccari et al., 2020*). Therefore, the Class I genes in the two groups have been generated by a largely separate series of duplications, neofunctionalizations, and losses. This means that turnover has occurred on a relatively short timescale, and understanding the pseudogenes within the apes and OWM can thus shed light on the evolution of the region more granularly. These ancient remnants could provide clues as to when genes or whole blocks were duplicated, which regions are more prone to duplication, and how the MHC may have functioned in ancestral species.

The Class I MHC region is further divided into three polymorphic blocks—α, κ, and β—that each contain MHC genes but are separated by well-conserved non-MHC genes (*Kulski et al., 2002*; *Dawkins et al., 1999*). The majority of the Class I genes are located in the α-block, which in humans includes 12 MHC genes and pseudogenes (*Shiina et al., 2017*). The α-block also contains a large number of repetitive elements and gene fragments belonging to other gene families, and their specific repeating pattern in humans led to the conclusion that the region was formed by successive block duplications (*Shiina et al., 1999*; *Kulski et al., 1997*; *Kulski et al., 2000*). Later, comparison of macaque and chimpanzee α-block haplotypes with the sequenced human haplotype bolstered this hypothesis, although the proposed series of events is not always consistent with phylogenetic data (*Kulski et al., 2005*; *Kulski et al., 2004*; *Geraghty et al., 1992*; *Hughes, 1995*; *Messer et al., 1992*; *Alexandrov et al., 2023*; *Gleimer et al., 2011*; see Appendix 3 for more detail). Improving existing theories about the evolution of this block is useful for disentangling the global pattern of MHC evolution from locus- and gene-specific influences. This could help us understand how selection on specific genes has affected entire linked regions. We therefore created an $\alpha$-block-focused tree involving sequences from more species than ever before in order to strengthen and update previous hypotheses about the evolution of the block, shown in *Figure 5*.

*Figure 5A* shows the Class I α-block-focused tree for exon 3, with an expanded MHC-V clade. MHC-V is a fragment pseudogene containing exons 1–3 which is located near MHC-F in the α-block. Previous work disagrees on the age of this fragment, with some suggesting it was fixed relatively early while others claiming it arose from one of the more recent block duplications (*Shiina et al., 1999*; *Kulski et al., 2005*; *Kulski et al., 2004*). Our tree groups ape and OWM MHC-V together and places them as an outgroup to all of the classical and non-classical genes, including those of the NWM. Thus, the MHC-V fragment may be an ancient remnant of one of the ancestral Class I genes. We also dispute the hypothesis that MHC-V (a 5'-end fragment) and -P (a 3'-end fragment) are since-separated pieces of the same original gene (*Horton et al., 2008*), as we found that both contain exon 3 and their exon 3 sequences clearly do not group together in our trees. Therefore, we support past work that has deemed MHC-V an old fragment.

We next focus on MHC-U, a previously uncharacterized fragment pseudogene containing only exon 3. In *Figure 5B*, we zoom in on the MHC-U clade within the exon 3 tree, corresponding to the asterisk in panel A. Our tree groups MHC-U with a clade of human, chimpanzee, and bonobo MHC-A, suggesting it duplicated from MHC-A in the ancestor of these three species. However, it is present on both chimpanzee haplotypes and nearly all human haplotypes, and we know that these haplotypes diverged earlier—in the ancestor of human and gorilla. Therefore, we presume that MHC-U will be found in the gorilla when more haplotypes are sequenced. Ours is the first work to show that MHC-U is actually an MHC-A-related gene fragment and that it likely originated in the human-gorilla ancestor.

Next, we expand the clade for MHC-K, a full-length pseudogene present in apes and OWM (*Figure 5C*). In humans, only MHC-K is present, but on some chimpanzee haplotypes, both MHC-K and its duplicate MHC-KL are present. In gorillas, haplotypes can contain either MHC-K or -KL, and in OWM, there are many copies of MHC-K as they are part of one of the basic block duplication units (*Figure 5—figure supplement 2*; *Karl et al., 2023*). These pieces of evidence suggest that MHC-K and -KL duplicated in the ancestor of the apes. Indeed, *Figure 5C* shows that MHC-K and -KL are closely related and OWM MHC-K groups outside of both, supporting that the duplication (which also copied MHC-W, -A, and -T) occurred after the split of apes and OWM. We did not detect MHC-K or -KL sequences in either the gibbon or orangutan reference genomes during our *BLAST* search, so we

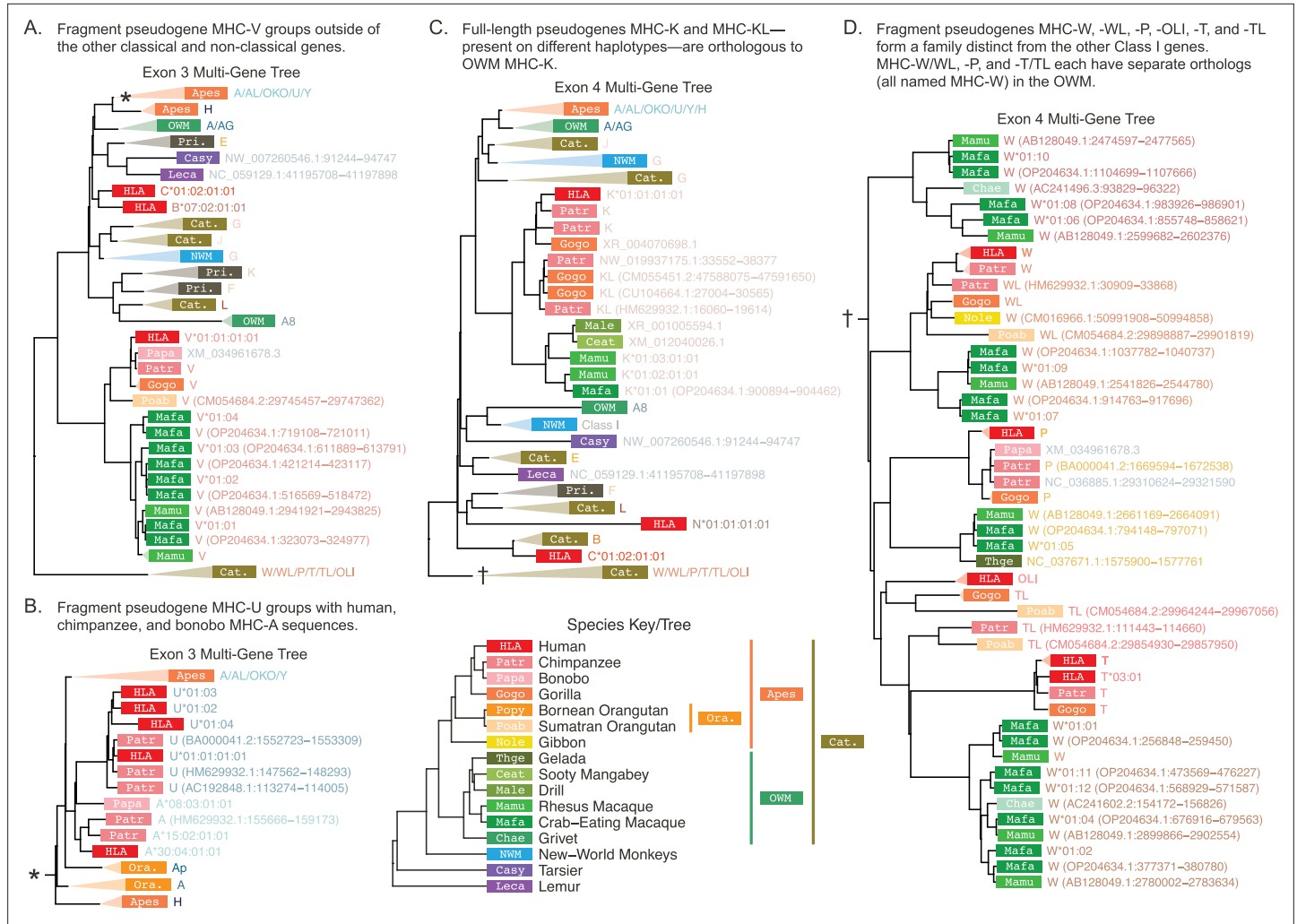

**Figure 5.** Class I α-block-focused multi-gene *BEAST2* trees. The α-block-focused trees use the common backbone sequences as well as additional sequences from our custom *BLAST* search of available reference genomes. For the purposes of visualization, some clades are collapsed and labeled with the species group and gene content of the clade (colored text to the right of each rectangle). The white labels on colored rectangles indicate the species group of origin, while the colored text to the right of each rectangle indicates the gene or sequence name (see Appendix 2). The species abbreviations are defined in the species key at the bottom. (**A**) Exon 3 α-block-focused *BEAST2* tree with expanded MHC-V clade. (**B**) The expanded MHC-A/AL/OKO/U/Y clade from the exon 3 tree (corresponding to the clade in panel A marked by a⋆), focusing on MHC-U. (**C**) Exon 4 α-block-focused *BEAST2* tree with expanded MHC-K/KL clade. (**D**) The expanded MHC-W/WL/P/T/TL/OLI clade from the exon 4 tree (marked by a † in panel **C**). OWM: old-world monkeys; NWM: new-world monkeys; Cat.: Catarrhini—apes and OWM; Pri.: Primates—apes, OWM, and NWM.

The online version of this article includes the following figure supplement(s) for figure 5:

**Figure supplement 1.** Class I α-block pseudogene-focused multi-gene *BEAST2* trees.

**Figure supplement 2.** All-pseudogene-focused multi-gene *BEAST2* trees.

cannot date this duplication event more precisely. The pseudogene may have been deleted from both genomes entirely, or it may be present on non-reference haplotypes. Sequencing of more haplotypes may help resolve the timing of this duplication event.

Another large group of related fragment pseudogenes in the Class I α-block includes MHC-W, -P, and -T (see Appendix 3 for more detail). Both our exon 3 and exon 4 trees indeed show a clear separation between the clade of MHC-W, -WL, -P, -T, -TL, and -OLI pseudogenes and the rest of the genes (*Figure 5A and C*). On the chromosome, members of these two Class I subgroups are inter-leaved throughout the α-block, suggesting that both groups are old and a series of block duplications occurred to form the current physical arrangement. Previous work on human sequences has shown that HLA-P, -W, -T, and -OLI are related (*Alexandrov et al., 2023*; *Hughes, 1995*; *Kulski et al., 2005*).

However, humans do not have orthologs of every single primate gene, so utilizing other primate sequences is critical to understanding this subfamily's evolution.

Thus, we next focus on the behavior of this subgroup in the trees. The MHC-W/WL/P/T/TL/OLI clade, marked with a † in *Figure 5C*, is expanded in panel D. We expected OWM MHC-W sequences to form a monophyletic clade either outside of all of the ape genes or with a single ape MHC gene, demonstrating orthology. Surprisingly, OWM MHC-W sequences instead formed four distinct clades, with one grouping with ape MHC-W/WL, one with ape MHC-P, one with ape MHC-T/TL, and one outside of all. Furthermore, based on the alleles present, each of these OWM MHC-W clades corresponds to a type of basic repeat block (as revealed by the published macaque MHC haplotype; *Karl et al., 2023*; *Kulski et al., 2004*). The correspondence between the distinct OWM MHC-W clades and the sequences' physical locations on a haplotype lends further support to the hypothesis that macaque haplotypes were generated by tandem duplications. Additionally, the fact that the different OWM MHC-W clades each group with a different ape pseudogene suggests that there are actually three ape/OWM orthologous groups (see Appendix 3 for further explanation). Thus, for the first time, we show that there must have been three distinct MHC-W-like genes in the ape/OWM ancestor.

We also learned more about HLA-OLI, a recently discovered MHC pseudogene found on the same insertion segment that carries HLA-Y in a small fraction of the human population. Its discoverers analyzed only human sequences, finding that HLA-OLI was most similar (88%) to HLA-P (*Alexandrov et al., 2023*). Our inclusion of non-human primate genes revealed that HLA-OLI is actually most similar in both structure and sequence to MHC-TL, a gene not found in humans and thus not included in the previous analysis (*Figure 5—figure supplement 1*). Furthermore, since MHC-Y and -OLI are fully linked in humans and are located in close proximity, it is likely that they duplicated as a unit. Because MHC-Y is similar to MHC-AL/OKO and HLA-OLI is similar to MHC-TL, we hypothesize that they duplicated together from the latter genes, which are adjacent to each other on non-human haplotypes. MHC-Y has also been identified in gorillas (Gogo-Y; *Hans et al., 2017*), so we anticipate that Gogo-OLI will soon be confirmed. This evidence suggests that the MHC-Y and -OLI-containing haplotype is at least as old as the human-gorilla split. Our study is the first to place MHC-OLI in the overall story of MHC haplotype evolution.

With these findings, in addition to many other observations from our trees and results from past literature (references in *Figure 6—figure supplement 1*), we propose a new hypothesis for the evolution of the Class I α-block. *Figure 6* shows a possible evolutionary path for α-block haplotypes that could have led to the currently observed haplotypes. Haplotypes found so far in each species are at the bottom of the figure (with additional never-before-reported haplotypes from our *BLAST* search shown in *Figure 1—figure supplement 2*). In particular, our work has revealed that MHC-V is an old fragment, three MHC-W-like genes were already established at the time of the ape/OWM ancestor, MHC-U is closely related to African ape MHC-A, and MHC-OLI is closely related to MHC-TL. Additionally, the OWM MHC-A fragment pseudogene is actually more similar to the ape MHC-A genes than to the other OWM MHC-A genes (*Figure 5—figure supplement 1C*), supporting the existence of two MHC-A-like genes in the ape/OWM ancestor. Appendix 3 explains the pieces of evidence leading to all of these conclusions (and more!) in more detail.

## Evolution of the MHC-DRB region

The Class I vs. Class II division reflects a major functional distinction within the MHC gene family, but even within these subfamilies, evolution is not homogeneous. Among the Class II genes, there are few duplicated genes and generally only one way for the protein products to pair, for example MHC-DPA1 with MHC-DPB1. However, the MHC-DR genes are a notable exception to the general pattern; MHC-DRA can pair with any of the multiple functional MHC-DRB genes. In addition, MHC-DRB has many more related pseudogenes compared to the rest of the Class II genes, making the MHC-DR region's pattern of evolution more reminiscent of Class I (see *Figure 2* to see the varied landscape of MHC-DRB genes in different species). We explored the evolution of the MHC-DRB region in greater detail by creating focused trees with a larger set of MHC-DRB sequences.

In exon 2, which codes for the binding site, the MHC-DRB genes group mostly by name (e.g. MHC-DRB3) across apes, OWM, and NWM (*Figure 4—figure supplement 8A*). The exon 2 tree considered alone thus suggests that the genes are orthologous across apes, OWM, and NWM—which is how the genes were named in the first place. However, looking at this exon alone does not give us a complete

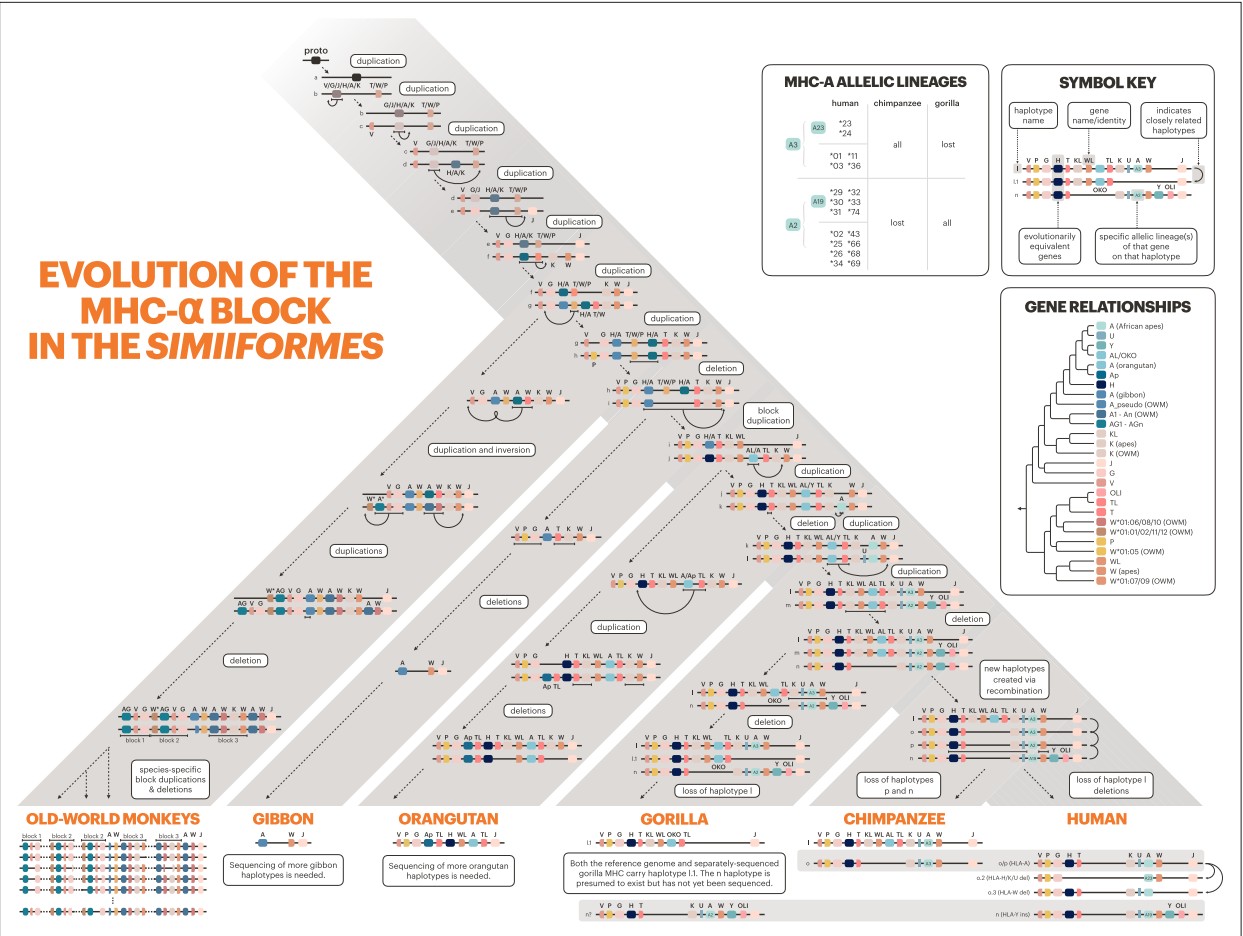

**Figure 6.** Evolution of the Class I $\alpha$-block. The primate evolutionary tree is shown in gray (branches not to scale). The bottom of the tree shows currently known haplotypes in each species or species group. Horizontal gray bars indicate haplotypes shared among the African apes. The history of the genes/haplotypes in the $\alpha$-block is overlaid on the tree, synthesizing previous work with our own observations (see Methods, **Figure 8**, and Appendix 3 for explanations and citations). Genes are represented by colored rectangles, while haplotypes are shown as horizontal lines containing genes. MHC-F—marking the telomeric end of the $\alpha$-block—was fixed early on and is located immediately to the left on all haplotypes shown, but is not pictured due to space constraints. Dashed arrows with descriptive labels represent evolutionary events. In the upper right, the 'Symbol Key' explains the icons and labels. The 'Gene Relationships' panel shows the relationships between the loci shown on the tree, without the layered complexity of haplotypes and speciation events. The 'MHC-A Allelic Lineages' panel shows which MHC-A allele groups are present in human, chimpanzee, and gorilla.

The online version of this article includes the following figure supplement(s) for figure 6:

**Figure supplement 1.** Evolution of the Class I $\alpha$-block (with references).

picture. Exon 3 does not encode the binding site and is less likely to be affected by convergent evolution; its tree is shown in *Figure 4—figure supplement 8B*. In this tree, all NWM sequences group together (clade with blue boxes about halfway up the tree) instead of with other ape/OWM sequences, suggesting that the genes expanded separately in NWM and apes/OWM. Additionally, OWM MHC-DRB1 and -DRB3 form their own clade (green boxes near the top of the tree), and OWM MHC-DRB4 sequences group outside of several ape and NWM clades. We see that only three ape/OWM MHC-DRB genes/pseudogenes (MHC-DRB5, -DRB2/6, and -DRB9) form monophyletic clades, indicating that these three are the only orthologous MHC-DRB genes. Further, none are 1:1 orthologous to any particular NWM gene. Thus, the longevity of individual MHC-DRB genes in the primates appears to be less than 38 million years.

The longevity of MHC-DRB haplotypes is even shorter. Only one haplotype is shared between human and chimpanzee, and none are shared with gorilla (*de Groot et al., 2009*; *Heijmans et al., 2020*; *Hans et al., 2015*). This shows that the region is evolving even more rapidly than Class I (where haplotypes are shared among human, chimpanzee, and gorilla; *Figure 6*). These haplotypes,

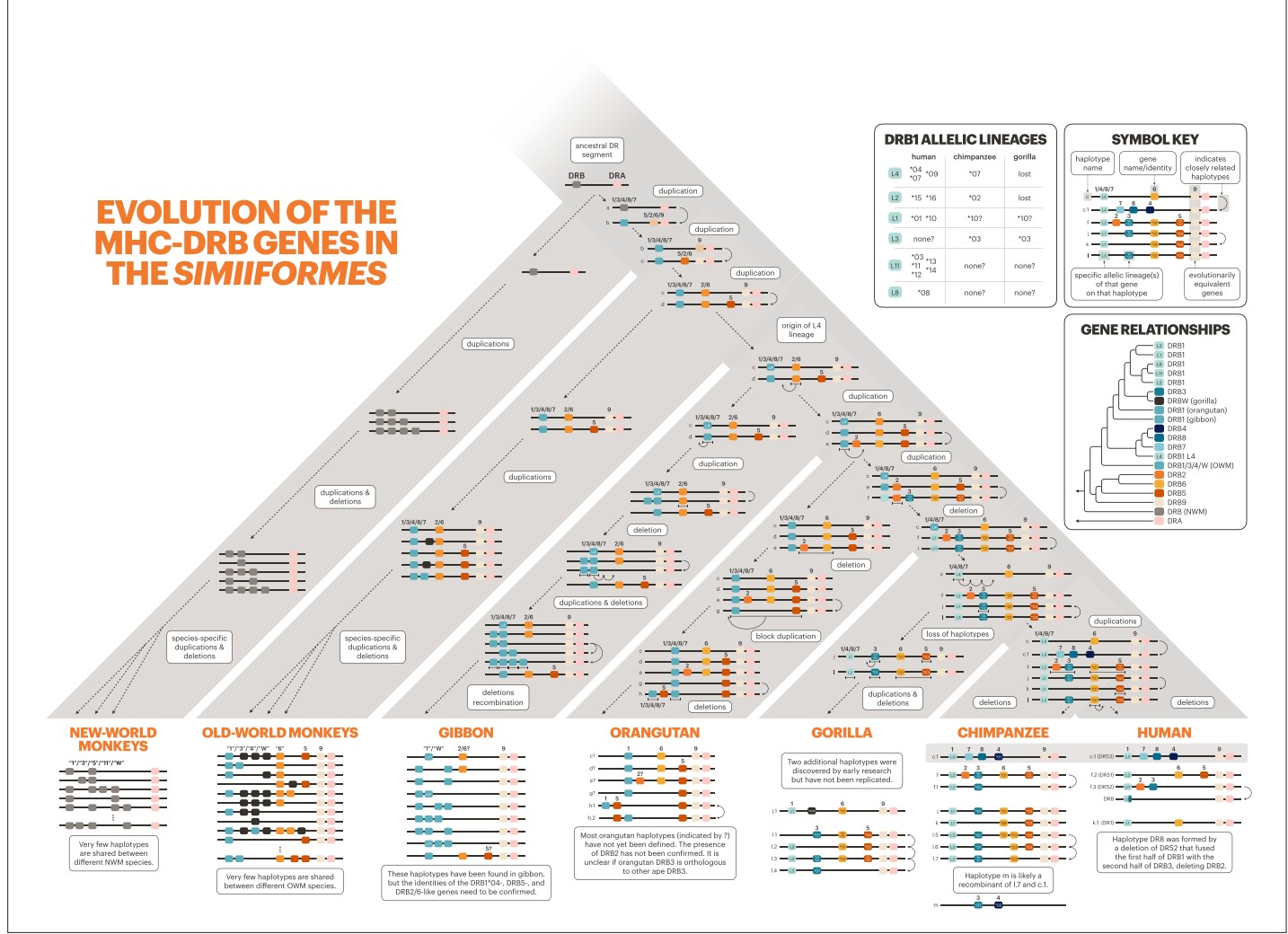

**Figure 7.** Evolution of MHC-DRB. The bottom of the tree shows current haplotypes in each species or species group; human, chimpanzee, gorilla, and old-world monkey haplotypes are well characterized, while orangutan, gibbon, and new-world monkey haplotypes are partially known. The history of the genes/haplotypes in the MHC-DRB region is overlaid on the tree, synthesizing previous work with our own observations (see Materials and methods and *Figure 8*). The rest of the figure design follows that of *Figure 6*.

The online version of this article includes the following figure supplement(s) for figure 7:

**Figure supplement 1.** Evolution of the MHC-DRB region (with references).

combined with past literature (cited in *Figure 7—figure supplement 1*) and our trees, allowed us to trace backward and propose a hypothesis for the evolution of the region, shown in *Figure 7*.

*Figure 7* shows plausible steps that might have generated the current haplotypes and patterns of variation that we see in present-day primates. However, some species are poorly represented in the data, so the relationships between their genes and haplotypes are somewhat unclear. In our exon 3 tree (*Figure 4—figure supplement 8B*), orangutan alleles do not group definitively with any other ape lineages. Furthermore, the orangutan MHC-DRB3 gene groups with orangutan MHC-DRB1, suggesting that it may not be orthologous to the African ape MHC-DRB3 gene. We also found an orangutan sequence that groups with the human HLA-DRB2 pseudogene, suggesting that this gene has an ortholog in the orangutan. Several haplotypes have been previously identified in the gibbon, but since they rely on exon 2 sequence alone, it is unclear how these alleles relate to the known ape lineages (*de Groot et al., 2017b*). Analysis of more orangutan and gibbon haplotypes will be essential for understanding how the region has evolved in the apes.

Overall, the MHC-DRB genes are not evolving in the same fashion as the rest of the Class II genes, even though they have a shared structure and function. This peculiar case illustrates that there are multiple ways to achieve a functional immune response from the same basic parts.

## Differences between MHC subfamilies

We explored the evolution of the Class I and Class II genes separately and noticed several differences between the classes. First, sequences group by gene rather than by species group in the Class II gene trees (*Figure 4*, *Figure 4—figure supplements 1 and 2*). Our inclusion of RefSeq sequences from distant groups of placental mammals confirms that most of the primate Class II genes have maintained orthology at least since the ancestor of placentals, 105 million years ago (*Foley et al., 2023*). In contrast, our Class I trees (*Figure 3*, *Figure 3—figure supplement 2*) showed sequences more often grouping by species group than by gene, indicating that the genes turn over quickly and 1:1 orthology is often lost. Only non-classical MHC-F (and possibly MHC-E) are truly orthologous among the apes, OWM, and NWM, consistent with previous findings (*Piontkivska and Nei, 2003*; *Adams and Parham, 2001a*; *Sawai et al., 2004*). Additionally, our tarsier and *Strepsirrhini* sequences group outside of all *Simiiformes* Class I sequences, setting an upper bound on the maintenance of Class I orthology of 58 million years (*Kuderna et al., 2023*; *Flügge et al., 2002*).

This turnover of genes at the MHC—rapid for Class I and slower for Class II—is generally believed to be due to host-pathogen co-evolution (*Radwan et al., 2020*). The MHC genes are critically important for survival, yet no single gene is so vital that its role must be preserved. For example, in the apes, the MHC-G gene is non-classical, but in the OWM, it has been inactivated and its role largely replaced by an MHC-A-related gene called MHC-AG (*Heijmans et al., 2020*). This process of turnover ultimately results in different sets of MHC genes being used in different lineages. For instance, separate expansions generated the classical Class I genes in NWM (all called MHC-G) and the $\alpha$-block genes in apes/OWM. Similarly, separate expansions generated the MHC-DRB genes of the NWM and of the apes/OWM. Aside from MHC-DRB, the other Class II genes have been largely stable across the mammals, although we do see some lineage-specific expansions and contractions (*Figure 2*, *Figure 2—figure supplement 2*).

Class I and Class II also differ in their degree of gene conversion. Our GENECONV analysis revealed two types of gene conversion events: (1) specific, more-recent events involving paralogous genes or particular allelic lineages and (2) broad-scale, very-old events involving two dissimilar loci (*Figure 3—source data 1*, *Figure 4—source data 1*). We discovered far more 'specific' events in Class I, while 'broad-scale' events were predominant in Class II. This could reflect the different age of these gene groups: Class I genes turn over more rapidly and allelic lineages are less diverged from each other, making gene conversion more likely. In contrast, Class II genes have much longer-lived (and more-diverged) allelic lineages, potentially explaining why we mainly picked up older events in the Class II GENECONV analysis.

The non-classical vs. classical distinction is another functionally meaningful way to partition the genes. The classical genes (of both classes) perform peptide presentation to T-cells, making them direct targets of host-pathogen co-evolution. In contrast, the non-classical genes are involved in innate immune surveillance or niche roles and may be less directly affected by this co-evolution. In our trees, sequences from non-classical genes of both classes often group by gene with tree topology matching the species tree, while sequences from classical genes do neither (*Figure 3—figure supplement 2*, *Figure 4—figure supplements 1 and 2*). We further explore the differences between classical and non-classical genes in our companion paper, finding ancient trans-species polymorphism at the classical genes but not at the non-classical genes (*Fortier and Pritchard, 2025*). These pieces of evidence show that classical genes experience more turnover and are more often affected by long-term balancing selection or convergent evolution. Ultimately, selection acts upon functional differences between classical and non-classical genes in a manner that is largely independent of whether they belong to Class I or Class II, although the classes differ in their rate of evolution.

## Discussion

The MHC proteins serve diverse roles in innate and adaptive immunity (*Adams and Luoma, 2013*). They are critically important to infection resistance, autoimmune disease susceptibility, and organ

transplantation success and can provide insight into human evolution, inform disease studies, and improve upon non-human-primate disease models (*Kennedy et al., 2017*). Despite their varied functions, all Class I and Class II MHC genes are derived from a common ancestor, allowing us to compare genes to learn more about the evolution of the gene family as a whole (*Hansen et al., 2007*; *Kupfermann et al., 1999*; *Kaufman, 2022*; *Adams and Luoma, 2013*). A few ~20-year-old studies addressed the overall evolution of the MHC gene family via multi-gene alignment and phylogenetics, but the trees had many polytomies (*Adams and Parham, 2001a*; *Sawai et al., 2004*; *Cardenas et al., 2005*; *Piontkivska and Nei, 2003*; *Takahashi et al., 2000*). Since then, most work has focused on particular genes or small sets of species, meaning our knowledge of primate MHC evolution is scattered across hundreds of papers (*Urvater et al., 2000*; *van der Wiel et al., 2013*; *Geller et al., 2002*; *Hans et al., 2017*; *Maibach et al., 2017*; *Wroblewski et al., 2017*; *Wroblewski et al., 2019*; *Lafont et al., 2004*; *Flügge et al., 2002*; *Go et al., 2005*; *Go et al., 2003*; *Shiina et al., 2017*; *Abi-Rached et al., 2010*; *Gleimer et al., 2011*; *de Groot et al., 2015*; *de Groot et al., 2012*; *de Groot et al., 2022*; *Cao et al., 2015*; *Otting et al., 2020*; *Fukami-Kobayashi et al., 2005*; *Lugo and Cadavid, 2015*; *Averdam et al., 2011*; *Figueroa et al., 1994*; *Doxiadis et al., 2012*; *Buckner et al., 2021*; *Doxiadis et al., 2006*; *Diaz et al., 2000*; *Gongora et al., 1997*; *Kasahara et al., 1992*; *de Groot et al., 2009*; *Satta et al., 1996*). In this project, we revisited primate MHC evolution with more data from a wider range of species and a coherent analysis framework. We confirm and unify past findings, as well as contribute many new insights into the evolution of this complex family.

We found that the Class I genes turn over rapidly, with only the non-classical gene MHC-F being clearly orthologous across the *Simiiformes*. In the rest of the Class I $\alpha$-block, genes expanded entirely separately in the ape/OWM and NWM lineages. This process of expansion generated many full-length and fragment pseudogenes, which we found were equally important as the functional genes to understanding the evolution of the region as a whole. Specifically, we found that MHC-U is an MHC-A-related pseudogene, MHC-V is not closely related to MHC-P, and that there were at least three genes of the MHC-W/P/T/OLI family present in the ape/OWM ancestor. Including these pseudogenes in our trees helped us construct a new model of α-block haplotype evolution.

Generally, Class II genes do not turn over as rapidly as Class I genes, although there were exceptions. The classical MHC-DRB genes were even shorter-lived than the Class I genes, with most human MHC-DRB genes lacking 1:1 orthologs beyond the great apes. We also found that the classical MHC-DQA and -DQB genes were not as clearly orthologous across the primates as we expected; rather, they likely expanded separately in the ape/OWM and NWM lineages. In contrast, the classical MHC-DPA and -DPB genes were orthologous across the *Simiiformes*, and the non-classical Class II genes were 1:1 orthologous across most of the mammals we included. In both Class I and Class II, classical genes turned over more rapidly than non-classical genes and their trees exhibited more deviations from the expected species tree. Overall, our treatment of the genes as related entities instead of distinct cases helped us understand shared patterns of evolution across classes and species groups.

While there are clear differences in evolutionary rate between different subsets of the MHC gene family, it is unclear how rapidly the overall family has evolved compared to other immune and non-immune gene families. Over the evolution of the placental mammals, chromosomal breakpoints are more often located near immune genes (*Muffato et al., 2023*), and the MHC region is significantly structurally divergent within the primates (*Mao et al., 2024*). In one study of human, chimpanzee, and macaque gene families, MHC Class I genes showed significantly accelerated rates of evolution and were among the most-rapidly evolving gene families; however, many non-immune gene families were also identified (*Hahn et al., 2007*). One hypothesis is that the MHC might evolve more rapidly than other gene families because many of its members have direct interaction with pathogen peptides, can bind with multiple different peptides, are expressed on the cell surface rather than in the cytosol, and are constitutively expressed rather than induced; however, more evidence is needed to support this rationale (*Vinkler et al., 2023*). There are many other large immune-related gene families in vertebrates, such as killer Ig-like receptors (KIRs), leukocyte Ig-like receptors (LILRs), sialic acid-binding Ig-type lectin receptors, Toll-like receptors (TLRs), and NOD-like receptors (NLRs; *Vinkler et al., 2023*). In the future, performing a comparable analysis on these other families will provide insight into whether our observations are unique to the MHC, are representative of immune gene families, or translate to gene families in general.

One concern when discussing gene families is the relative importance of birth-and-death and concerted evolution by gene conversion (*Gu and Nei, 1999*; *Nei and Rooney, 2005*; *Klein et al., 2007*; *Bergström and Gyllensten, 1995*; *Gyllensten et al., 1991*; *Nei et al., 1997*). Gene conversion can cause adjacent small sequence tracts to have wildly different evolutionary histories, making it difficult to interpret a tree constructed from larger regions. Our phylogenetic analyses reveal different tree topologies depending on exon, and our GENECONV analysis pulled out several different sequence pairs, revealing that gene conversion has played a significant role in the evolution of the MHC genes. With this in mind, comparing trees across exons helps us interpret the overall trees and strengthens our conclusions. Neither birth-and-death nor concerted evolution can be ignored when discussing gene families.

Short-read sequencing has long been the primary approach for generating MHC sequence data (*Cheng et al., 2022*). However, the MHC region is difficult to assemble owing to the large number of related genes, extreme polymorphism, and abundant repetitive regions (*Heijmans et al., 2020*; *Cheng et al., 2022*). In addition, the extreme diversity of MHC haplotypes within some species has not been appreciated until recently (*de Groot et al., 2017b*; *de Groot et al., 2024*; *de Groot et al., 2015*; *Gleimer et al., 2011*; *Hans et al., 2015*; *Heijmans et al., 2020*). To understand MHC evolution in the primates, it is imperative to fully characterize the many genes and haplotypes present in each species. Better MHC maps will allow us to estimate gene gain and loss rates, pinpoint orthologs across species, and understand how individual MHC repertoires translate to different functional responses. Long-read sequencing is already starting to gain traction in the MHC world, helping to resolve even the most complex haplotypes. For humans, high-quality MHC sequences have already been created using the Oxford Nanopore and PacBio HiFi methods (*Wenger et al., 2019*; *Jain et al., 2018*; *Liu, 2021*; *Bruijnesteijn, 2023*). Outside of humans, long-read sequencing has also been applied to the MHC regions of the Tasmanian devil, horse, yellow cardinal, duck, and macaque (*Cheng et al., 2022*; *Domínguez et al., 2025*; *Hu et al., 2024*; *de Groot et al., 2024*; *Karl et al., 2023*; *Viļuma et al., 2017*). Studying more MHC regions in even more species is needed to understand the myriad evolutionary strategies for a successful immune response. Long-read MHC sequencing across a rich array of primates will also prove essential. This data will help us answer evolutionary questions with better precision as well as provide necessary context for infectious and autoimmune disease pathogenesis in humans.

By treating the MHC genes as a gene family and including more data than ever before, this work enhances our understanding of the evolutionary history of this remarkable region. Our extensive set of trees incorporating classical genes, non-classical genes, pseudogenes, gene fragments, and alleles of medical interest across a wide range of species will provide context for future evolutionary, genomic, disease, and immunologic studies. For example, this work provides a jumping-off point for further exploration of the evolutionary processes affecting different subsets of the gene family and the nuances of immune system function in different species. This study also provides a necessary framework for understanding the evolution of particular allelic lineages within specific MHC genes, which we explore further in our companion paper (*Fortier and Pritchard, 2025*). Both studies shed light on MHC gene family evolutionary dynamics and bring us closer to understanding the evolutionary trade-offs involved in MHC disease associations.

## Materials and methods
### Data collection

We downloaded MHC allele nucleotide sequences for all human and non-human genes from the IPD Database (collected January 2023; *Barker et al., 2023*; *Maccari et al., 2017*; *Maccari et al., 2020*; *Robinson et al., 2024*). To supplement the alleles available in the database, we also collected nucleotide sequences from NCBI using the Entrez E-utilities with query 'histocompatibility AND txidX AND alive[prop]', where X is a taxon of interest. This resulted in a very large collection of sequences from a large number of species. While Class II genes were generally assigned to loci, most Class I sequences had ambiguous or no locus assignments. Therefore, we performed a refined search for additional sequences by running *BLAST* on the available primate reference genomes (GenBank accession numbers listed in *Table 1*).

**Table 1.** GenBank accession numbers for reference genomes used in this study.

Accessions point to the MHC-containing chromosome (or partial chromosome) from each genome.

| Species | Chromosome | GenBank Accession |
|---|---|---|
| Human | 6 | CM000668.2 |
| Chimpanzee | 5 | CM054439.2 |
| Bonobo | 5 | CM055477.2 |
| Gorilla | 5 | CM055451.2 |
| Sumatran Orangutan | 5 | CM054684.2 |
| Bornean Orangutan | 5 | CM054635.2 |
| Pileated Gibbon | linkage group LG22 | CM038537.1 |
| Siamang | 23 | CM054531.2 |
| Northern White-Cheeked Gibbon | 22 a | CM016966.1 |
| Olive Baboon | 6 | CM018185.2 |
| Guinea Baboon | 6 | CM053423.1 |
| Gelada | 4 | CM009953.1 |
| Tibetan Macaque | 4 | CM045091.1 |
| Crab-Eating Macaque | 4 | CP141358.1 |
| Formosan Rock Macaque | 4 | CM049490.1 |
| Mantled Guereza | 5 | CM058078.1 |
| Snub-Nosed Monkey | 4 | CM017354.1 |
| Cotton-top Tamarin | 4 | CM063172.1 |
| Golden-handed Tamarin | linkage group LG04 | CM038394.1 |
| Common Marmoset | 4 | CM021918.1 |
| Coppery Titi | 3 | CM080817.1 |
| Gray Mouse Lemur | 6 | CM007666.1 |
| Black-and-white Ruffed Lemur | 6 | CM052441.1 |
| Mongoose Lemur | 15 | CM052867.1 |
| Ring-tailed Lemur | 2 | CM036473.1 |
| Bengal Slow Loris | linkage group LG08 | CM043617.1 |
| Sunda Slow Loris | 9 | CM050145.1 |
| Philippine Flying Lemur | 5 | CM050031.1 |
| Mouse | 17 | CM001010.3 |

## BLAST search

To create the *BLAST* database, we first compiled all nucleotide MHC sequences from the IPD-MHC and IPD-IMGT/HLA databases into three fasta files: one containing the Class I sequences, one containing the Class II sequences, and one containing MHC-DRB9 sequences (see note below). We then constructed three custom databases from these sets of sequences using the `makeblastdb` command in *BLAST* version 2.11.0 (*Camacho et al., 2009*).

We then queried each of the three custom databases using the above reference genomes and screened the hits manually. This manual step was necessary because the reference sequences included highly similar genes that needed to be carefully teased apart; in addition, the MHC region contains many small, repeated fragments of genes that could show up as hits. We looked for both high sequence identity and a long alignment length while keeping in mind synteny and our expectations

for reasonable alignment lengths and sequence identities (which differ by gene subgroup and the species being compared). In most cases, we were able to identify loci unambiguously, resulting in several newly reported haplotypes (*Figure 1—figure supplement 2*, *Figure 2—figure supplement 2*). The discovery of various genes in various species also allowed us to fill in gaps in *Figures 1 and 2*.

We generated a separate *BLAST* database for the MHC-DRB9 sequences because MHC-DRB9 was not reliably detected when *BLAST*ing the genomes against the all-Class II database. MHC-DRB9 is a partial-length pseudogene (aligning to just one exon), and it is more diverged from the other MHC-DRB sequences. As a result, queries to the all-Class II *BLAST* database only found the pseudogene occasionally (usually matching it to a different MHC-DRB gene with poor overall score or alignment length). To be sure that we were detecting MHC-DRB9 in particular (as opposed to other MHC-DRB genes or motifs across the region) and consistently across all of the primate genomes, we set up a separate *BLAST* database containing only MHC-DRB9 sequences as references. We queried the genomes against this database (in addition to the other databases) to maximize the number of MHC-DRB9 sequences we could find.

## Sequence selection

Because *BEAST2* is computationally limited by the number of sequences, it was necessary to prioritize certain sequences. To do this, we (very roughly) aligned as many exon 2 and 3 sequences as possible (from both NCBI RefSeq and the IPD database) using MUSCLE (*Edgar, 2004*) with default settings. We then constructed UPGMA trees in R to visualize the sequences. We preferentially selected sequences that were (1) in primate species not represented by the IPD database or (2) grouped with genes not well represented by the IPD database, and which were not similar/identical to other sequences. We also included several non-primate species to provide context and explore orthology beyond the primates. After choosing sequences with this preliminary screening method, we collected the full-length sequences for inclusion in further analyses. We limited sequences to one per species-gene pair for building the Class I, Class IIA, and Class IIB multi-gene trees (lists of alleles provided as Supplementary Files).

For Class I, we then re-aligned all genes together for each exon separately using MUSCLE (*Edgar, 2004*) with default settings (and manually adjusted). For Class II, alleles for each gene group (MHC-DMA, -DMB, -DOA, -DOB, -DPA, -DPB, -DQA, -DQB, -DRA, and -DRB) were aligned separately for each exon using MUSCLE (*Edgar, 2004*) with default settings (and manually adjusted). Since some Class II genes are too far diverged from one another to be reliably aligned automatically, the nucleotide alignments were then combined manually based on published amino acid alignments (*Radley et al., 1994*; *Dijkstra et al., 2013*; *Dijkstra and Yamaguchi, 2019*; *Cuesta et al., 2006*; *Chen et al., 2006*; *Chazara et al., 2011*). For Class IIA, exons 4 and 5 were concatenated together before this manual combination process because some analogous sites between genes are located across exons. For the same reason, exons 5 and 6 were concatenated together for Class IIB before combining. This produced three multi-gene alignments: Class I, Class IIA, and Class IIB.

We also aligned a larger set of sequences for each gene group to create our 'focused' trees that each zoomed in on a different subtree of the multi-gene trees. Details for this are located in the Methods of our companion paper (*Fortier and Pritchard, 2025*).

## Bayesian phylogenetic analysis

We constructed phylogenetic trees using *BEAST2* (*Bouckaert et al., 2014*; *Bouckaert et al., 2019*) with package substBMA (*Wu et al., 2013*). SubstBMA implements a spike-and-slab mixture model that simultaneously estimates the phylogenetic tree, the number of site partitions, the assignment of sites to partitions, the nucleotide substitution model, and a rate multiplier for each partition. Since we were chiefly interested in the partitions and their rate multipliers, we used the RDPM model as described by *Wu et al., 2013*. In the RDPM model, the number of nucleotide substitution model categories is fixed to 1, so that all sites, regardless of rate partition, share the same estimated nucleotide substitution model. This reduces the number of parameters to be estimated and ensures that only evolutionary rates vary across site partitions, reducing overall model complexity. We used an uncorrelated lognormal relaxed molecular clock because we wanted evolutionary rates to be able to vary among branches.

## Priors

For the Dirichlet process priors, we used the informative priors constructed by *Wu et al., 2013* for their mammal dataset. This is appropriate because they include several of the same species and their mammals span approximately the same evolutionary time that we consider in our study. We also use their same priors on tree height, base rate distribution, and a Yule process coalescent prior. We did not specify a calibration point—a time-based prior on a node—because we did not expect our sequences to group according to the species tree.

## Running *BEAST2*

We ran *BEAST2* on various subsets of the three alignments. Considering exons separately helped to minimize the effects of recombination on the tree, while also allowing us to compare and contrast tree topologies for exons encoding the binding site vs. exons encoding the other domains. For Class I, we repeated the analysis for (1) exon 2 only (PBR), (2) exon 3 only (PBR), and (3) exon 4 only (non-PBR). For Class IIA, we used (1) exon 2 only (PBR) and (2) exon 3 only (non-PBR). For Class IIB, we analyzed (1) exon 2 only (PBR) and (2) exon 3 only (non-PBR). In the following, each 'analysis' refers to a collection of *BEAST2* runs using a particular subset of either the Class I, Class IIA, or Class IIB alignment. The procedure is exactly the same for the 'focused' trees, which each focus on a particular gene group within the Class I, Class IIA, or Class IIB alignment. More detail about the generation of the focused trees is located in the Materials and methods of our companion paper (*Fortier and Pritchard, 2025*).

The XML files we used to run *BEAST2* were based closely on those used for the mammal dataset with the RDPM model and uncorrelated relaxed clock in *Wu et al., 2013* (https://github.com/jessiewu/substBMA/blob/master/examples/mammal/mammal_rdpm_uc.xml; *Vaughan, 2016*). Running a model with per-site evolutionary rate categories and a relaxed clock means there are many parameters to estimate. Along with the large number of parameters, highly polymorphic and highly diverged sequences make it difficult for *BEAST2* to explore the state space. Thus, we undertook considerable effort to ensure good mixing and convergence of the chains. First, we employed coupled MCMC for all analyses. Coupled MCMC is essentially the same as the regular MCMC used in *BEAST2*, except that it uses additional 'heated' chains with increased acceptance probabilities that can traverse unfavorable intermediate states and allow the main chain to move away from an inferior local optimum (*Müller and Bouckaert, 2020*). Using coupled MCMC, both speeds up *BEAST2* runs and improves mixing and convergence. We used four heated chains for each run with a delta temperature of 0.025. Second, we ran each BEAST2 run for 40,000,000 states, discarding the first 4,000,000 states as burn-in and sampling every 10,000 states. Third, we ran at least eight independent replicates of each analysis. The replicates use the exact same alignment and coupled MCMC settings but explore state space independently and thus are useful for improving the effective sample size of tricky parameters. As recommended by *BEAST2*, we examined all replicates in Tracer version 1.7.2 (*Rambaut et al., 2018*) to ensure that they were sampling from the same parameter distributions and had reached convergence. We excluded replicates for which this was not true, as these chains were probably stuck in suboptimal state space. Additionally, Tracer provides estimates of the effective sample size (ESS) for the combined set of states from all chosen replicates, and we required that the combined ESS be larger than 100 for all parameters. If there were fewer than 4 acceptable replicates or if the ESS was below 100 for any parameter, we re-ran more independent replicates of the analysis until these requirements were satisfied. We obtained between 7 and 14 acceptable replicates (median 8) per analysis for the Class I, Class IIA, and Class IIB runs.

For some analyses, computational limitations prevented *BEAST2* from being able to reach 40,000,000 states. In these situations, more replicates (of fewer states) were usually required to achieve good mixing and convergence. Regardless of how far these *BEAST2* runs got, the first 4,000,000 states from each run were still discarded as burn-in even though this represented more than 10% of states. The XML files required to run all our analyses are provided as Supplementary Files.

This extremely stringent procedure ensured that all of the replicates were exploring the same parameter space and were converging upon the same global optimum, allowing the ≥ 4 independent runs to be justifiably combined. We combined the acceptable replicates (discarding the first 4,000,000 states as burn-in) using *LogCombiner* version 2.6.7 (*Drummond and Rambaut, 2007*), which aggregates the results across all states. We then used the combined results for downstream analyses.

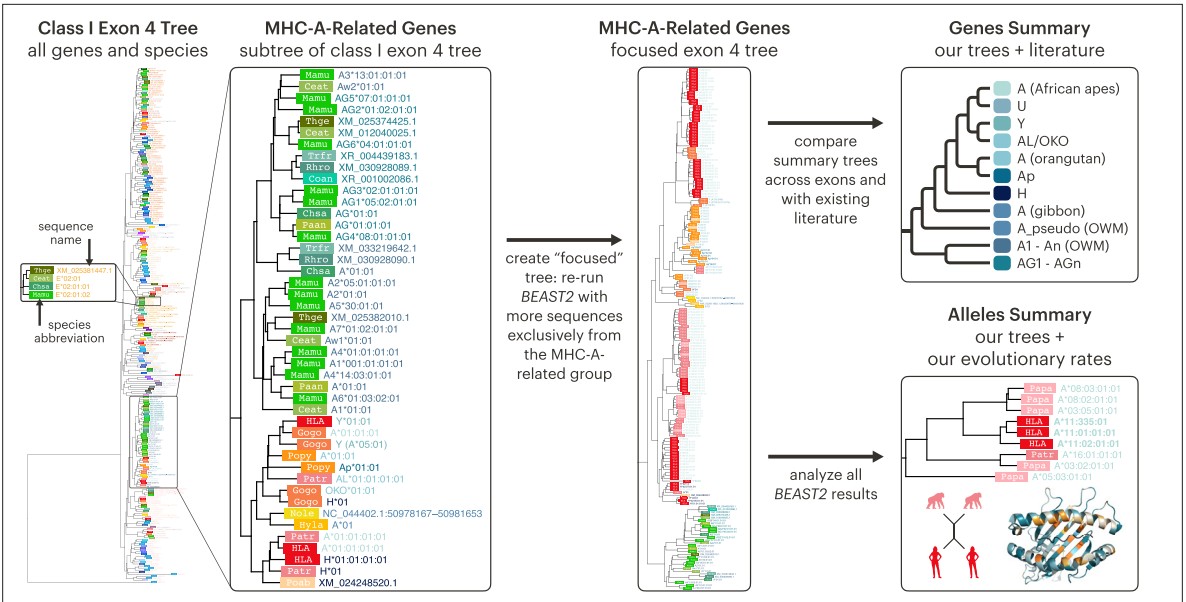

**Figure 8.** *BEAST2* trees provide insight into MHC gene and allele relationships. We first created multi-gene Bayesian phylogenetic trees using sequences from all genes and species, separated into Class I, Class IIA, and Class IIB groups. We then focused on various subtrees of the multi-gene trees by adding more sequences for each subtree and running *BEAST2* using only sequences from that group (in addition to the 'backbone' sequences common to all trees). Our trees gave us insight into both overall gene relationships (this paper) and allele relationships within gene groups (see our companion paper, *Fortier and Pritchard, 2025*).

## Phylogenetic trees

After combining acceptable replicates, we obtained 17,927–28,384 phylogenies per gene/sequence subset for the Class I, Class IIA, and Class IIB trees (mean 25,154). We used TreeAnnotator version 2.6.3 (*Drummond and Rambaut, 2007*) to summarize each set of possible trees as a maximum clade credibility tree, which is the tree that maximizes the product of posterior clade probabilities. Since *BEAST2* samples trees from the posterior, one could in principle reduce the large set of trees to a smaller 95% credible set of trees representing the 'true' tree (*BEA, 2024*). However, given the high complexity of the model space, all our posterior trees were unique, meaning this was not possible in practice. Throughout this paper, we rely on summary trees for our observations.

## Integration with literature

Hundreds of authors have contributed to the study of MHC evolution, and their myriad published results played a key role in this project. *Figure 8* illustrates our approach to this project, including how we used existing literature and how we divided results among this paper and its companion (*Fortier and Pritchard, 2025*). We first constructed large multi-gene trees encompassing all Class I, Class IIA, and Class IIB genes. These provided a backbone for us to investigate subtrees in more depth, adding more sequences and more species to construct 'focused trees' for each gene group. These, in combination with the literature, allowed us to create hypotheses about the evolution of the Class I α-block (*Figure 6*) and Class II MHC-DRB region (*Figure 7*).

## Gene conversion

We inferred gene conversion fragments using *GENECONV* version 1.81 a (*Sawyer, 1999*) on each focused alignment. It is generally advisable to use only synonymous sites when running the program on a protein-coding alignment, since silent sites within the same codon position are likely to be correlated. However, the extreme polymorphism in these MHC genes meant there were too few silent sites to use in the analysis. Thus, we considered all sites but caution that this could slightly overestimate the lengths of our inferred conversion tracts. For each alignment, we ran *GENECONV* with options `ListPairs`, `Allouter`, `Numsims = 10000`, and `Startseed = 310`. We collected all inferred 'Global Inner' (GI) fragments with sim_pval < 0.05 (this is pre-corrected for multiple comparisons by

the program). GI fragments indicate a stretch of similar sequence shared by two otherwise-dissimilar sequences in the alignment. This suggests that a gene conversion event occurred between the ancestors of the two sequences.

Many of the thousands of GI hits were redundant, involving very closely related alleles, slightly different fragment bounds, or even a wide range of species all implicating the same gene. We manually grouped and summarized these hits for *Figure 3—source data 1*, *Figure 4—source data 1*. The 'start' and 'end' columns indicate the smallest start and largest end position (along the alignment) for the group of redundant hits, and the sequences involved are summarized as specifically as possible.

## Acknowledgements

We acknowledge support from NIH grants R01 HG011432 and R01 HG008140. This material is based upon work supported by the National Science Foundation Graduate Research Fellowship under Grant No. DGE-1656518. We appreciate helpful comments from Jeffrey Spence, the Pritchard lab, and the reviewers of the previous version of our companion paper, which jumpstarted this project.

## Additional information

### Funding

| Funder | Grant reference number | Author |
| --- | --- | --- |
| National Institutes of Health | R01 HG011432 | Alyssa Lyn Fortier Jonathan K Pritchard |
| National Institutes of Health | R01 HG008140 | Alyssa Lyn Fortier Jonathan K Pritchard |
| National Science Foundation | DGE-1656518 | Alyssa Lyn Fortier |

The funders had no role in study design, data collection and interpretation, or the decision to submit the work for publication.

### Author contributions

Alyssa Lyn Fortier, Data curation, Software, Formal analysis, Investigation, Visualization, Methodology, Writing – original draft; Jonathan K Pritchard, Conceptualization, Resources, Supervision, Funding acquisition, Investigation, Project administration, Writing - review and editing

### Author ORCIDs

Alyssa Lyn Fortier ⓘ https://orcid.org/0000-0001-5964-2540
Jonathan K Pritchard ⓘ https://orcid.org/0000-0002-8828-5236

Joint Public Review: https://doi.org/10.7554/eLife.103545.3.sa1
Author response https://doi.org/10.7554/eLife.103545.3.sa2

## Additional files

### Supplementary files

Supplementary file 1. List of alleles used in the Class I multi-gene trees.
Supplementary file 2. List of alleles used in the Class IIA multi-gene trees.
Supplementary file 3. List of alleles used in the Class IIB multi-gene trees.
Supplementary file 4. List of alleles used in the Class I α-block-focused trees.
Supplementary file 5. Class I multi-gene nucleotide sequence alignment for exon 2, in fasta format.
Supplementary file 6. Class I multi-gene nucleotide sequence alignment for exon 3, in fasta format.
Supplementary file 7. Class I multi-gene nucleotide sequence alignment for exon 4, in fasta format.
Supplementary file 8. Class IIA multi-gene nucleotide sequence alignment for exon 2, in fasta

format.

Supplementary file 9. Class IIA multi-gene nucleotide sequence alignment for exon 3, in fasta format.

Supplementary file 10. Class IIB multi-gene nucleotide sequence alignment for exon 2, in fasta format.

Supplementary file 11. Class IIB multi-gene nucleotide sequence alignment for exon 3, in fasta format.

Supplementary file 12. Class I α-block nucleotide sequence alignment for exon 2, in fasta format.

Supplementary file 13. Class I α-block nucleotide sequence alignment for exon 3, in fasta format.

Supplementary file 14. Class I α-block nucleotide sequence alignment for exon 4, in fasta format.

MDAR checklist

Source code 1. This zip file contains all xml files we used to run *BEAST2* with *SubstBMA* on each gene group/exon alignment.

## Data availability

The current manuscript is a computational study, and all data used is publicly available. Lists of alleles used in this study, sequence alignments, and xml files for running *BEAST2* are available as supplementary files. Additional citations for Figures 1 and 2 are also available as supplementary files. Sets of posterior trees from *BEAST2* for each gene group and gene region are available at https://doi.org/10.5061/dryad.37pvmcvz7.

The following dataset was generated:

| Author(s) | Year | Dataset title | Dataset URL | Database and Identifier |
|---|---|---|---|---|
| Fortier AL, Pritchard JK | 2025 | The primate Major Histocompatibility Complex: Sets of posterior trees from BEAST2 for the whole-class multi-gene alignments | https://doi.org/10.5061/dryad.37pvmcvz7 | Dryad Digital Repository, 10.5061/dryad.37pvmcvz7 |

The following previously published dataset was used:

| Author(s) | Year | Dataset title | Dataset URL | Database and Identifier |
|---|---|---|---|---|
| Fortier AL, Pritchard JK | 2025 | The primate Major Histocompatibility Complex: Sets of posterior trees from BEAST2 for each gene group and region | https://doi.org/10.5061/dryad.zcrjdfnrz | Dryad Digital Repository, 10.5061/dryad.zcrjdfnrz |

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

# Appendix 1

## General roles of MHC and MHC-like genes

The extremely polymorphic 'classical' MHC genes are mainly involved in adaptive immunity via their interaction with T cells. In contrast, the 'non-classical' genes display little polymorphism and have niche functions, sometimes with limited tissue expression. Additionally, certain classical and non-classical genes are involved in innate immunity, serving as ligands for receptors on Natural Killer (NK) cells and some T cells (*Heijmans et al., 2020*; *Vollmers et al., 2021*). This appendix provides an overview of these two main functions; the roles of specific genes are discussed in Appendix 3.

## Peptide presentation to T cells

Each T cell has the capacity to recognize and respond to one of a countless number of possible foreign invaders. The 'classical' MHC gene products are responsible for peptide presentation—displaying small fragments of protein (called peptides) at the surface of the cell for recognition by these T cells. This process allows T cells to constantly monitor the body for non-self peptides, which could indicate a tumor or infection. T cell recognition of a foreign peptide then triggers destruction of the affected cell (*Dendrou et al., 2018*; *Neefjes et al., 2011*). There are two groups of classical MHC molecules, Class I and Class II, which have slightly different roles.

Classical Class I MHC complexes are expressed on almost all nucleated cells and present cytosolic protein fragments 8-9 amino acids in length to the surface of the cell. The presented peptide is recognized by the T cell receptor (TCR) of a cytolytic CD8+ T cell. In contrast, Classical Class II complexes present mostly endocytosed extracellular protein fragments 12-25 amino acids in length, which are recognized by the TCRs of helper CD4+ T cells. Unlike the widely expressed Class I proteins, Class II proteins are expressed only on specific antigen-presenting cell types, such as B cells, dendritic cells, and macrophages (*Gfeller and Bassani-Sternberg, 2018*; *Heijmans et al., 2020*; *Neefjes et al., 2011*). Both classes of molecules are retained within the cell and are eventually degraded unless they bind a peptide, at which point they are stable and transported to the cell surface (*Appendix 1—figure 1*; *Neefjes et al., 2011*). The only exception is non-classical gene MHC-F, which can be stable as a peptide-unbound 'open conformer' (see Appendix 3) (*Otting et al., 2020*).

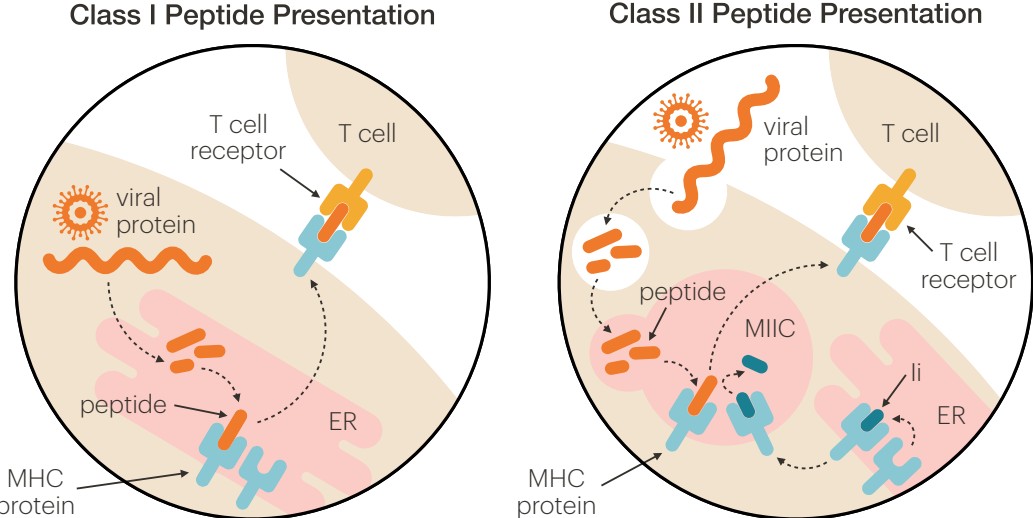

**Appendix 1—figure 1.** Class I and Class II differ in their mechanism of peptide presentation. For Class I (left), endogenous proteins are broken down by a proteasome and imported into the endoplasmic reticulum (ER), where they are loaded onto an awaiting MHC molecule. The peptide-bound MHC molecule is then transported to the cell surface via the Golgi, where it can interact with the T cell receptor (TCR) of a CD8+ T cell (*Neefjes et al., 2011*). The Class II pathway (right) is more complex. Here, exogenous proteins are endocytosed and broken down on their way to the MHC class II compartment (MIIC). MHC molecules originate in the ER, where they are loaded

*Appendix 1—figure 1 continued on next page*

*Appendix 1—figure 1 continued*

with the invariant chain (Ii). The Ii-bound MHC molecule is transported to the MIIC, where the Ii is trimmed and swapped out for an endocytosed peptide. This swap is catalyzed by the non-classical MHC molecule MHC-DM (*Neefjes et al., 2011*; *Dijkstra and Yamaguchi, 2019*; *Schulze and Wucherpfennig, 2012*). The peptide-bound MHC molecule can then be transported to the cell surface to interact with the T cell receptor (TCR) of a CD4+ T cell (*Neefjes et al., 2011*). Throughout the figure, solid lines indicate labels while dashed lines indicate movement or processes.

The genes encoding these molecules have a standard structure. Class I MHC genes consist of 7–8 exons, of which exons 2 and 3 encode the peptide-binding region (PBR) in the resulting protein. The MHC protein forms a nonconvalently associated heterodimer with another protein called $\beta_2$-microglobulin to form the final Class I molecule (*Adams and Parham, 2001a*; *Hughes and Nei, 1988*). Class II molecules also consist of two noncovalently associated chains, α and β, but do not contain a $\beta_2$-microglobulin (*Gu and Nei, 1999*; *Yeager and Hughes, 1999*). The genes encoding the α chains consist of 4–5 exons and those for the β chains consist of 5–6 exons; exon 2 of each chain encodes half of the PBR of the final dimer (*Klein et al., 1998*; *Salomonsen et al., 2003*). Variation among classical MHC alleles is concentrated in the PBR, which also exhibits high $\frac{dN}{dS}$, a population-genetic indicator of positive selection (*Klein et al., 1998*; *Hughes and Hughes, 1995*). Variants in the PBR create allele-specific binding affinities, and thus differential responsiveness to self- and non-self peptides (*Dendrou et al., 2018*; *Pierini and Lenz, 2018*). This results in thousands of GWAS associations between various MHC alleles and autoimmune and infectious diseases (*Kennedy et al., 2017*; *Dendrou et al., 2018*; *Buniello et al., 2019*).

## Ligands for Natural Killer cells

Peptide presentation to T cells is not the only way that MHC molecules influence the immune response. Some Class I alleles have epitopes that serve as ligands for the killer cell immunoglobulin-like receptors (KIR) or C-type lectin receptors (CD94/NKG2) expressed on Natural Killer (NK) cells or the inhibitory leucocyte immunoglobulin-like receptors (LILR) expressed on monocytes and some T-, B-, and NK cells (*Heijmans et al., 2020*; *Biassoni and Malnati, 2018*; *Guethlein et al., 2015*). Unlike T cell receptors, which are part of the 'adaptive' immune response due to their recognition of specific foreign peptides, these other types of receptors are part of the innate immune response, able to detect the general loss of self that occurs under cellular stress (*Biassoni and Malnati, 2018*; *Parham and Moffett, 2013*). KIR, CD94/NKG2, and LILR come in two flavors: activating and inhibitory. Activating receptors trigger an immune response when they recognize an epitope, while inhibitory receptors prevent an immune response from taking place. The receptors that recognize MHC are generally inhibitory—if they detect the right amount of MHC molecules on the surface of the cell, then all is well and no immune response is needed (*Biassoni and Malnati, 2018*).

MHC and KIRs have co-evolved across the primates, and their interactions have also shaped polymorphism in the MHC region. Each receptor can only recognize a particular epitope, and not all MHC alleles have one of the recognizable epitopes. While many MHC-receptor interactions are conserved, such as that of non-classical MHC-E and CD94–NKG2A, others have been free to diversify (*Parham and Moffett, 2013*; *Guethlein et al., 2015*). Over evolutionary time, particular lineages of KIRs have expanded or been lost as the allelic lineages they recognize have also expanded or been lost. For example, the classical MHC-C gene originated around the time of orangutan divergence. This corresponds with the expansion of the lineage III KIRs that are able to recognize MHC-C alleles—while old-world monkeys have just one (possibly non-functional) lineage III KIR, the great apes have up to 9 of these receptors. The expansion of lineage III also corresponds with the contraction of lineage II KIRs (which can recognize some MHC-A and -B alleles) in the great apes, supporting the fact that MHC-C has evolved to be the dominant KIR receptor in this group (*Parham and Moffett, 2013*; *Biassoni and Malnati, 2018*; *Hilton and Parham, 2017*). Additionally, MHC-KIR interaction is involved in the maternal-fetal interface during pregnancy, and the expansion of lineage III with MHC-C also correlates with an increasing amount of trophoblast invasion within the great ape species (*Wroblewski et al., 2019*; *Moffett-King, 2002*). The NK cell interactions of particular genes are discussed in Appendix 3.

## Appendix 2

## MHC nomenclature

The large number of genes in the MHC, some with thousands of alleles, necessitates a consistent naming scheme (*Appendix 2—figure 1*). Known alleles are given names such as 'Aole-DQB1*23:01', and names are maintained and updated by the WHO Nomenclature Committee for Factors of the HLA System (*Robinson et al., 2024*; *Marsh et al., 2010*). First, the species of origin is indicated by a four-letter prefix consisting of the first two letters of the genus name and the first two letters of the species name, for example 'Chsa-' for Chlorocebus sabaeus, the green monkey. There are some exceptions, usually because these MHC systems were first investigated before the naming scheme was put into place. These include 'HLA-' for human, 'H2-' for mouse, 'RT1-' for rat, and 'SLA-' for swine, among others (*de Groot et al., 2020*; *de Groot et al., 2012*).

After the hyphen is the locus designation. Some species, such as humans, have a relatively simple landscape of MHC genes, making it easy to identify sequences that belong to a particular gene. However, other species have recent gene expansions and considerable region conformation diversity, making it difficult to assign alleles to genes. In some cases, these are given generic locus designations; for example, rhesus macaques have at least 19 paralogous B loci, but most are given the ambiguous name 'Mamu-B' (with the exception of a few well-characterized genes such as Mamu-B17). In other cases, unassigned sequences are given a working designation indicated by a 'W', such as 'Popy-DRB*W113:01'. In this example, the allele definitely belongs to a DRB paralog, but it is unclear which one. Some locus names are given a 'Ps' suffix to indicate they are pseudogenes, such as 'Caja-G5Ps'. However, not all pseudogenes are labeled this way, so one should not assume the lack of a 'Ps' suffix means a gene is functional (*Appendix 2—figure 1B*; *de Groot et al., 2020*; *de Groot et al., 2012*).

**A.** Human (HLA) alleles are named hierarchically with standardized nomenclature.

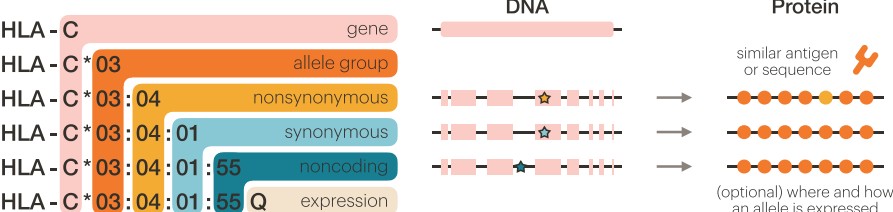

**B.** Non-human alleles follow the same general pattern, but with some peculiarities.

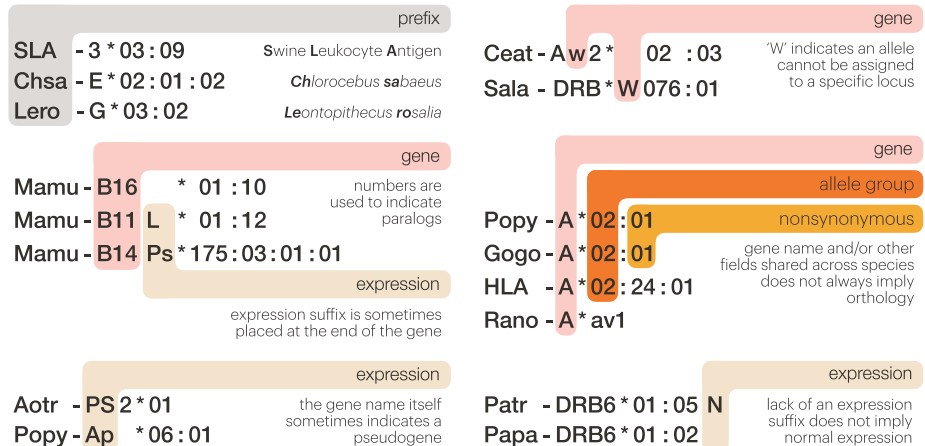

**Appendix 2—figure 1.** MHC allele nomenclature. (**A**) Human HLA alleles are named in a standard fashion, with the gene name followed by four colon-separated fields. The first field indicates a broad-scale allele group which sometimes corresponds to a serological antigen. The second field denotes a specific HLA protein. The third field indicates synonymous changes to the nucleotide sequence in the coding region, while the fourth field is used to distinguish alleles with differences in the noncoding regions. If an allele's expression has been characterized, an

*Appendix 2—figure 1 continued*

informative suffix is sometimes added (**Robinson et al., 2024**; **Marsh et al., 2010**). (**B**) Researchers have applied the same format to non-human alleles, with some key differences. Instead of 'HLA', a prefix which concatenates the first two letters of the genus name with the first two letters of the species name is used, except in certain cases where the species' MHC system was named long ago. Paralogs can be distinguished using numbers, but sequences unassigned to a particular locus or paralog might incorporate a 'W' in the gene name. Use of expression tags varies, with some being added to the end of the gene name instead of the end of the entire allele name. Pseudogenes can be denoted with a 'p' or 'Ps' in the gene name suffixes, gene names themselves, expression suffixes, or not at all. For both human and non-human alleles, the lack of an expression suffix does not imply normal expression (**de Groot et al., 2020**). SLA: Swine Leukocyte Antigen; Chsa: *Chlorocebus sabaeus*— green monkey; Lero: *Leontopithecus rosalia*—golden lion tamarin; Mamu: *Macaca mulatta*—rhesus macaque; Aotr: *Aotus trivirgatus*—three-striped night monkey; Popy: *Pongo pygmaeus*—Bornean orangutan; Ceat: *Cercocebus atys*—sooty mangabey; Sala: *Saguinus labiatus*—white-lipped tamarin; Gogo: *Gorilla gorilla*—Western gorilla; Rano: *Rattus norvegicus*—brown rat; Patr: *Pan troglodytes*—chimpanzee; Papa: *Pan paniscus*—bonobo.

After the species and locus name, each MHC allele is designated by up to four fields separated by colons. The first field designates the type or family. Types often, but not always, correspond to the broad serological reactivity of the allele, as many were named before full sequences were known. To facilitate comparison across closely related species, researchers generally try to give related MHC alleles the same first-field designation, for example, Gogo-A*02 and HLA-A*02. However, certain genes do not follow this general rule. For example, MHC-DPB1 has undergone considerable gene conversion, resulting in no distinct types; thus, a shared first-field designation between species is meaningless for this gene (**de Groot et al., 2012**; **de Groot et al., 2020**). The second field designates the allele subtype, or unique amino acid sequence. For example, 'Patr-A*08:01' and 'Patr-A*08:02' are part of the same allelic family, but have some nonsynonymous differences. Synonymous changes are specified by the third field. For example, 'Paan-DPB1*03:01:01' and 'Paan-DPB1*03:01:02' have silent substitutions which ultimately result in the same protein. Lastly, the fourth field is used to describe changes to the noncoding regions—that is, the 5' and 3' UTRs and the introns. Of course, this requires that these regions have been sequenced, so not all alleles will have a fourth field. Finally, alleles can also be followed by an optional suffix to describe expression changes, most commonly 'N' for a null/nonexpressed allele or 'L' for a lowly-expressed allele (**Hurley, 2021**; **Douillard et al., 2021**).

In general, caution must be taken in interpreting allele names. First, because not all alleles are resolved at three- and four-field resolution, the names are not all strictly hierarchical; alleles which have all four fields cannot always simply be truncated to obtain the two-field version. Second, because human alleles were named in order of discovery, alleles with very different one- or two-field designations could ultimately have the same nucleotide or amino acid sequence in the peptide-binding groove. When discussing functional consequences, it is relevant to group alleles by their nucleotide or amino acid sequence in the peptide-binding-site-encoding exons (designated G- and P-groups, respectively) and not necessarily by their one- or two-field name (**Hurley, 2021**; **Douillard et al., 2021**). Additionally, the suffixes can be misleading because not every allele has had its expression level characterized—the absence of an 'L' does not mean that an allele has normal expression (**Hurley, 2021**). Despite these small issues, the naming system is generally intuitive and very useful for understanding alleles at a glance. In this work, alleles obtained from the IPD-MHC and IPD-IMGT/HLA databases are named this way, but sequences obtained from RefSeq are labeled by accession number or location along a chromosome.

## Appendix 3

### The MHC genes

### The Class I subfamily

Humans have a (relatively) simple landscape of Class I genes (see *Figure 1*). HLA-A, -B, and -C are the classical genes and HLA-E, -F, and -G are the non-classical genes. Additionally, HLA-C, -E, -F, -G, and some alleles of HLA-A and -B serve as ligands for NK cell receptors. There are also a large number of pseudogenes: HLA-H, -J, -K, -L, -Y, and -OLI are full-length MHC pseudogenes (7–8 exons) while HLA-N, -P, -S, -T, -U, -V, -W, -X, and -Z are partial-length pseudogenes (*Robinson et al., 2024*; *Barker et al., 2023*). Few genes (and even fewer whole haplotypes) have been sequenced for non-human primates, making it difficult to understand orthology between them and the human genes. Here, we review past work on each specific gene and add our own contributions.

The MHC-A-related genes

MHC-A in the apes

Across the apes and OWM, MHC-A is a highly polymorphic classical Class I gene. In addition to classical peptide presentation, in humans ~40% of alleles of this gene also serve as ligands for KIR (0–10% in the other great apes; *Parham and Moffett, 2013*; *Wroblewski et al., 2019*). Human HLA-A has true orthologs in the African apes—chimpanzee/bonobo (Patr/Papa-A) and gorilla (Gogo/Gobe-A). However, the orangutan MHC-A gene (Popy/Poab-A) is not orthologous to this group and is in fact part of a second MHC-A-related orthogroup containing orangutan Popy/Poab-A, gorilla-specific Gogo/Gobe-OKO, and chimpanzee-specific Patr-AL (see *Figure 1*; *Adams and Parham, 2001a*; *Hans et al., 2017*; *Heijmans et al., 2020*). Although they are named differently, these purported gorilla-specific and chimpanzee-specific genes are actually orthologous to each other and the orangutan MHC-A gene. Humans lack an equivalent to this gene.

This makes two groups of MHC-A genes to worry about, HLA-A/Patr-A/Papa-A/Gogo-A/Gobe-A and Patr-AL/Gogo-OKO/Gobe-OKO/Popy-A/Poab-A. The chimpanzee has both genes, called Patr-A and Patr-AL (A-Like). In our exon 4 tree (*Figure 3—figure supplement 3C*), Patr-A groups with HLA-A alleles of the 'A3' lineage, while Patr-AL groups with orangutan Poab/Popy MHC-A, as expected. Patr-A is fixed while Patr-AL is present on only about 50% of chimpanzee haplotypes and is absent in the chimpanzee's sister species, the bonobo (Papa; *Gleimer et al., 2011*; *Maibach et al., 2017*). Patr-AL has a similar peptide-binding repertoire to alleles of the group HLA-A*02, potentially allowing Patr-AL+ individuals to bind more peptides than is possible with Patr-A alone (which lacks this allele family; *Gleimer et al., 2011*; *Goyos et al., 2015*). However, the Patr-AL gene has limited polymorphism, low cell surface expression, and does not serve as a ligand for KIR, suggesting a niche non-classical role (*Gleimer et al., 2011*; *Goyos et al., 2015*; *Wroblewski et al., 2019*). Additionally, the fact that Patr-AL has not fixed over a sufficient period of time suggests balancing selection is acting, rather enigmatically, to retain both the Patr-AL⁺ and Patr-AL⁻ haplotypes (*Gleimer et al., 2011*).

Gorillas also have both genes: Gogo/Gobe-A is part of the first group, while Gogo/Gobe-OKO is part of the second (*Figure 1*; *Heijmans et al., 2020*; *Gleimer et al., 2011*). Haplotypes containing the Gogo/Gobe-OKO locus lack the Gogo/Gobe-A locus, with Gogo/Gobe-OKO appearing in 44% of gorillas. The fact that neither gene is fixed suggests that Gogo/Gobe-OKO is likely sufficient as an MHC-A-like classical molecule and that balancing selection is acting to retain both (*Hans et al., 2017*; *Wroblewski et al., 2019*; *Watkins et al., 1991*). Few Gogo/Gobe-A alleles and no Gogo/Gobe-OKO alleles serve as ligands for KIR, suggesting that both molecules are mainly involved in peptide presentation (*Wroblewski et al., 2019*; *Hans et al., 2017*). Gogo/Gobe-OKO also has a complex recombinant structure (*Hans et al., 2017*; *Gleimer et al., 2011*; *Adams and Parham, 2001a*). In our exon 2 tree, Gogo/Gobe-OKO groups with MHC-H, while in our exon 3 tree, only some Gogo/Gobe-OKO alleles group with MHC-H, while others group with human HLA-A (*Figure 3—figure supplement 3A-B*). In exon 4, these sequences group outside of a clade containing 'A2' lineage alleles and orangutan MHC-A sequences (*Figure 3—figure supplement 3C*).

The gibbon MHC is, in general, poorly studied, and although gibbons appear to have at least one functionally equivalent MHC-A gene, it has been unclear whether it is part of either of the great

ape A-related orthogroups (*Adams and Parham, 2001a*; *Abi-Rached et al., 2010*). Our *BLAST* search of three gibbon reference genomes revealed one MHC-A gene in the siamang and one in the Northern white-cheeked gibbon, but none in the pileated gibbon (*Figure 3—figure supplement 2*); however, this could be due to an incomplete assembly in this species. Our exon 4 tree (*Figure 5— figure supplement 1C*) groups gibbon MHC-A sequences with an OWM MHC-A-related fragment pseudogene (MHC-Apseudo) rather than with any particular ape MHC-A-related gene. This suggests that: (1) OWMs contain a remnant of the ancestral gene that also generated the ape MHC-A genes, and (2) the gibbon MHC-A gene 'branched off' prior to the expansion of the various known great ape MHC-A-related genes.

## MHC-A in the OWM

In OWM, the MHC-A gene has expanded massively, with up to eight named MHC-A genes in macaque and even more which have not yet been given separate locus names (*Heijmans et al., 2020*; *de Groot et al., 2020*). None of the genes are fixed, at least in macaques, and they are part of repeating blocks (which will be discussed below). Of all these genes, the MHC-A1 genes are generally highly expressed ('major'), while the others are generally lowly expressed ('minor'). Further, not all appear to be classical; the MHC-A2*05 gene (named so because it was previously thought to be an allele) serves a specialized function, possibly to bind specific simian immunodeficiency virus (SIV) epitopes (*de Groot et al., 2022*; *Heijmans et al., 2020*; *de Groot et al., 2017a*). It is often unclear which OWM MHC-A sequences belong to which loci, which sequences are part of shared allelic lineages, and whether all MHC-A sequences are even related (*Adams and Parham, 2001a*). For example, the MHC-A8 gene has been found in just one species of macaque (crab-eating, Mafa) with a frequency of 15%. However, despite being mapped to the MHC-A locus, it has low similarity to the other OWM MHC-A genes (and equally low similarity to the other Class I genes) (*Shiina et al., 2015*; *de Groot et al., 2017a*). Our trees (*Figure 3—figure supplements 2 and 3*, *Figure 5— figure supplement 1*) do not group MHC-A8 with the other OWM or ape MHC-A genes, suggesting that it is not actually MHC-A-related. Instead, it appears on a long branch among the 'backbone' sequences in each tree, suggesting it is an entirely different (and deeply diverged) type of gene or pseudogene. Furthermore, we discovered a sequence in the green monkey (Chsa) that groups with macaque Mafa-A8 in every tree, suggesting that the green monkey—whose lineage split from macaque 15 million years ago—has an MHC-A8 ortholog (*Kuderna et al., 2023*). Thus, we show that MHC-A8 is a distinct type of Class I gene and is much older than previously surmised.

Other OWM species have received less research attention than the macaque, but they similarly appear to have a variable number of major and minor MHC-A genes (*Heimbruch et al., 2015*; *van der Wiel et al., 2018*). Some genes seem to be orthologous between different OWM species, but shockingly few haplotypes are shared (*de Groot et al., 2022*; *de Groot et al., 2020*). Our *BLAST* search of several OWM reference genomes revealed 3–4 MHC-A genes on the reference haplotypes for different macaque species, four in the gelada, two in the golden snub-nosed monkey, just one in the olive baboon, and none (only MHC-AG) in the mantled guereza (*Figure 3—figure supplement 2*). Across the macaques, the arrangement of these genes on the chromosome is fairly consistent, but they are less regularly arranged in more distantly-related OWM species. The sequencing of more OWM haplotypes will help us tease apart the complicated relationships between these expanded gene subfamilies.

The complexity of the MHC-A genes illustrates a difference in strategy among different primates— whereas humans have a single copy of each classical gene with substantial allelic variation, the OWMs have a large number of unfixed gene paralogs with limited polymorphism, distributed across a wide variety of haplotypes (*de Groot et al., 2015*; *de Groot et al., 2022*).

## MHC-AG

Not only do the OWM have a massively expanded set of paralogous MHC-A loci, but they also have an additional family of MHC-A-related genes called MHC-AG (AG1-AG6). Although many paralogous MHC-AG loci have been identified in different species, they do not have any locus-defining substitutions, suggesting recent origins or frequent genetic exchange (*Adams and Parham, 2001a*). Each of them is also unfixed, at least in macaques (*Karl et al., 2023*). Although the MHC-AG genes are evolutionarily related to the MHC-A genes, they have taken on a non-classical role. In fact,

they have converged in function, expression pattern, and alternative splicing options to the MHC-G gene, corresponding with the pseudogenization of MHC-G in the OWM lineage (*Bondarenko et al., 2007*; *Adams and Parham, 2001a*; *Nicholas et al., 2022*; *Karl et al., 2023*). These genes' function will be discussed in more detail in the MHC-G section.

By including additional species in our analysis, we discovered the presence of MHC-AG in other OWM. Previously, this gene group had only been confirmed in macaque, baboon, and green monkey, all part of the OWM subgroup called the *Cercopithecinae*. The other OWM subgroup, the *Colobinae*, had been effectively ignored despite composing nearly half of all OWM species. We found sequences that grouped unambiguously with MHC-AG in all three *Colobinae* species we included in our analysis (*Figure 3—figure supplements 2 and 3*). This dates the origin of MHC-AG to prior to the radiation of all OWM, rather than just prior to the expansion of the *Cercopithecinae* (*Bondarenko et al., 2009*). Since MHC-AG is exclusively found in OWM, it must have duplicated from MHC-A in the ancestor of the OWM between 19 and 30 million years ago (*Adams and Parham, 2001a*; *Kuderna et al., 2023*).

Our *BLAST* search of the reference genomes (*Figure 1—figure supplement 2*) revealed only one MHC-AG copy on the *Colobinae* haplotypes, but multiple in the macaques. Therefore, the MHC-AG locus may have begun to expand after the split of the *Colobinae* and *Cercopithecinae*. Sequencing more haplotypes from the OWM will be necessary to determine if this is the case. This will also help resolve whether the MHC-AG locus began to duplicate before the diversification of the macaques or if most of the paralogs are species-specific.

## MHC-A and -AG repeat blocks

Macaque haplotypes have multiple copies of both MHC-A and -AG. They (along with pseudogenes MHC-G, -W, -V, and -K) are arranged on the chromosome in repeating patterns, suggesting that the current arrangement arose via block duplications (*Karl et al., 2023*; *Kulski et al., 2004*). However, it has been unclear whether these duplications occurred within the macaque lineage or are older. As shown in *Figure 1—figure supplement 2*, the same repeating pattern of genes is found in the reference genomes of the closely related Formosan rock macaque and crab-eating macaque, but deviates slightly in the Tibetan macaque. Next most closely related are the baboon and gelada; the baboon's very short MHC-A region is arranged similarly to the macaque's, but the gelada haplotype has a very different arrangement. The snub-nosed monkey, which belongs to the *Colobinae*, has blocks that nearly, but not exactly, match the macaque's. However, there is only one copy of each duplication unit (in other words, non-duplicated). This suggests that this initial arrangement of genes—one copy each of MHC-AG, -G, -Apseudo, -K, and -A—was established prior to the radiation of all OWM. Independent deletions appear to have occurred in the baboon, gelada, and mantled guereza, while repeated block duplications likely occurred in the ancestor of the macaques and arrangements are shared by at least some species.

To investigate this further, we created Class I α-block-focused trees (*Figure 5—figure supplement 1*) and observed that OWM MHC-W sequences assort into clearly defined clades that are consistent with the region evolving by block duplications (*Figure 5D*, text, and *Figure 6*). Additionally, the crab-eating macaque haplotype contains a fragment pseudogene that *Karl et al., 2023* determined was MHC-A-related; we also found a similar fragment in our *BLAST* analysis of the Tibetan macaque and Formosan rock macaque genomes. We show that this sequence (MHC-A pseudo) groups clearly with the ape MHC-A genes instead of the OWM MHC-A and -AG genes, which form their own monophyletic clade in all trees (*Figure 5—figure supplement 1C*). Thus, the ape/OWM ancestor must have had two distinct MHC-A genes: one gave rise to all the ape MHC-A-related genes and became a pseudogene in the OWM lineage, and one was the ancestor of all the OWM MHC-A and -AG genes and was deleted from the ape lineage.

## MHC-A in the NWM

MHC-A is not present in the NWM; however, one study places the gene's origin prior to the divergence of the NWM from the *Catarrhini* and proposes that it was subsequently lost in the NWM ancestor (*Adams and Parham, 2001a*; *Heijmans et al., 2020*; *Sawai et al., 2004*; *Piontkivska and Nei, 2003*). Just as plausibly, it could have originated after the divergence of the NWM. Our trees show that of the Class I α-block genes, only MHC-F is truly orthologous between apes/OWM and

NWM. Further, no traces of the other α-block genes, pseudogenes, or pseudogene fragments have been found in any NWM. This suggests that MHC-F was established before the split of apes/OWM from NWM, but that the rest of the α-block genes were generated separately in each group from a common non-MHC-F precursor (see our hypothesis in *Figure 6*).

## MHC-H

MHC-H is present in humans (HLA-H) chimpanzees (Patr-H), bonobos (Papa-H), gorillas (Gogo/Gobe-H), and orangutans (Poab/Popy-H), but has different roles in each species (see *Figure 1*; *Adams and Parham, 2001a*; *Sawai et al., 2004*; *Grimsley et al., 1998*; *Paganini et al., 2019*). In humans, HLA-H has functional and non-functional alleles, and its functional alleles appear to interact with the innate immune system in a similar fashion to HLA-E, -F, and -G (*Hubert et al., 2022*; *Kulski et al., 2020*; *Carlini et al., 2016*). Curiously, despite this seemingly important role, most HLA-H alleles are non-functional, and the gene has been entirely deleted from 10–20% of humans (*Geraghty et al., 1992*; *Paganini et al., 2019*; *Alexandrov et al., 2023*). In chimpanzees, MHC-H is sometimes expressed, and in gorillas, it has been deemed functional (*Wilming et al., 2013*; *Adams and Parham, 2001b*). Unlike other pseudogenes, MHC-H is reasonably diverse and substitutions are not evenly distributed across the gene. Because it is located near classical MHC-A, this could be due to hitchhiking; however, since the gene is apparently functional in other species, this pattern could also be evidence of past selection (*Grimsley et al., 1998*; *Hughes and Hughes, 1995*; *Paganini et al., 2019*).

MHC-H has not been detected in gibbons, OWM, or NWM, suggesting the gene duplicated from MHC-A in the common ancestor of the great apes. However, since the gibbon genome exhibits a large deletion spanning the location of HLA-H, it is possible that MHC-H arose prior to the divergence of the gibbons and was lost via the deletion (*Adams and Parham, 2001a*; *Abi-Rached et al., 2010*; *Neehus et al., 2016*). MHC-H is closely related to the Patr-AL/Gogo-OKO/Gobe-OKO/Popy-A/Poab-A orthogroup and more distantly related to the HLA-A/Patr-A/Papa-A/Gogo-A/Gobe-A orthogroup. Previous work has suggested that MHC-H/Patr-AL/Gogo-OKO/Gobe-OKO/Popy-A/Poab-A duplicated from HLA-A/Patr-A/Papa-A/Gogo-A/Gobe-A 16-30mya, and MHC-H subsequently duplicated from Patr-AL/Gogo-OKO/Gobe-OKO/Popy-A/Poab-A 11-23mya (*Gleimer et al., 2011*; *Piontkivska and Nei, 2003*; *Geller et al., 2002*). Instead, our trees support a first duplication of MHC-H from the combined MHC-A/AL/OKO, followed by the split of combined MHC-A/AL/OKO into the two MHC-A genes: Patr-AL/Gogo-OKO/Gobe-OKO/Popy-A/Poab-A and HLA-A/Patr-A/Papa-A/Gogo-A/Gobe-A. Typically, we trust exon 4 trees to be less influenced by convergent evolution, but among the MHC-A-related genes, there seems to have been a lot of recombination affecting exon 4 (*Gleimer et al., 2011*). Indeed, our exon 4 trees (*Figure 3—figure supplements 2C and 3C*, *Figure 5—figure supplement 1C*) show MHC-H grouping with 'A3' lineage alleles, as previously observed, so this exon is less informative. Based on past work, exon 3 shows no evidence of recombination affecting MHC-H and its relatives, so we focus on the exon 3 trees (*Gleimer et al., 2011*). In exon 3 (*Figure 3—figure supplements 2B and 3B*, *Figure 5—figure supplement 1B*), the clade of MHC-H sequences groups outside of the clade containing sequences from both of the MHC-A orthogroups. This supports that the two MHC-A orthogroups are more closely related, and together they are equally related to MHC-H.

## MHC-Y and -OLI

HLA-Y and -OLI are two physically linked pseudogenes present on 29% of known human HLA haplotypes (*Alexandrov et al., 2023*; *Zhou et al., 2024*; *Liao et al., 2023*). They are always found together and are linked to the HLA-A alleles HLA-A*29:01, *30:01, *31:01:02, *33:01, *33:03, *34:01, *68:01:02, *68:02, and *02:05, suggesting that they were duplicated as a unit on a haplotype containing an early MHC-A 'A2' lineage ancestor (*Alexandrov et al., 2023*).

The location of HLA-Y and -OLI was recently discovered; they reside in the Class I α-block on a 60 kb indel between HLA-W and -J (*Alexandrov et al., 2023*; *Zhou et al., 2024*; *Liao et al., 2023*). This is inconsistent with the previous hypothesis that HLA-Y is the human ortholog of Patr-AL/Gogo-OKO/Gobe-OKO/Popy-A/Poab-A, which is located in a completely different part of the region (*Gleimer et al., 2011*; *Wroblewski et al., 2019*; *Hans et al., 2017*). Furthermore, there is a gorilla ortholog of HLA-Y—known as Gogo-Y or Gogo-A*05—that is present in 79% of gorillas

(*Wroblewski et al., 2019*). A nearly identical 60 kb indel to human was found in gorilla, confirming the location of Gogo-Y and supporting their orthologous relationship (*Alexandrov et al., 2023*). As a reminder, gorilla haplotypes contain either Gogo/Gobe-OKO (related to the first orthogroup) or Gogo/Gobe-A (related to the second orthogroup), but not both (*Hans et al., 2017*; *Heijmans et al., 2020*). Our trees (*Figure 3—figure supplement 3*) show that Gogo-Y groups with human HLA-Y are clearly distinct from Gogo/Gobe-OKO, further lending support to the fact that MHC-Y is a separate gene that arose on an old haplotype. Nevertheless, MHC-Y is likely recombinant; it is similar to Patr-AL/Gogo-OKO/Gobe-OKO/Popy-A/Poab-A in exon 1, intron 1, exon 2, and intron 2, but it is similar to HLA-A/Patr-A/Papa-A/Gogo-A/Gobe-A across the rest of the gene (*Gleimer et al., 2011*; *Hans et al., 2017*). Based on this structure and our trees, it is unclear which group MHC-Y is most closely related to. However, because MHC-Y and -OLI are 100% linked and MHC-Y is broadly similar to MHC-A-related genes while MHC-OLI is clearly similar to MHC-TL (see later section about MHC-OLI), we hypothesize that they arose via a block duplication of the nearby MHC-AL and -TL in the human/gorilla ancestor. This haplotype was then lost in the chimpanzee lineage. See *Figure 6* for a detailed visual explanation of our hypothesis. Interestingly, the human and gorilla MHC-Y genes have different nonsense mutations, showing that selection has twice acted to inactivate this gene (*Hans et al., 2017*).

## MHC-Ap

A final species-specific MHC-A-related pseudogene is orangutan Poab/Popy-Ap. It has been found in both the Bornean and Sumatran orangutan species (*Wroblewski et al., 2019*), but our *BLAST* search only detected MHC-Ap on the Sumatran haplotype (Poab-Ap) (*Figure 3—figure supplement 2*), so it may be unfixed in the Bornean orangutan. Evolutionary analyses suggest MHC-Ap is not orthologous to the MHC-A genes of the African apes, but it has been unclear whether it is more closely related to Patr-AL/Gogo-OKO/Gobe-OKO/Popy-A/Poab-A or to MHC-H (*Adams and Parham, 2001a*). It is similar to MHC-H in exon 1 through intron 2, but exon 4 and the introns are similar to Patr-AL/Gogo-OKO/Gobe-OKO/Popy-A/Poab-A (*Gleimer et al., 2011*; *Adams and Parham, 2001a*). Our trees (*Figure 3—figure supplements 2 and 3*, *Figure 5—figure supplement 1*) group Poab/Popy-Ap with MHC-H in exon 2 and with Patr-AL/Gogo-OKO/Gobe-OKO/Popy-A/Poab-A in exon 4, as expected. However, in exon 3, Poab/Popy-Ap sequences group with Poab/Popy-A or MHC-H and form an outgroup to the other MHC-A sequences. Coupled with the species specificity of the gene, this suggests that MHC-Ap duplicated from MHC-A or -H within the orangutan lineage. Because it groups differently depending on exon, we cannot determine which gene (MHC-A or -H) it is most closely related to.

Curiously, all alleles of pseudogene Popy-Ap but no alleles of functional Popy/Poab-A encode a KIR epitope. Luckily, although the orangutan MHC-A genes cannot currently serve as KIR ligands, orangutan MHC-B and -C genes retain this ability (*Wroblewski et al., 2019*).

## MHC-U

MHC-U is a partial pseudogene, reported by IPD-IMGT/HLA to contain a single (unknown) exon. To our knowledge, only one study has used sequence from MHC-U in a phylogenetic tree; in this study restricted to human MHC genes, HLA-U did not group clearly with any other human gene (*Alexandrov et al., 2023*). IPD-IMGT/HLA has a few human HLA-U sequences available, and we were able to extract additional chimpanzee Patr-U sequences from our *BLAST* search (see Materials and methods). We did not find any MHC-U sequences in the gorilla, orangutan, or gibbon. These human and chimpanzee MHC-U sequences aligned well with exon 3 of the other genes, demonstrating that MHC-U is an exon-3-only fragment pseudogene. In our exon 3 α-block-focused tree (*Figure 5B*, *Figure 5—figure supplement 1B*), MHC-U sequences formed a monophyletic clade, showing human and chimpanzee MHC-U are orthologs. The MHC-U clade then groups with a clade of human, chimpanzee, and bonobo MHC-A sequences, suggesting that it duplicated from the HLA-A/Patr-A/Papa-A/Gogo-A/Gobe-A ancestral gene. MHC-U is present on all human and chimpanzee haplotypes (except the human-specific haplotype which has a deletion of MHC-H, -K, and -U) (*Gleimer et al., 2011*; *Liao et al., 2023*; *Zhou et al., 2024*). This includes the human haplotype containing HLA-Y and -OLI; because this same haplotype has been detected in gorillas, we expect that MHC-U is relatively old and will be found in that species as more haplotypes are

sequenced (**Alexandrov et al., 2023**). Thus, we show that MHC-U is an exon-3-only MHC-A-related pseudogene and that it likely originated from HLA-A/Patr-A/Papa-A/Gogo-A/Gobe-A in the human/gorilla ancestor.

## The MHC-B-related genes

### MHC-B in the apes

The MHC-B genes are functional and classical in the apes. In humans, HLA-B is the most polymorphic Class I gene, and ~35% of HLA-B alleles are also ligands for KIR (35–65% in other great apes) (**Guethlein et al., 2015**; **Wroblewski et al., 2019**; **Parham and Moffett, 2013**). Human HLA-B has clear orthologs in chimpanzee (Patr-B), bonobo (Papa-B), and gorilla (Gogo-B) (**Adams and Parham, 2001a**; **Sawai et al., 2004**). Twenty-four percent of gorillas also have a second MHC-B gene, Gogo-B*07 (as it was previously thought to be a family of alleles) (**Heijmans et al., 2020**; **Hans et al., 2017**; **Wroblewski et al., 2019**). Haplotypes contain either just Gogo-B or both Gogo-B and Gogo-B*07; our *BLAST* analysis shows that both the gorilla reference genome and specially-sequenced gorilla MHC (**Wilming et al., 2013**) carry the single-MHC-B haplotype. All Gogo-B*07 alleles also serve as ligands for KIR (**Wroblewski et al., 2019**). In our exon 4 MHC-B-focused tree (**Figure 3—figure supplement 4C**), there is a large clade containing human, chimpanzee, and bonobo sequences as well as gorilla MHC-B*01 and *05 sequences, suggesting that a gorilla MHC-B*01/05-like allele was the ancestor of human, chimpanzee, and bonobo MHC-B.

In the orangutan, three separate loci are defined: Poab/Popy-B is fixed and only ~5% of its alleles serve as ligands for KIR; Poab/Popy-B*08 is fixed, lowly expressed, and all its alleles serve as ligands for KIR; and Poab/Popy-B*03 is unfixed, even more lowly expressed, and all its alleles serve as ligands for KIR (**Wroblewski et al., 2019**; **de Groot et al., 2016**). Additionally, we detected a fourth MHC-B gene (and no MHC-C gene) on the Bornean orangutan (Popy) reference haplotype, but it is unknown whether it is functional (**Figure 3—figure supplement 2**). Our exon 4 tree (**Figure 3—figure supplement 4C**) shows Poab/Popy-B falling in two distinct clades, one containing Poab/Popy-B*05/06/07/08/10/11/12/13, gibbon MHC-B, and human HLA-C, and one containing Poab/Popy-B*01/02/03/04 and all other human/chimp/bonobo/gorilla alleles. The Poab/Popy-B*01/02/03/04 clade is clearly divided into Poab/Popy-B*01/02/04 and Poab/Popy-B*03, showing that the Poab/Popy-B*03 gene originated from an ancestral Poab/Popy-B*01/02/04-like gene. The fact that Poab/Popy-B*01/02/03/04 alleles group with the African ape MHC-B genes as well as the genes' arrangement on the chromosome (**Figure 3—figure supplement 2**) suggests that this gene may actually be orthologous to the African ape MHC-B genes.

Similarly, Poab/Popy-B*05/06/07/08/10/11/12/13 is clearly divided into Poab/Popy-B*08 and Poab/Popy-B*05/06/07/10/12/13, showing that the Poab/Popy-B*08 gene originated from an ancestral Poab/Popy-B*05/06/07/10/12/13-like gene. These orangutan clades are outgroups to the African ape MHC-B clades and are more closely related to MHC-C and gibbon MHC-B. Along with their position on the chromosome (**Figure 3—figure supplement 2**), this shows that the other orangutan MHC-B genes (not Poab/Popy-B*01/02/03/04) expanded separately in the orangutan, and that one of these duplicates became known as MHC-C in the African apes.

Gogo-B*07 (a separate gene from the rest of Gogo-B) is also an outgroup to African ape MHC-B, as expected based on its closer similarity to orangutan MHC-B genes. Interestingly, in exons 2 and 3 (**Figure 3—figure supplement 4**), Gogo-B*07 is clearly separated from the other Gogo-B gene, but in exon 4, it groups with Gogo-B*02/03/04/12 and HLA-B*73:01:01:01 (an inter-locus recombinant) (**Gu and Nei, 1999**; **de Groot et al., 2016**). This calls into question whether Gogo-B*02/03/04/12 has been involved in a gene conversion or if they actually belong to the Gogo-B*07 locus.

In the gibbon, past work has revealed three B-like genes on a haplotype (**Abi-Rached et al., 2010**). We analyzed three reference genomes and found two MHC-B genes in the pileated gibbon, one MHC-B and two unknown nearby genes in the siamang, and two MHC-B and one unknown nearby gene in the Northern white-cheeked gibbon (**Figure 3—figure supplement 2**). However, without more sequenced haplotypes, it is impossible to tell if these are fixed. In **Figure 3—figure supplement 4** gibbon sequences group with orangutan and Gogo-B*07 sequences, suggesting they branched off before the origin of the second MHC-B gene in the ancestor of the African apes.

## MHC-B in the OWM

In the OWM, there are many MHC-B paralogs. The MHC-B genes of the OWM are even more complicated than their MHC-A genes, and their orthologous and paralogous relationships are poorly understood (*de Groot et al., 2020*). In macaques, there can be up to 19 paralogous B genes per haplotype, each including 1–6 highly transcribed 'major' genes, 1–10 lowly transcribed 'minor' genes, and several MHC-B-related pseudogenes (*Heijmans et al., 2020*; *de Groot et al., 2022*; *Karl et al., 2023*). While a few macaque MHC-B-related genes have been named—such as pseudogenes MHC-B02, -B10, -B14, -B19, and -B21 and transcribed genes MHC-B11, -B12, -B16, -B17, -B18, -B20, and -B22—most alleles remain unassigned to loci (*Heijmans et al., 2020*; *Shiina et al., 2017*). Some genes even have alternative functions, such as rhesus macaque Mamu-B*098:01 (separate gene), which is specialized to bind a 5-mer lipopeptide derived from simian immunodeficiency virus (SIV) (*de Groot et al., 2017a*).

As with the OWM MHC-A genes, the OWM MHC-B genes are arranged in distinct repeating blocks, implying that they were generated by block duplications (at least in macaques). From the complete macaque MHC haplotype, we see that there are three types of duplication blocks: one large block with six flanking pseudogenes (including fragment MHC-related pseudogene MHC-S), one medium block with two or three flanking pseudogenes, and one small block with just a single flanking pseudogene (*Karl et al., 2023*; *Shiina et al., 2011*). Furthermore, the large block with MHC-S is also found in human, gorilla, and orangutan, suggesting that this block was present in the ancestor of apes/OWM (*Karl et al., 2023*). From the ancestral large block, all MHC-B genes were generated separately in the ape and OWM lineages, meaning no gene is 1:1 orthologous between the groups; this is reflected in our trees by all OWM MHC-B genes grouping together outside of all ape MHC-B genes (*Figure 3—figure supplement 2*). As noted previously, this also means that the OWM MHC-B genes present in the large blocks are likely the oldest and that the OWM MHC-B genes present in the medium and small blocks were derived from these within the OWM (or possibly just macaque) lineage (*Karl et al., 2023*). Looking at our *BLAST* results for OWM reference genomes (*Figure 3—figure supplement 2*), we see that a similar repeating pattern of MHC-B and -S genes (large block) appears on the macaque references and the distantly-related snub-nosed monkey reference. This supports the fact that the large block was established early on. However, another pattern of MHC-B genes (intermingled small/medium blocks, without MHC-S) also seems to match between macaque and snub-nosed monkey, suggesting that the small and medium blocks of MHC-B genes were also established early on, at the beginning of OWM evolution. Different OWM haplotypes were then created via species-specific duplications and deletions.

Interestingly, while ape MHC-B genes are highly expressed, the older OWM MHC-B genes are all lowly expressed 'minors', and the newer OWM MHC-B genes are mostly highly expressed 'majors', showing how the roles of these genes can change over time (*Karl et al., 2023*). Although we included a limited number of OWM MHC-B genes in our trees, *Figure 3—figure supplements 2C and 4C* are consistent with the previously proposed block relationships, with old genes like MHC-B*098, -B02Ps, and -I forming a clade distinct from that of the newer genes, such as MHC-B11L and -B22. MHC-B19Ps (located on the far end of the OWM MHC-B segment) appears to be somewhat of an outlier, not grouping with the other OWM MHC-B sequences in ours and others' trees (*Shiina et al., 2011*). In our trees, we also included MHC-B genes from other OWM species, which scatter throughout the OWM clade. Without more sequences, it is difficult to determine whether some MHC-B genes were present before the diversification of various OWM species or if they expanded separately in different lineages.

## MHC-I

MHC-I is an OWM-specific MHC-B-related gene previously named MHC-B3. Although some past work has claimed MHC-I is fixed, other work disagrees (*Budde et al., 2010*; *Shiina et al., 2015*; *Heijmans et al., 2020*). In macaques, it occurs in one of the large repeat blocks (described above in the OWM MHC-B section) which appears to be no different than any of the other MHC-B blocks (*Karl et al., 2023*). Therefore, MHC-I is likely just a regular MHC-B paralog and is probably unfixed. Other researchers have given it a different name and deemed it non-classical because of its low levels of polymorphism and mainly intracellular location. However, this is also characteristic of 'minor' OWM MHC-B alleles, which are typically found in the older large blocks (*Adams and*

*Parham, 2001a*; *Heijmans et al., 2020*; *Karl et al., 2023*). MHC-I is supposedly present in macaque and sooty mangabey, but not in baboon, which is more closely related to the sooty mangabey than to macaque (*Heijmans et al., 2020*). Macaque (Maar/Maas/Malo/Mamu/Mane/Math-I) and sooty mangabey (Ceat-I) alleles group together in our exon 2 and exon 3 trees, but apart in our exon 4 tree (*Figure 3—figure supplement 2*). It is possible that the MHC-I genes were created separately in these species and then converged in the binding site, or that they are identical by descent and one underwent gene conversion in exon 4. Based on these trees, it seems that some MHC-B genes were established before the diversification of the macaque species, but it is unclear whether any were established earlier. More sequences are needed to determine whether MHC-I is really a special, conserved MHC-B-like gene or whether it is simply another of the many MHC-B paralogs that have been rapidly generated in the OWM.

## MHC-B in the NWM

There are multiple functional and non-functional NWM MHC-B genes, although many have low or tissue-specific expression (*Heijmans et al., 2020*). Like the OWM MHC-B genes, they are also laid out in distinct blocks, supporting their generation via block duplication (*Shiina et al., 2011*). However, the basic NWM duplication unit does not appear similar to the base unit inferred to be present in the ape/OWM ancestor (*Shiina et al., 2011*; *Karl et al., 2023*). While some researchers have questioned the orthologous relationship between ape/OWM MHC-B and NWM MHC-B, their sequences all group together in ours and others' exon 4 phylogenies (*Figure 3—figure supplements 2C and 4C*; *Shiina et al., 2011*; *Adams and Parham, 2001a*; *Sawai et al., 2004*). Thus, while no single NWM MHC-B gene is orthologous to any ape or OWM MHC-B gene, they may be broadly orthologous.

Within the NWM, the MHC-B genes show various levels of expansion in different species. The locus appears to have duplicated several times prior to the radiation of the NWM species, and duplicates have been selectively lost in different lineages (*Lugo and Cadavid, 2015*; *Sawai et al., 2004*; *Heijmans et al., 2020*). Our analysis of the reference genomes (*Figure 3—figure supplement 2*) detected 13–18 distinct MHC-B genes per species, although it is unknown whether they are fixed or even functional. In the marmoset, night monkey, and spider monkey, at least one MHC-B gene is transcribed, whereas no transcripts have been detected in the tamarin (*Kono et al., 2014*; *Lugo and Cadavid, 2015*). Additionally, $dN/dS < 1$ in the NWM suggests that the MHC-B genes are no longer classical in these species (*Cao et al., 2015*; *Cardenas et al., 2005*; *Lugo and Cadavid, 2015*). Indeed, classical peptide presentation appears to be performed by the MHC-G genes in these species (see the MHC-G section below) (*Heijmans et al., 2020*; *Lugo and Cadavid, 2015*).

## MHC-C

MHC-C is a classical Class I gene found in human (HLA-C), chimpanzee (Patr-C), bonobo (Papa-C), gorilla (Gogo/Gobe-C), and orangutan (Poab/Popy-C), but not gibbon (*Heijmans et al., 2020*; *Abi-Rached et al., 2010*). It is currently present in only 50% of orangutans but fixed in the African apes, suggesting it arose around the time of orangutan divergence (*Piontkivska and Nei, 2003*; *Fukami-Kobayashi et al., 2005*; *Abi-Rached et al., 2010*; *de Groot et al., 2016*). It is believed to have duplicated from MHC-B, and our trees confirm that it is most closely related to MHC-B (*Figure 3—figure supplement 5*; *Adams and Parham, 2001a*; *Piontkivska and Nei, 2003*; *Heijmans et al., 2020*; *Fukami-Kobayashi et al., 2005*).

Human HLA-C is less polymorphic, is less efficient at triggering T cell responses, and has one-tenth the cell surface expression of HLA-A and -B (*Guethlein et al., 2015*; *Goyos et al., 2015*; *Vollmers et al., 2021*). These data indicate that the gene has taken on a niche role. In fact, it has become the dominant Class I molecule for interacting with KIRs (*Adams and Parham, 2001a*; *Guethlein et al., 2015*; *Vollmers et al., 2021*). In all great apes, whereas a minority of MHC-A and -B alleles serve as ligands for KIR, all MHC-C alleles have this capability (*Guethlein et al., 2015*; *Vollmers et al., 2021*; *Wroblewski et al., 2019*). MHC-C alleles may either have the C1 or C2 KIR epitope, which exist as a balanced dimorphism in human, chimpanzee, and gorilla. In orangutans, all alleles contain the C1 epitope, suggesting that MHC-C duplicated from a C1-containing MHC-B allele and that the C2 epitope arose in the ancestor of the African apes. Additionally, C2 has been lost in the bonobo (*Hans et al., 2017*; *Heijmans et al., 2020*; *Wroblewski et al., 2019*).

MHC-C's interaction with KIR is not only relevant to infection, but also to pregnancy, where fetal MHC-C on extravillous trophoblasts interacts with maternal KIR to facilitate embryo implantation. Humans, chimpanzees, and gorillas all experience deep trophoblast invasion in pregnancy, while gibbons and OWM experience shallow invasion (orangutan is unknown). This may relate to the presence and absence, respectively, of MHC-C in these species, although it could also be related to MHC-G (discussed below) (*de Groot et al., 2016*).

## The MHC-E-related genes

MHC-E is a non-classical Class I gene that is present in the apes, OWM, and NWM (*Sawai et al., 2004*; *Guethlein et al., 2015*; *Paganini et al., 2019*). Expressed in most tissues, MHC-E primarily presents fragments of the leader sequences of MHC-A, -B, -C, and -G (and possibly -H) for detection by both activating and inhibitory C-type lectin receptors (CD94/NKG2) on NK cells, although it sometimes also presents pathogen-derived peptides to T cells (*Paganini et al., 2019*; *Heijmans et al., 2020*; *Lafont et al., 2004*; *Adams and Luoma, 2013*; *Lampen et al., 2013*; *Hubert et al., 2022*). MHC-E thus allows NK cells to monitor for changes to MHC Class I synthesis that could be caused by infection or other cellular stressors (*Guethlein et al., 2015*; *Otting et al., 2020*; *Lafont et al., 2004*; *Adams and Luoma, 2013*).

The PBR of MHC-E, the leader sequences that bind to it, and the KIRs that interact with it are all highly conserved, suggesting that the MHC-E–NK cell interaction has been established since prior to the diversification of the *Simiiformes* (*Adams and Parham, 2001a*; *Lafont et al., 2004*; *Knapp et al., 1998*; *Heijmans et al., 2020*). Indeed, $dN/dS < 1$ demonstrates purifying selection is acting on MHC-E (*Boyson et al., 1995*). However, humans also have a balanced MHC-E polymorphism; HLA-E*01:01 and HLA-E*01:03 (which together capture 97% of world MHC-E diversity) differ in one non-binding-site amino acid and have slightly different expression levels, suggesting heterozygote advantage (*Paganini et al., 2019*; *Lampen et al., 2013*). In the OWM, some macaques have more than one functional MHC-E gene per haplotype (*Heijmans et al., 2020*; *Karl et al., 2023*). The macaque genes are also more polymorphic, but it is unclear whether this has functional implications (*Heijmans et al., 2020*; *Shiina et al., 2017*; *Lafont et al., 2004*). In our exon 4 tree (*Figure 3—figure supplement 6C*), OWM MHC-E sequences are intermingled in a large and diverse clade, which could indicate trans-species polymorphism; this is explored further in our companion paper (*Fortier and Pritchard, 2025*). In the NWM, there may also be more than one MHC-E-like gene per haplotype (*Figure 3—figure supplement 2*). Thus, selection may be acting differently on this non-classical gene in different lineages.

In our trees (*Figure 3—figure supplements 2 and 6*, *Figure 5—figure supplement 2*), ape, OWM, and NWM MHC-E sequences form a monophyletic clade in exons 2 and 3, suggesting they are orthologous. However, in exon 4, NWM MHC-E sometimes groups more closely with MHC-B, OWM MHC-A8, or pseudogenes MHC-N or -L than with ape/OWM MHC-E. Thus, NWM MHC-E may not be 1:1 orthologous with ape/OWM MHC-E, or gene conversion could have affected the branching pattern of these genes in exon 4. On NWM reference genomes, the MHC-E region contains MHC-E-like genes as well as unknown genes, so this region has clearly deviated from the corresponding ape/OWM region (*Figure 3—figure supplement 2*).

## The MHC-F-related genes

MHC-F is a non-classical Class I gene that is believed to have fixed early on in primate evolution; it is present in apes, OWM, and NWM (*Adams and Parham, 2001a*; *Piontkivska and Nei, 2003*; *Otting et al., 2020*; *Kulski et al., 2005*; *Kulski et al., 2004*). Our trees (*Figure 3—figure supplements 2 and 7*) place ape, OWM, and NWM sequences in a monophyletic clade in all exons, supporting the orthology of this gene across these primate groups. However, tarsier and *Strepsirrhini* sequences do not group in this clade, so MHC-F was likely formed in the *Simiiformes* ancestor.

Unlike the other Class I genes, MHC-F can exist as an open conformer—that is, it can be transported to the cell surface without being bound to a peptide (see *Appendix 1—figure 1*). There, it can serve as a ligand for various activating and inhibitory KIRs involved in the innate immune system (*Paganini et al., 2019*; *Otting et al., 2020*; *Heijmans et al., 2020*). While it is typically found intracellularly, its presence on the cell surface is upregulated when the immune system is activated (*Carlini et al., 2016*; *Paganini et al., 2019*; *Otting et al., 2020*). In humans and orangutans, MHC-F

can also bind and present peptides, but they are recognized by LILRs instead of TCRs. The peptides bound can be unusually long (>30 amino acids) due to the gene's open-ended PBR. Further, this adaptation is not found in chimpanzee or gorilla, and since HLA-F and Poab/Popy-F have different mutations that enable this open-ended PBR, this function must have evolved twice (*Otting et al., 2020*; *Heijmans et al., 2020*).

In apes and OWM, there is one MHC-F gene per haplotype. In these species, $dN/dS < 1$ and variation is spread throughout the coding region instead of concentrated in the PBR, indicating the gene is under purifying selection (*Otting et al., 2020*). In the NWMs, MHC-F has expanded in some lineages, most notably in the common marmoset (Caja). Despite these expansions, $dN/dS$ is still less than 1 and most duplicates have become pseudogenes, again indicating purifying selection. Typically, only one MHC-F gene is functional in each species, except for the marmoset, which may have two (*Otting et al., 2020*; *Lugo and Cadavid, 2015*). Because the MHC-F genes are interspersed with MHC-G genes in NWM, their expansion is likely a side effect of the expansion of MHC-G, which took over classical functionality in this lineage (*Kono et al., 2014*; *Lugo and Cadavid, 2015*).

## The MHC-G-related genes

### MHC-G in the apes

In the great apes, MHC-G is a non-classical gene; there is one copy of MHC-G per haplotype and it has limited polymorphism and a specialized role in reproduction (*Heijmans et al., 2020*; *Piontkivska and Nei, 2003*). Human HLA-G is expressed only in extravillous trophoblasts—fetal cells which invade the uterine wall during embryo implantation. Uterine NK cells interact with trophoblast HLA-G via LILRs, preventing maternal immune cells from destroying fetal cells and facilitating deep trophoblast invasion of the uterine wall (*Adams and Parham, 2001a*; *Guethlein et al., 2015*; *Otting et al., 2020*). MHC-G was deleted from the gibbon genome and inactivated in OWM, and the lack of MHC-G, MHC-C, and the MHC-C-associated KIRs in both groups correlates with a shallower trophoblast invasion in these species during pregnancy compared to humans, chimpanzees, and gorillas (*Abi-Rached et al., 2010*; *Carter, 2021*; *de Groot et al., 2016*).

### MHC-G in the OWM

MHC-G is a pseudogene in the OWM. However, apparently compensating for the pseudogenization of MHC-G in the OWM, a new gene group called MHC-AG arose from OWM MHC-A (see above section on MHC-A) (*Heijmans et al., 2020*). The number of MHC-G genes per haplotype varies by species in the OWM (*Figure 1—figure supplement 1*). In the macaques, there are multiple MHC-AG genes and MHC-G pseudogenes per haplotype, as they are both part of repeat units that make up the α-block (*Karl et al., 2023*; *Kulski et al., 2004*). The MHC-AG genes have converged in function to MHC-G, with the same exclusive tissue distribution, limited polymorphism, and even similar splice variants (*Adams and Parham, 2001a*; *Bondarenko et al., 2007*; *Shiina et al., 2011*; *Nicholas et al., 2022*). Our trees (*Figure 3—figure supplement 8*) firmly group the MHC-AG genes with OWM MHC-A and the OWM MHC-G genes with ape MHC-G genes, showing that this is indeed an example of convergent evolution. In humans, MHC-G exerts its influence on placental vascularization via activating KIRs, and indeed MHC-AG genes primarily interact with activating KIRs (*Anderson et al., 2023*). Thus, MHC-AG has somewhat rescued the deep-trophoblast-invasion phenotype in the OWM, although it is not quite as deep as in human, suggesting that the genes are not entirely functionally equivalent or that other genes (like MHC-C) are also important (*Moffett-King, 2002*; *Carter, 2021*; *Nicholas et al., 2022*).

### MHC-G in the NWM

NWM MHC-G is not thought to be 1:1 orthologous to ape/OWM MHC-G, as it groups outside of several ape/OWM genes instead of forming a monophyletic clade with ape/OWM MHC-G (*Sawai et al., 2004*; *Adams and Parham, 2001a*). Our trees agree (*Figure 3—figure supplements 2 and 8*), with ape/OWM MHC-G grouping with various other genes (depending on exon) instead of with NWM MHC-G. In the Class I α-block, only MHC-F is shared by apes/OWM and NWM (see above section on MHC-F); no other shared genes or pseudogenes have been found. Since NWM MHC-G groups apart from all other ape/NWM genes and none of the other related genes in the block are shared, independent expansions likely occurred in apes/OWM (generating all other α-block genes)

and NWM (generating the many MHC-G paralogs). The grouping of some NWM MHC-G alleles with ape and OWM MHC-G alleles in exon 2 thus appears to be the work of convergent evolution (*Figure 3—figure supplement 8A*).

Other than MHC-G, the only Class I genes in the NWM are non-classical MHC-E, non-classical MHC-F, and lowly expressed and often pseudogenized MHC-B. Classical peptide presentation in the NWM is entirely governed by the greatly expanded MHC-G genes, which have ubiquitous expression and extreme polymorphism (*Watkins et al., 1990*; *Heijmans et al., 2020*; *van der Wiel et al., 2013*; *Kono et al., 2014*). As is typically the case with MHC gene expansions, NWM MHC-G sequences have generally not been assigned to loci. This makes it difficult to determine which genes (if any) were generated before which speciation events. Our trees (*Figure 3—figure supplement 8*) show many NWM clades which are each limited to just one or two closely related species; these clades may represent individual genes or species-specific allelic lineages of the same gene. One named MHC-G-related gene, a processed pseudogene called MHC-PS2, appears to be shared by the tamarin, marmoset, and night monkey (*Figure 3—figure supplement 8C*), showing it is relatively long-lived. Although the expansion of MHC-G is evident in nearly all NWMs, different MHC-G genes do not often appear to be shared across species, suggesting independent expansions in different species or extremely rapid birth-and-death evolution (*Lugo and Cadavid, 2015*; *Watkins et al., 1990*; *van der Wiel et al., 2013*; *Sawai et al., 2004*).

Because NWM MHC-G is not directly orthologous to ape/OWM MHC-G and no other genes appear to have taken over the ape/OWM MHC-G role, the NWM do not benefit from the maternal-fetal tolerance function of MHC-G. Thus, the MHC Class I genes' role in pregnancy immunotolerance appears to be nonessential to the NWM. Indeed, both NWM and the *Strepsirrhini* have successful pregnancies despite lacking both MHC-G-like genes and the deep trophoblast invasion phenotype (*Parham and Moffett, 2013*; *Carter, 2021*).

## MHC-J

MHC-F—fixed early in primate evolution—marks the telomeric start of the Class I $\alpha$-block, while full-length pseudogene MHC-J marks its centromeric end (*Adams and Parham, 2001a*; *Kulski et al., 2004*; *Shiina et al., 2017*). It is present as a single copy in apes and most OWM (there are two in the gelada, *Figure 3—figure supplement 2*), but so far has not been found in NWM (*Karl et al., 2023*; *Gleimer et al., 2011*; *Adams and Parham, 2001a*; *Abi-Rached et al., 2010*). Thus, it must have been established fairly early on in the evolution of the α-block, around the time of the ape/OWM ancestor. In humans, it also appears to be transcribed even though it is a pseudogene (*Horton et al., 2008*).

In humans, HLA-G and -J have similar insertion elements in their 5'-end flanking regions, strongly supporting a close relationship (*Sawai et al., 2004*). Previous phylogenetic analyses have also shown MHC-G and -J grouping together (*Hughes, 1995*; *Alexandrov et al., 2023*; *Messer et al., 1992*; *Sawai et al., 2004*). Our exon 3 trees group ape/OWM MHC-J with ape/OWM MHC-G, while our exon 2 trees group ape/OWM MHC-J outside of a combined ape/OWM MHC-K and -G clade and our exon 4 trees group ape/OWM MHC-J with MHC-G and -A (*Figure 3—figure supplement 2*, *Figure 5—figure supplements 1 and 2*). Thus, we support a close relationship of MHC-J and -G, although MHC-K and -A may also be related to them in specific regions because of gene conversion or recombination early on in the genes' history.

## Other pseudogenes in the Class I α-block

### The MHC-W-related subfamily

Human HLA-T, -OLI, -P, and -W are pseudogenes interspersed among functional genes in the Class I α-block (*Shiina et al., 2017*). They are known to be related based on phylogenetic analysis and structural comparison and, in fact, form a clearly distinct group apart from the other human Class I α-block genes (*Alexandrov et al., 2023*). Indeed, our trees (*Figure 5—figure supplement 1*) also definitively divide human and non-human genes into the MHC-T/OLI/P/W subfamily or into the subfamily containing all other genes. This implies that the ancestral Class I region contained two distinct proto-genes which were repeatedly duplicated together (creating the interspersed arrangement of the α-block) and separately (as there appear to be no MHC-T/OLI/P/W subfamily genes in the κ- or β-blocks).

MHC-T and -W also have close relatives—MHC-TL and -WL—present on some great ape haplotypes (*Gleimer et al., 2011*). This is due to a large block duplication that occurred in the great ape ancestor, separating MHC-H from MHC-A/AL/OKO, -T from -TL, -K from -KL, and -W from -WL (see *Figure 6*; *Gleimer et al., 2011*). The repeated arrangement of these similar genes allowed researchers to deduce that a block duplication occurred; pseudogenes can therefore provide clues as to the birth-and-death processes that have shaped the MHC region. Our work gives special attention to these often-ignored pseudogenes and sheds light on their relationships between species.

## MHC-T and -TL

MHC-TL (T-Like) is a fragment pseudogene containing exons 3–7, while MHC-T suffered a deletion and now contains only exons 4–7. Only MHC-T is present on human haplotypes, while chimpanzee and gorilla haplotypes can contain either just MHC-T or both MHC-T and -TL (*Figure 6*). As shown in *Figure 5—figure supplement 1C*, human, chimpanzee, and gorilla MHC-T sequences group together, so they are likely orthologous.

Our *BLAST* search uncovered two MHC-TL-like genes in the Sumatran orangutan (Poab) and one in the Bornean orangutan (Popy) (*Figure 5—figure supplement 2*). Our trees show that both Sumatran orangutan sequences have exon 3 and group with MHC-TL sequences (*Figure 5—figure supplement 1*), suggesting that they are both actually MHC-TL orthologs and that the MHC-T gene may have been deleted from the orangutan (at least from some haplotypes). Interestingly, one orangutan MHC-TL sequence groups with chimpanzee MHC-TL, while another groups with gorilla MHC-TL.

The OWM do not have any named MHC-T or -TL genes; instead, the OWM $\alpha$-block contains repeating blocks with fragment pseudogenes all called MHC-W. However, our work suggests that some of these genes are in fact orthologous to MHC-T/TL of the apes and should be renamed (see below section on the OWM MHC-W genes).

## MHC-W and -WL

MHC-W and -WL (W-Like) are both fragment pseudogenes containing exons 3–7. Only MHC-W is found on human haplotypes, while chimpanzees have haplotypes with both MHC-W and -WL or just MHC-W. Gorilla haplotypes have either MHC-W or -WL, but not both (*Figure 6*). Our trees (*Figure 5—figure supplement 1*) show that ape MHC-W is closely related to ape MHC-WL, as expected.

Our analysis of the orangutan reference genomes (*Figure 3—figure supplement 2*) reveals a MHC-W-like gene between MHC-H and -A; this placement suggests it is orthologous to MHC-WL. The siamang and Northern white-cheeked gibbon reference genomes also contain MHC-W-like genes between MHC-A and -J. They group outside of the other ape MHC-W and -WL genes (*Figure 5—figure supplement 1*), so the block duplication separating MHC-W and -WL likely occurred after the divergence of the gibbon. Curiously, we did not detect a MHC-W (or MHC-A) gene in the pileated gibbon genome; sequencing of more haplotypes will be necessary to explore this. OWM MHC-W will be discussed below.

## MHC-P

MHC-P is a fragment pseudogene containing exons 3–7 (*Alexandrov et al., 2023*). In the human genome, HLA-P is located very close to HLA-V, and they may even be transcribed together. This has led some to conclude that they are parts of the same gene, but we disagree (see below section on MHC-V) (*Horton et al., 2008*).

MHC-P is present in the same location in human, chimpanzee, bonobo, gorilla, and both species of orangutan. These sequences clearly group together in our trees (*Figure 5—figure supplement 1*), so they are likely orthologous. They also group with some OWM MHC-W genes (which will be discussed below).

## MHC-OLI

MHC-OLI is a recently discovered fragment pseudogene that has a nearly identical structure (but only 88% similarity) to MHC-P (*Alexandrov et al., 2023*). In humans, it is completely linked to

pseudogene HLA-Y, and the two are found on a 60 kb insertion present in 29% (27/94) of human haplotypes (*Alexandrov et al., 2023*; *Zhou et al., 2024*). Its discoverers also found a nearly identical 60 kb sequence in the gorilla (we could not replicate the result as the sequence was suppressed). While this is unconfirmed, the presence of MHC-Y in the gorilla implies that MHC-OLI is also present in that species (*Alexandrov et al., 2023*). MHC-OLI also could not be found in the chimpanzee, consistent with the lack of MHC-Y in this species.

Our trees (*Figure 5—figure supplement 1*) show that MHC-OLI sequences group with MHC-TL. Because MHC-Y and -OLI are completely linked, it is reasonable to assume they were duplicated as a unit. Since MHC-Y is believed to be most closely related to MHC-A/AL/OKO, and MHC-TL is located right next to MHC-A/AL/OKO in chimpanzee and gorilla haplotypes, it is likely that MHC-Y and -OLI were created from MHC-A/AL/OKO and MHC-TL simultaneously via a block duplication event (*Figure 6*). This is also consistent with a structural comparison of these genes; although non-human genes were not included in the original HLA-OLI paper (*Alexandrov et al., 2023*), our work reveals that MHC-TL (exons 3–7) is longer than MHC-T (exons 4–7) and so has the same exon content as MHC-OLI. Additionally, while MHC-P is also the right size, MHC-P is only 88% similar to MHC-OLI, whereas MHC-T and -TL are much more similar to it (*Figure 5—figure supplement 1*).

## The MHC-W-related genes of the OWM

MHC-T/TL and -P genes have not yet been reported in the OWM. Instead, the published macaque genomes include multiple copies of genes named MHC-W (*Karl et al., 2023*). In *Figure 5—figure supplement 1C*, we see that OWM MHC-W genes fall into four distinct clades, three of which are each more closely related to an ape clade than to each other. Furthermore, the alleles which compose each clade correspond perfectly to different types of repeat blocks on the macaque genome. The first clade contains alleles Mafa-W*01:01, Mafa-W*01:02, Mafa-W*01:11, and Mafa-W*01:12 and groups with ape MHC-T, -TL, and -OLI. These macaque MHC-W alleles are all located on the antisense strand and are part of the MHC-W – -AG – -V – -G repeating block (*Karl et al., 2023*). The second clade contains the macaque allele Mafa-W*01:05 and groups with ape MHC-P; this allele is located next to MHC-Apseudo and is not part of either repeating block on the macaque haplotype (*Figure 5—figure supplement 2*; *Karl et al., 2023*). The third clade contains the macaque alleles Mafa-W*01:07 and Mafa-W*01:09 and groups with the ape MHC-W and -WL genes. These alleles are associated with the MHC-A – -W – -K – -W repeating block in the macaque, specifically as the second MHC-W in each repeat (*Karl et al., 2023*). The final clade contains Mafa-W*01:06, Mafa-W*01:08, and Mafa-W*01:10 and groups outside of all of the other OWM MHC-W and ape MHC-W/P/T/OLI genes. These alleles are found as the first MHC-W of each MHC-A – -W – -K – -W repeat block in the macaque (*Karl et al., 2023*).

This suggests that the many MHC-W genes in the OWM were not derived from a single gene which expanded separately in apes and OWM; instead, separate genes were already established (one in each type of repeat block) in the ape/OWM ancestor. This is consistent with our *BLAST* results of the reference genomes (*Figure 5—figure supplement 2*), which show that the common ancestor of all OWM probably had an $\alpha$-block containing MHC-F, -W (first type), -AG, -V, -G, -Apseudo, -W (second type), -K, -W (fourth type), -A, -W (third type), and -J. Combining all of this evidence, *Figure 6* shows our new hypothesis for the evolution of this region. In the ape/OWM ancestor, there was likely a MHC-T/TL-like gene (Mafa-W*01:06/08/10/01/02/11/12) near the OWM MHC-A ancestor, an MHC-P-like gene (Mafa-W*05) near the OWM MHC-Apseudo/ape MHC-A ancestor, and an MHC-W-like gene (Mafa-W*01:07/09) near the ape/OWM MHC-K ancestor. In the OWM lineage, the MHC-T/TL-like gene then duplicated and inverted (along with MHC-A) to form Mafa-W*01:01/02/11/12 and MHC-AG (*Kulski et al., 2004*). These OWM MHC-W genes/alleles should therefore be renamed to more clearly correspond with closely related loci in the apes.

## MHC-K and -KL

MHC-K and -KL (K-Like) are closely related full-length pseudogenes in the $\alpha$-block that were separated as part of the block duplication that occurred in the apes (which separated MHC-H from MHC-A/AL/OKO, MHC-T from MHC-TL, and MHC-W from MHC-WL). Only MHC-K is found on human haplotypes. Chimpanzees have haplotypes with both MHC-K and -KL or just MHC-K, and

gorilla haplotypes have either MHC-K or -KL, but not both (*Figure 6*). We did not detect MHC-K or -KL on the orangutan or gibbon reference genomes (*Figure 5—figure supplement 2*).

MHC-K is present in the OWM as well. We detected one copy of MHC-K in the gelada and golden snub-nosed monkey, but none in the mantled guereza or baboon (*Figure 5—figure supplement 2*). In the macaque, there can be many copies of MHC-K because it is part of the repeat block containing MHC-A (*Karl et al., 2023*).

Our trees (*Figure 5C*, *Figure 5—figure supplement 1*) clearly show that MHC-K and -KL are closely related. The OWM MHC-K sequences form a clade outside of the combined MHC-K/KL clade, as expected. Because it is orthologous between apes and OWM, MHC-K is likely an old pseudogene, formed around the same time as the rest of the Class I $\alpha$-block in the ape/OWM ancestor. The insertion elements in MHC-K are similar to those in MHC-G, -J, -F, and -A (*Sawai et al., 2004*; *Neehus et al., 2016*). Reflecting this uncertainty, our exon 2 trees show MHC-K grouping outside of MHC-A and -F, while in exon 3, it groups with MHC-F and in exon 4, it groups outside of MHC-A, G, and -J (*Figure 5—figure supplement 2*). This difference in branching pattern between exons reveals an early history of recombination and/or gene conversion in the region as the genes were first formed. Our hypothesis for the formation of MHC-K can be found in *Figure 6*.

## MHC-V

MHC-V is a fragment pseudogene containing only the 5' end of a typical MHC gene, exons 1–3. It is located near MHC-F at the telomeric end of the $\alpha$-block. A single copy is present in human, chimpanzee, and gorilla, but multiple copies are present in the macaque, as is part of a repeating unit containing MHC-AG (*Shiina et al., 2017*; *Anzai et al., 2003*; *Wilming et al., 2013*; *Karl et al., 2023*). One study has claimed that human MHC-V is transcribed together with the nearby 3'-end fragment pseudogene MHC-P (and thus they should be considered the same gene), but other than that, nothing is known about MHC-V (*Horton et al., 2008*).

We built trees with the available MHC-V exons (exons 2 and 3; *Figure 5—figure supplements 1A-B and 2A-B*) and discovered that MHC-V does not group strongly with any particular gene. In exon 2, it groups with MHC-E and NWM MHC-G; since these are deeply diverged genes located in entirely different blocks, this suggests MHC-V is also old. In exon 3, the MHC-V clade is an outgroup to all other Class I genes except for the MHC-W/P/T/OLI family of pseudogenes, also supporting its old age. MHC-V is located near MHC-F—which was fixed early, before the ape/OWM and NWM divergence—further supporting its early origins. We claim that MHC-V is an old remnant of the early evolution of the region, distinct from both the MHC-W/P/T family of pseudogenes and the rest of the Class I genes. Lastly, since both MHC-V and -P contain an exon 3 and MHC-V's exon 3 clearly does not group with the MHC-W/P/T family of pseudogenes, we doubt that MHC-V and -P started as two halves of the same gene, even if they might now be transcribed together (*Horton et al., 2008*)

## MHC-U

MHC-U is a single-exon pseudogene known to be present in human and chimpanzee, but nothing else was previously known (*Shiina et al., 2017*; *Gleimer et al., 2011*). We discovered MHC-U in the bonobo (*Figure 5—figure supplement 2*) and found that MHC-U sequences aligned well with other genes' exon 3 sequences. Our exon 3 tree (*Figure 5B*) groups the MHC-U sequences with a clade of human, chimpanzee, and bonobo MHC-A, suggesting it duplicated from MHC-A in the ancestor of these three species. Because one MHC-U-containing human haplotype is shared between human and gorilla (*Figure 6*), we expect that MHC-U will also be found in the gorilla as well as more haplotypes are sequenced (the reference genome and separate gorilla MHC haplotype (*Wilming et al., 2013*) are the non-MHC-U-containing haplotype). Ours is the first work to show that MHC-U is actually an MHC-A-related gene fragment and that it likely originated in the human/gorilla ancestor.

## Other pseudogenes in the Class I κ-block

### MHC-L

MHC-L is a full-length pseudogene located in the Class I κ-block along with fragment pseudogene MHC-N and non-classical MHC-E. Unlike the $\alpha$-block, this region has undergone relatively few changes in the history of the primates. Our *BLAST* search revealed a single copy of MHC-L in all ape

and OWM reference genomes that we tested, but none in the NWM. The gelada was an exception with two MHC-L copies, apparently owing to duplication of a very large region spanning part of the $\alpha$-block and all of the κ-block (*Figure 3—figure supplement 2*).

MHC-L is currently classified as a pseudogene in human, chimpanzee, gorilla, gibbon, and macaque (*Shiina et al., 2017*; *Anzai et al., 2003*; *Wilming et al., 2013*; *Abi-Rached et al., 2010*; *Karl et al., 2023*). It is unknown whether it is currently functional in some species or whether it was functional in the past.

Our trees group MHC-L sequences of the apes and OWM together, showing they are orthologous (*Figure 5—figure supplement 2*). However, MHC-L's relationship to other genes is somewhat ambiguous. It shares similar insertion elements and sequence homology with MHC-B and -C (*Sawai et al., 2004*; *Adams and Parham, 2001a*). However, in our trees, the MHC-L clade groups with MHC-G/J/K in exon 2, with MHC-K/F in exon 3, and outside of MHC-W/T in exon 4. The uncertain placement of MHC-L and its orthology between apes and OWM means that it was probably formed in the ape/OWM ancestor and was subject to gene conversion/recombination early in the genes' history.

## MHC-N

MHC-N is a fragment pseudogene also located in the $\kappa$-block along with full-length pseudogene MHC-L and non-classical MHC-E. Aside from its presence in human, chimpanzee, gorilla, and macaque, nothing is known about it (*Shiina et al., 2017*; *Anzai et al., 2003*; *Wilming et al., 2013*; *Karl et al., 2023*). Our *BLAST* search of the reference genomes shows that MHC-N is present on all ape and OWM haplotypes that we tested. It is present as a single copy in all species except for gelada, which has had a large block duplication and thus has two copies (*Figure 3—figure supplement 2*).

We show that the MHC-N sequence aligns well with exon 4 of the other genes. Additionally, our exon 4 trees place MHC-N on a long branch, and it is not strongly associated with any other gene. Our Class I tree groups it with NWM MHC-E and -B (*Figure 3—figure supplement 2*), our "other" genes tree (*Figure 5—figure supplement 2C*) places it with the tarsier sequence and ape/OWM/NWM MHC-F sequence, and our $\alpha$-block-focused tree (*Figure 5—figure supplement 1C*) shows it most closely related to ape/OWM MHC-F and -L. Its presence in apes/OWM and its association with genes in all blocks and with ape/OWM/NWM/tarsier sequences could mean that it is a very old fragment, or that it has experienced relaxed selection and no longer contains many phylogenetically informative variants.

## Other Pseudogenes in the Class I β-Block
### MHC-S

MHC-S is a partial pseudogene spanning exons 6–8 that is located near MHC-B in the Class I β-block. Our *BLAST* search of the reference genomes uncovered one MHC-S copy in each of the great ape species, located a consistent distance from MHC-B on all the haplotypes (*Figure 3—figure supplement 2*). We also found two copies of MHC-S in the Northern white-cheeked gibbon (also the same distance from the two MHC-B genes) and one in the pileated gibbon (closer to MHC-B), but none in the siamang.

In the OWM, there are multiple copies of MHC-S, as they are associated with MHC-B in one of the three types of MHC-B-region duplication blocks. This block ("large" block) was formed before the divergence of the apes and OWM, as pseudogenes in the exact same arrangement are found in humans and macaques (*Karl et al., 2023*). There are varying numbers of large blocks (and thus MHC-S copies) per haplotype in the OWM, and we found two in the Formosan rock macaque, nine in the crab-eating macaque, and four in the snub-nosed monkey (*Figure 3—figure supplement 2*). There were also additional copies of MHC-S that appeared to be outside of large blocks: one additional in the baboon, two in the gelada, and one in the Formosan rock macaque. The Tibetan macaque had six copies of MHC-S that were arranged in opposite-orientation pairs, the result of an inversion and subsequent duplications. We did not uncover any MHC-S copies in the mantled guereza.

We did not include MHC-S in our trees because we focused on exons 2–4.

## MHC-X

MHC-X is an intronic MHC fragment pseudogene that has been identified in both human and gibbon, but it was deleted from both chimpanzee and gorilla as a consequence of the fusion of MICA and MICB (*Abi-Rached et al., 2010*; *Shiina et al., 2017*; *Wilming et al., 2013*; *Anzai et al., 2003*). We did not include it in our analyses because it does not align with any exons.

## Even More Pseudogenes

### MHC-Z

Curiously, MHC-Z is a Class I pseudogene that is located in the heart of the Class II region; it has been identified in human and macaque (*Shiina et al., 2017*). It is a partial pseudogene with homology to intronic MHC Class I sequences, so it is not studied here.

## Class I genes beyond the *Haplorrhini*

Orthology among the Class I genes is generally short-lived due to rapid birth-and-death evolution. This is true even for the most conserved Class I genes, MHC-E and -F (*Nei and Rooney, 2005*). As a result, true orthologs for the Class I genes have only been detected among the apes, OWM, and NWM. Although not much is known about the MHC of the tarsiers or the *Strepsirrhini*, two studies on the Class I genes of lemurs show that they group separately from all *Haplorrhini* genes in phylogenetic trees (*Flügge et al., 2002*; *Go et al., 2003*). The MHC region of the mouse and rat has been well characterized, and the overall configuration of the region is conserved, even though birth-and-death evolution has resulted in different sets of expanded genes and the loss of orthology between rodents and primates (*Shiina et al., 2017*). In the mouse, the H2-K, -D, and -L genes are classical, while the H2-Q, -M, and -T genes are non-classical (*Riegert et al., 1998*; *Shiina et al., 2017*; *Gu and Nei, 1999*). The H2-Q genes have diverse functions, ranging from MHC-E-like self-peptide presentation to broad peptide-binding roles (*Riegert et al., 1998*). The H2-M and -T genes vary widely in function, with some completely unrelated to immunity (*Shiina et al., 2017*). In the rat, non-classical loci include RT1-N, which is orthologous to mouse H2-T, RT1-M, which is orthologous to mouse H2-M, and RT1-CE, which is orthologous to H2-D/L/Q. The only classical locus in the rat is RT1-A, containing 4 genes which are not all present on every haplotype (*Walter, 2020*).

## The Class II subfamily

Each Class II molecule is made up of two proteins, an $\alpha$ chain and a β chain. In the primates, there are three classical molecules—MHC-DP, MHC-DQ, and MHC-DR—and two non-classical molecules—MHC-DM and MHC-DO. Each of these has a corresponding locus containing at least one A gene which encodes the $\alpha$ chain and at least one B gene which encodes the β chain. The A and B genes for each pair are usually located near each other in opposite transcriptional orientation. This has led to the conclusion that the loci arose via block duplications each copying both members of the pair (*Kaufman, 2022*; *Takahashi et al., 2000*).

The MHC Class II genes do not appear to undergo rapid birth-and-death evolution like the Class I genes. Proposed explanations for this include the fact that the Class II genes currently exhibit exclusive A-B pairing (so potentially need to co-evolve as pairs) and that they may have relaxed selective pressure owing to either their more limited tissue distribution or their ability to bind peptides more flexibly compared to Class I (*Go et al., 2003*; *Yeager and Hughes, 1999*). In any case, individual Class II genes are generally older than Class I genes. Researchers generally agree that MHC-DMA and -DMB are the oldest genes, present in birds, fish, and amphibians as well as mammals (*Dijkstra and Yamaguchi, 2019*). One study estimated divergence times for the Class II genes—for the Class IIA genes, the ancestral MHC-DPA/DRA gene diverged from the ancestral MHC-DOA/DQA gene around 190mya, and each pair subsequently diverged ~175mya. For the Class IIB genes, MHC-DOB diverged from the ancestral MHC-DRB/DQB/DPB gene around 250mya, followed by the divergence of MHC-DRB from ancestral MHC-DQB/DPB around 185mya. Lastly, the divergence of MHC-DQB from MHC-DPB occurred ~175mya (*Takahashi et al., 2000*). Note that the evolutionary histories of the Class IIA and Class IIB genes are different, which is unexpected given the current physical locations and exclusive pairings of corresponding A and B genes (*Takahashi et al., 2000*; *Yeager and Hughes, 1999*). This suggests that A-B pairings were more promiscuous

in the past, or that some genes were not duplicated as a pair. These ages suggest that the Class IIB genes originated in the common ancestor of all mammals, so orthology may be intact between species as diverged as humans and marsupials (*Yeager and Hughes, 1999*; *Takahashi et al., 2000*; *Benton et al., 2015*).

## The MHC-DP region

All apes, OWM, and NWM have two MHC-DP pairs. In apes, OWM, and most NWM, MHC-DPA1/DPB1 encode functional products, while MHC-DPA2/DPB2 are pseudogenes (*Heijmans et al., 2020*). All apes, OWM, and NWM also appear to have an additional partial pseudogene MHC-DPA3 (*Figure 2—figure supplement 2*; *Shiina et al., 2017*). Additionally, even the MHC-DPA1 and -DPB1 genes of the marmoset (NWM) appear to be inactive (*Heijmans et al., 2020*).

Gene conversion has played a major role in the diversification of MHC-DPB1, resulting in thousands of alleles which share short motifs but otherwise do not cluster into clear allelic lineages (*Go et al., 2003*; *de Groot et al., 2020*).

In our *BLAST* search of the reference genomes, we found a single copy of MHC-DPA and -DPB in the gray mouse lemur, black-and-white ruffed lemur, loris, and flying lemur, and two pairs in the ring-tailed lemur. We did not detect MHC-DP (or MHC-DR) in the mongoose lemur, so these genes may be located on other chromosomes, the reference may be incomplete, or these genes may be absent in this species (*Figure 2—figure supplement 2*). The primate DP region is orthologous to H2-P in the mouse and RT1-H in the rat, but the rodent genes are nonfunctional (*Shiina et al., 2017*; *Walter, 2020*; *Gu and Nei, 1999*).

## The MHC-DQ region

There are also two pairs of DQ genes in humans: HLA-DQA1, -DQB1, -DQA2, and -DQB2. Both pairs are functional; however, HLA-DQA2/DQB2 has even more limited expression than the other Class II molecules, appearing only on epidermal Langerhans cells. Additionally, these genes do not always pair exclusively. HLA-DQA2 can appear in a mixed heterodimer with MHC-DQB1, whereas the reverse is not true—HLA-DQA1 does not appear to associate with HLA-DQB2 (*Lenormand et al., 2012*).

In the OWM, MHC-DQA2/DQB2 have been deleted, leaving behind only a small MHC-DQB2-like fragment (*Heijmans et al., 2020*). However, there appears to have been an additional duplication in the Tibetan macaque, leaving this species with two pairs of MHC-DQB genes (although their functionality is unknown) (*Figure 2—figure supplement 2*). The NWM have two or even three pairs of MHC-DQ genes, but they may not be 1:1 orthologous with the ape/OWM genes. Their position on the haplotypes varies (*Figure 2—figure supplement 2*).

Our *BLAST* search of the reference genomes also revealed two pairs of MHC-DQ genes in the gray mouse lemur and one pair in each of the other lemurs. There were no MHC-DQ genes on the reference haplotypes of the loris or flying lemur (*Figure 2—figure supplement 2*). Rodents have functional orthologs of MHC-DQ named H2-A in the mouse and RT1-B in the rat (*Gu and Nei, 1999*; *Walter, 2020*).

## The MHC-DR region

Whereas the other Class II loci exist as dedicated A-B gene pairs, the MHC-DR locus consists of just one MHC-DRA gene and many MHC-DRB genes. The MHC-DRA gene has limited polymorphism and is highly conserved across species, while the MHC-DRB genes have expanded and diversified in many lineages (*de Groot et al., 2020*; *Yeager and Hughes, 1999*; *Takahashi et al., 2000*; *Slierendregt et al., 1992*). Because MHC-DRA is essentially monomorphic, the MHC-DRB genes have presumably experienced relaxed coevolution and have been able to evolve by processes similar to that of the Class I genes (*Yeager and Hughes, 1999*). Insertion elements suggest that by the time of the common ancestor of the apes and OWM, there were at least 4 MHC-DRB genes present (*Doxiadis et al., 2012*). These genes have continued to diversify; currently, there are 9 HLA-DRB genes in humans, but only 3–4 (HLA-DRB1, -DRB3, -DRB5, and sometimes -DRB4) are functional (*Doxiadis et al., 2012*; *Klein et al., 2007*). Additionally, not every haplotype contains every gene; each of the five known human haplotypes consists of one HLA-DRA and 1–4 HLA-DRB genes. In chimpanzees, there are nine different MHC-DR haplotypes, each consisting of one MHC-

DRA and 2–5 MHC-DRB genes. Interestingly, humans and chimpanzees share just one of these MHC-DR configuration haplotypes, suggesting very rapid evolution of the MHC-DR region.

The OWM have even more haplotype diversity, with each of the >30 haplotypes consisting of one MHC-DRA gene and 2–6 MHC-DRB genes (*Heijmans et al., 2020*; *Doxiadis et al., 2012*). Of all of these genes, only MHC-DRB9, duplicates MHC-DRB2/DRB6, and potentially MHC-DRB5 appear to be orthologous between OWM and apes (*Doxiadis et al., 2012*; *Klein et al., 2007*). In contrast to the extreme diversity of MHC-DRB in the apes and OWM, most NWM have few haplotypes. For example, the only known marmoset haplotype consists of one MHC-DRA and three MHC-DRB genes. The MHC-DRB genes in the NWM exhibit limited polymorphism, and one even appears to be a pseudogene. Further work is needed to characterize MHC-DRB haplotype diversity in the NWM. It is also unclear whether any of the NWM MHC-DRB genes are orthologous to any of the ape or OWM genes, or if birth-and-death evolution has erased 1:1 orthology between these groups (*Heijmans et al., 2020*).

Beyond the primates, orthologs of the MHC-DR genes are also present in rodents, named RT1-D in the rat and H2-E in the mouse. However, they have some quirks. The H2-E genes are not present on all mouse haplotypes (*Walter, 2020*; *Shiina et al., 2017*). Additionally, the mouse and rat genes have a new Class IIB-unrelated terminal exon replacing the typical Class IIB exons 4–6, and exon 3 (instead of exon 2) is the highly polymorphic exon (*Walter, 2020*).

## The MHC-DM region

MHC-DM is a non-classical Class II molecule that helps the other Class II molecules with peptide loading. Like the classical Class II molecules, it is expressed in all antigen-presenting cells (*Welsh and Sadegh-Nasseri, 2020*). The MHC-DM genes are also the oldest of the Class II genes, having originated early in the history of the MHC in the ancestor of all jawed vertebrates (*Takahashi et al., 2000*; *Dijkstra and Yamaguchi, 2019*; *Flajnik and Kasahara, 2001*).

As a reminder of the Class II peptide-presentation pathway (*Appendix 1—figure 1*), Class II molecules are synthesized in the ER, bind the invariant chain (Ii), then are transported from the ER to a specialized compartment (*Welsh and Sadegh-Nasseri, 2020*; *Neefjes et al., 2011*). Once there, Ii is trimmed (and is thereafter known as CLIP, the Class II-associated invariant chain peptide).

MHC-DM has a very similar structure to the classical Class II molecules, but it does not bind peptides itself (*Welsh and Sadegh-Nasseri, 2020*). Instead, it catalyzes the removal of CLIP, freeing up the Class II molecules to bind other relevant peptides (*Neefjes et al., 2011*; *Welsh and Sadegh-Nasseri, 2020*). In the absence of MHC-DM, CLIP would be presented on the cell surface at high levels along with other peptides that were able to bind without the help of the catalyst (*Budeus et al., 2024*; *Olsson et al., 2022*).

This process means that MHC-DM has a role in peptide selection. Specifically, MHC-DM interacts with peptide-bound MHC-DR molecules that have an empty P1 pocket—that is, those carrying an ill-fitting peptide. In doing so, MHC-DM changes the conformation of the MHC-DR molecule's binding site to release these suboptimal peptides (*Welsh and Sadegh-Nasseri, 2020*). This helps filter out peptides that are too small or bind too weakly; these would make the MHC-DR-peptide molecule unstable, reducing its half-life and thus limiting the window of possible detection by T cells (*Budeus et al., 2024*; *Dijkstra and Yamaguchi, 2019*). This process (called 'peptide editing') repeats until the MHC-DR molecule carries a well-fitting peptide, thus shaping the repertoire of peptides presented (*Welsh and Sadegh-Nasseri, 2020*).

MHC-DM does not interact with all molecules equally, mainly affecting MHC-DR. MHC-DQ molecules have very low susceptibility to MHC-DM binding, so very high levels of MHC-DM are required for it to perform peptide editing on them (*Olsson et al., 2022*; *Welsh et al., 2019*). Similarly, MHC-DP receives only minor benefit from MHC-DM's peptide-editing function (*van Lith et al., 2010*).

## The MHC-DO region

MHC-DO is another non-classical Class II molecule that acts as a modifier of MHC-DM activity. MHC-DO has a recognizable Class II structure including an open binding groove, but it does not bind peptides. It is always found alongside MHC-DM in a limited subset of antigen-presenting cells: the

thymic medulla, B cells, and some dendritic cells. Its expression in the thymus may mean it has an important role in self-reactivity (*Welsh and Sadegh-Nasseri, 2020*).

MHC-DO is thought to bind to MHC-DM, inhibiting its process of removing CLIP from the classical Class II molecules. As a consequence, MHC-DO also limits the peptide editing that MHC-DM performs on MHC-DR, resulting in more CLIP and fewer less-stable peptides being presented on the cell surface (*Welsh and Sadegh-Nasseri, 2020*; *Olsson et al., 2022*). However, this process is not necessarily negative nor one-dimensional. Different ratios of MHC-DO to MHC-DM affect the lengths and types of peptides that are ultimately presented at the cell surface (*Olsson et al., 2022*). Therefore, MHC-DO can be thought of as a fine-tuner of the immunopeptidome rather than simply as an inhibitor of MHC-DM (*Welsh and Sadegh-Nasseri, 2020*).

