## [Editor Report · eLife Assessment]

This **important** manuscript presents a thorough analysis of the evolution of Major Histocompatibility Complex gene families across Primates. A key strength of this analysis is the use of state-of-the-art phylogenetic methods to estimate rates of gene gain and loss, accounting for the notorious difficulty to properly assemble MHC genomic regions. Overall the evidence for the authors' conclusions - that there is considerable diversity in how MHC diversity is deployed across species - **compelling**.

---

## [Referee Report · Joint Public Review]

Summary:

The Major Histocompatibility Complex (MHC) region is a collection of numerous genes involved in both innate and adaptive immunity. MHC genes are famed for their role in rapid evolution and extensive polymorphism in a variety of vertebrates. This paper presents a summary of gene-level gain and loss of orthologs and paralogs within MHC across the diversity of primates, using publicly available data.

Strengths:

This paper provides a strong case that MHC genes are rapidly gained (by paralog duplication) and lost over millions of years of macroevolution. The authors are able to identify MHC loci by homology across species, and from this infer gene duplications and losses using phylogenetic analyses. There is a remarkable amount of genic turnover, summarized in Figure 6 and Figure 7, either of which might be a future textbook figure of immune gene family evolution. The authors draw on state-of-the-art phylogenetic methods, and their inferences are robust.

Editorial note:

The authors have responded to the previous reviews and the Assessment was updated without involving the reviewers again.

---

## [Author Response]

The following is the authors’ response to the original reviews.

**Reviewer #1 (Public review):**
Summary:The Major Histocompatibility Complex (MHC) region is a collection of numerous genes involved in both innate and adaptive immunity. MHC genes are famed for their role in rapid evolution and extensive polymorphism in a variety of vertebrates. This paper presents a summary of gene-level gain and loss of orthologs and paralogs within MHC across the diversity of primates, using publicly available data.Strengths:This paper provides a strong case that MHC genes are rapidly gained (by paralog duplication) and lost over millions of years of macroevolution. The authors are able to identify MHC loci by homology across species, and from this infer gene duplications and losses using phylogenetic analyses. There is a remarkable amount of genic turnover, summarized in Figure 6 and Figure 7, either of which might be a future textbook figure of immune gene family evolution. The authors draw on state-of-the-art phylogenetic methods, and their inferences are robust insofar as the data might be complete enough to draw such conclusions.Weaknesses:One concern about the present work is that it relies on public databases to draw inferences about gene loss, which is potentially risky if the publicly available sequence data are incomplete. To say, for example, that a particular MHC gene copy is absent in a taxon (e.g., Class I locus F absent in Guenons according to Figure 1), we need to trust that its absence from the available databases is an accurate reflection of its absence in the genome of the actual organisms. This may be a safe assumption, but it rests on the completeness of genome assembly (and gene annotations?) or people uploading relevant data. This reviewer would have been far more comfortable had the authors engaged in some active spot-checking, doing the lab work to try to confirm absences at least for some loci and some species. Without this, a reader is left to wonder whether gene loss is simply reflecting imperfect databases, which then undercuts confidence in estimates of rates of gene loss.

Indeed, just because a locus has not been confirmed in a species does not necessarily mean that it is absent. As we explain in the Figure 1 caption, only a few species have had their genomes extensively studied (gray background), and only for these species does the absence of a point in this figure mean that a locus is absent. The white background rows represent species that are not extensively studied, and we point out that the absence of a point does not mean that a locus is absent from the species, rather undiscovered. We have also added a parenthetical to the text to explain this (line 156): “Only species with rows highlighted in gray have had their MHC regions extensively studied (and thus only for these rows is the absence of a gene symbol meaningful).”

While we agree that spot-checking may be a helpful next step, one of the goals of this manuscript is to collect and synthesize the enormous volume of MHC evolution research in the primates, which will serve as a jumping-off point for other researchers to perform important wet lab work.

Some context is useful for comparing rates of gene turnover in MHC, to other loci. Changing gene copy numbers, duplications, and loss of duplicates, are common it seems across many loci and many organisms; is MHC exceptional in this regard, or merely behaving like any moderately large gene family? I would very much have liked to see comparable analyses done for other gene families (immune, like TLRs, or non-immune), and quantitative comparisons of evolutionary rates between MHC versus other genes. Does MHC gene composition evolve any faster than a random gene family? At present readers may be tempted to infer this, but evidence is not provided.

Our companion paper (Fortier and Pritchard, 2025) demonstrates that the MHC is a unique locus in many regards, such as its evidence for deep balancing selection and its excess of disease associations. Thus, we expect that it is evolving faster than any random gene family. It would be interesting to repeat this analysis for other gene families, but that is outside of the scope of this project. Additionally, allele databases for other gene families are not nearly as developed, but as more alleles become available for other polymorphic families, a comparable analysis could become possible.

We have added a paragraph to the discussion (lines 530-546) to clarify that we do not know for certain whether the MHC gene family is evolving rapidly compared to other gene families.

While on the topic of making comparisons, the authors make a few statements about relative rates. For instance, lines 447-8 compare gene topology of classical versus non-classical genes; and line 450 states that classical genes experience more turnover. But there are no quantitative values given to these rates to provide numerical comparisons, nor confidence intervals provided (these are needed, given that they are estimates), nor formal statistical comparisons to confirm our confidence that rates differ between types of genes.More broadly, the paper uses sophisticated phylogenetic methods, but without taking advantage of macroevolutionary comparative methods that allow model-based estimation of macroevolutionary rates. I found the lack of quantitative measurements of rates of gene gain/loss to be a weakness of the present version of the paper, and something that should be readily remedied. When claiming that MHC Class I genes "turn over rapidly" (line 476) - what does rapidly mean? How rapidly? How does that compare to rates of genetic turnover at other families? Quantitative statements should be supported by quantitative estimates (and their confidence intervals).

These statements refer to qualitative observations, so we cannot provide numerical values. We simply conclude that certain gene groups evolve faster or slower based on the species and genes present in each clade. It is difficult to provide estimates because of the incomplete sampling of genes that survived to the present day. In addition, the presence or absence of various orthologs in different species still needs to be confirmed, at which point it might be useful to be more quantitative. We have also added a paragraph to the discussion to address this concern and advocate for similar analyses of other gene families in the future when more data is available (lines 530-546).

The authors refer to 'shared function of the MHC across species' (e.g. line 22); while this is likely true, they are not here presenting any functional data to confirm this, nor can they rule out neofunctionalization or subfunctionalization of gene duplicates. There is evidence in other vertebrates (e.g., cod) of MHC evolving appreciably altered functions, so one may not safely assume the function of a locus is static over long macroevolutionary periods, although that would be a plausible assumption at first glance.

Indeed, we cannot assume that the function of a locus is static across time, especially for the MHC region. In our research, we read hundreds of papers that each focused on a small number of species or genes and gathered some information about them, sometimes based on functional experiments and sometimes on measures such as dN/dS. These provide some indication of a gene’s broad classification in a species or clade, even if the evidence is preliminary. Where possible, we used this preliminary evidence to give genes descriptors “classical,” “non-classical,” “dual characteristics,” “pseudogene,” “fixed”, or “unfixed.” Sometimes multiple individuals and haplotypes were analyzed, so we could even assign a minimum number of gene copies present in a species. We have aggregated all of these references into Supplementary Table 1 (for Class I/Figure 1) and Supplementary Table 2 (for Class II/Figure 2) along with specific details about which data points in these figures that each reference supports. We realize that many of these classifications are based on a small number of individuals or indirect measures, so they may change in the future as more functional data is generated.

**Reviewer #2 (Public review):**
Summary:The authors aim to provide a comprehensive understanding of the evolutionary history of the Major Histocompatibility Complex (MHC) gene family across primate species. Specifically, they sought to:(1) Analyze the evolutionary patterns of MHC genes and pseudogenes across the entire primate order, spanning 60 million years of evolution.(2) Build gene and allele trees to compare the evolutionary rates of MHC Class I and Class II genes, with a focus on identifying which genes have evolved rapidly and which have remained stable.(3) Investigate the role of often-overlooked pseudogenes in reconstructing evolutionary events, especially within the Class I region.(4) Highlight how different primate species use varied MHC genes, haplotypes, and genetic variation to mount successful immune responses, despite the shared function of the MHC across species.(5) Fill gaps in the current understanding of MHC evolution by taking a broader, multi-species perspective using (a) phylogenomic analytical computing methods such as Beast2, Geneconv, BLAST, and the much larger computing capacities that have been developed and made available to researchers over the past few decades, (b) literature review for gene content and arrangement, and genomic rearrangements via haplotype comparisons.(6) The authors overall conclusions based on their analyses and results are that 'different species employ different genes, haplotypes, and patterns of variation to achieve a successful immune response'.Strengths:Essentially, much of the information presented in this paper is already well-known in the MHC field of genomic and genetic research, with few new conclusions and with insufficient respect to past studies. Nevertheless, while MHC evolution is a well-studied area, this paper potentially adds some originality through its comprehensive, cross-species evolutionary analysis of primates, focus on pseudogenes and the modern, large-scale methods employed. Its originality lies in its broad evolutionary scope of the primate order among mammals with solid methodological and phylogenetic analyses.The main strengths of this study are the use of large publicly available databases for primate MHC sequences, the intensive computing involved, the phylogenetic tool Beast2 to create multigene Bayesian phylogenetic trees using sequences from all genes and species, separated into Class I and Class II groups to provide a backbone of broad relationships to investigate subtrees, and the presentation of various subtrees as species and gene trees in an attempt to elucidate the unique gene duplications within the different species. The study provides some additional insights with summaries of MHC reference genomes and haplotypes in the context of a literature review to identify the gene content and haplotypes known to be present in different primate species. The phylogenetic overlays or ideograms (Figures 6 and 7) in part show the complexity of the evolution and organisation of the primate MHC genes via the orthologous and paralogous gene and species pathways progressively from the poorly-studied NWM, across a few moderately studied ape species, to the better-studied human MHC genes and haplotypes.Weaknesses:The title 'The Primate Major Histocompatibility Complex: An Illustrative Example of GeneFamily Evolution' suggests that the paper will explore how the Major Histocompatibility Complex (MHC) in primates serves as a model for understanding gene family evolution. The term 'Illustrative Example' in the title would be appropriate if the paper aimed to use the primate Major Histocompatibility Complex (MHC) as a clear and representative case to demonstrate broader principles of gene family evolution. That is, the MHC gene family is not just one instance of gene family evolution but serves as a well-studied, insightful example that can highlight key mechanisms and concepts applicable to other gene families. However, this is not the case, this paper only covers specific details of primate MHC evolution without drawing broader lessons to any other gene families. So, the term 'Illustrative Example' is too broad or generalizing. In this case, a term like 'Case Study' or simply 'Example' would be more suitable. Perhaps, 'An Example of Gene Family Diversity' would be more precise. Also, an explanation or 'reminder' is suggested that this study is not about the origins of the MHC genes from the earliest jawed vertebrates per se (~600 mya), but it is an extension within a subspecies set that has emerged relatively late (~60 mya) in the evolutionary divergent pathways of the MHC genes, systems, and various vertebrate species.

Thank you for your input on the title; we have changed it to “A case study of gene family evolution” instead.

Thank you also for pointing out the potential confusion about the time span of our study. We have added “Having originated in the jawed vertebrates,” to a sentence in the introduction (lines 38-39). We have also added the sentence “Here, we focus on the primates, spanning approximately 60 million years within the over 500-million-year evolution of the family (Flajnik 2010).“ to be more explicit about the context for our work (lines 59-61).

Phylogenomics. Particular weaknesses in this study are the limitations and problems associated with providing phylogenetic gene and species trees to try and solve the complex issue of the molecular mechanisms involved with imperfect gene duplications, losses, and rearrangements in a complex genomic region such as the MHC that is involved in various effects on the response and regulation of the immune system. A particular deficiency is drawing conclusions based on a single exon of the genes. Different exons present different trees. Which are the more reliable? Why were introns not included in the analyses? The authors attempt to overcome these limitations by including genomic haplotype analysis, duplication models, and the supporting or contradictory information available in previous publications. They succeed in part with this multidiscipline approach, but much is missed because of biased literature selection. The authors should include a paragraph about the benefits and limitations of the software that they have chosen for their analysis, and perhaps suggest some alternative tools that they might have tried comparatively. How were problems with Bayesian phylogeny such as computational intensity, choosing probabilities, choosing particular exons for analysis, assumptions of evolutionary models, rates of evolution, systemic bias, and absence of structural and functional information addressed and controlled for in this study?

We agree that different exons have different trees, which is exactly why we repeated our analysis for each exon in order to compare and contrast them. In particular, the exons encoding the binding site of the resulting protein (exons 2 and 3 for Class I and exon 2 for Class II) show evidence for trans-species polymorphism and gene conversion. These phenomena lead to trees that do not follow the species tree and are fascinating in and of themselves, which we explore in detail in our companion paper (Fortier and Pritchard, 2025). Meanwhile, the non-peptide-binding extracellular-domain-encoding exon (exon 4 for Class I and exon 3 for Class II) is comparably sized to the binding-site-encoding exons and provides an interesting functional contrast. As this exon is likely less affected by trans-species polymorphism, gene conversion, and convergent evolution, we present results from it most often in the main text, though we occasionally touch on differences between the exons. See lines 191-196, 223-226, and 407-414 for some examples of how we discuss the exons in the text. Additionally, all trees from all of these exons can be found in the supplement.

We agree that introns would valuable to study in this context. Even though the non--binding-site-encoding exons are probably *less* affected by trans-species polymorphism, gene conversion, and convergent evolution, they are still functional. The introns, however, experience much more relaxed selection, if any, and comparing their trees to those for the exons would be valuable and illuminating. We did not generate intron trees for two reasons. Most importantly, there is a dearth of data available for the introns; in the databases we used, there was often intron data available only for human, chimpanzee, and sometimes macaque, and only for a small subset of the genes. This limitation is at odds with the comprehensive, many-gene-many-species approach which we feel is the main novelty of this work. Secondly, the introns that *are* available are difficult to align. Even aligning the exons across such a highly-diverged set of genes and pseudogenes was difficult and required manual effort. The introns proved even more difficult to try to align across genes. In the future, when more intron data is available and sufficient effort is put into aligning them, it will be possible and desirable to do a comparable analysis. We also added a sentence to the “Data” section to briefly explain why we did not include introns (lines 134-135).

We explain our Bayesian phylogenetics approach in detail in the Methods (lines 650-725), including our assumptions and our solutions to challenges specific to this application. For further explanation of the method itself, we suggest reading the original *BEAST* and *BEAST2* papers (Drummond & Rambaut (2007), Drummond et al. (2012), Bouckaert et al. (2014), and Bouckaert et al. (2019)). Known structural and functional information helped us validate the alignments we used in this study, but the fact that such information is not fully known for every gene and species should not affect the method itself.

Gene families as haplotypes. In the Introduction, the MHC is referred to as a 'gene family', and in paragraph 2, it is described as being united by the 'MHC fold', despite exhibiting 'very diverse functions'. However, the MHC region is more accurately described as a multigene region containing diverse, haplotype-specific Conserved Polymorphic Sequences, many of which are likely to be regulatory rather than protein-coding. These regulatory elements are essential for controlling the expression of multiple MHC-related products, such as TNF and complement proteins, a relationship demonstrated over 30 years ago. Non-MHC fold loci such as TNF, complement, POU5F1, lncRNA, TRIM genes, LTA, LTB, NFkBIL1, etc, are present across all MHC haplotypes and play significant roles in regulation. Evolutionary selection must act on genotypes, considering both paternal and maternal haplotypes, rather than on individual genes alone. While it is valuable to compile databases for public use, their utility is diminished if they perpetuate outdated theories like the 'birth-and-death model'. The inclusion of prior information or assumptions used in a statistical or computational model, typically in Bayesian analysis, is commendable, but they should be based on genotypic data rather than older models. A more robust approach would consider the imperfect duplication of segments, the history of their conservation, and the functional differences in inheritance patterns. Additionally, the MHC should be examined as a genomic region, with ancestral haplotypes and sequence changes or rearrangements serving as key indicators of human evolution after the 'Out of Africa' migration, and with disease susceptibility providing a measurable outcome. There are more than 7000 different HLA-B and -C alleles at each locus, which suggests that there are many thousands of human HLA haplotypes to study. In this regard, the studies by Dawkins et al (1999 Immunol Rev 167,275), Shiina et al. (2006 Genetics 173,1555) on human MHC gene diversity and disease hitchhiking (haplotypes), and Sznarkowska et al. (2020 Cancers 12,1155) on the complex regulatory networks governing MHC expression, both in terms of immune transcription factor binding sites and regulatory non-coding RNAs, should be examined in greater detail, particularly in the context of MHC gene allelic diversity and locus organization in humans and other primates.

Thank you for these comments. To clarify that the MHC “region” is different from (and contains) the MHC “gene family” as we describe it, we changed a sentence in the abstract (lines 8-10) from “One large gene family that has experienced rapid evolution is the Major Histocompatibility Complex (MHC), whose proteins serve critical roles in innate and adaptive immunity.” to “One large gene family that has experienced rapid evolution lies within the Major Histocompatibility Complex (MHC), whose proteins serve critical roles in innate and adaptive immunity.” We know that the region is complex and contains many other genes and regulatory sequences; Figure 1 of our companion paper (Fortier and Pritchard, 2025) depicts these in order to show the reader that the MHC genes we focus on are just one part of the entire region.

We love the suggestion to look at the many thousands of alleles present at each of the classical loci. This is the focus of our complimentary paper (Fortier and Pritchard, 2025) which explores variation at the allele level. In the current paper, we look mainly at the differences between genes and the use of different genes in different species.

Diversifying and/or concerted evolution. Both this and past studies highlight diversifying selection or balancing selection model is the dominant force in MHC evolution. This is primarily because the extreme polymorphism observed in MHC genes is advantageous for populations in terms of pathogen defence. Diversification increases the range of peptides that can be presented to T cells, enhancing the immune response. The peptide-binding regions of MHC genes are highly variable, and this variability is maintained through selection for immune function, especially in the face of rapidly evolving pathogens. In contrast, concerted evolution, which typically involves the homogenization of gene duplicates through processes like gene conversion or unequal crossing-over, seems to play a minimal role in MHC evolution. Although gene duplication events have occurred in the MHC region leading to the expansion of gene families, the resulting paralogs often undergo divergent evolution rather than being kept similar or homozygous by concerted evolution. Therefore, unlike gene families such as ribosomal RNA genes or histone genes, where concerted evolution leads to highly similar copies, MHC genes display much higher levels of allelic and functional diversification. Each MHC gene copy tends to evolve independently after duplication, acquiring unique polymorphisms that enhance the repertoire of antigen presentation, rather than undergoing homogenization through gene conversion. Also, in some populations with high polymorphism or genetic drift, allele frequencies may become similar over time without the influence of gene conversion. This similarity can be mistaken for gene conversion when it is simply due to neutral evolution or drift, particularly in small populations or bottlenecked species. Moreover, gene conversion might contribute to greater diversity by creating hybrids or mosaics between different MHC genes. In this regard, can the authors indicate what percentage of the gene numbers in their study have been homogenised by gene conversion compared to those that have been diversified by gene conversion?

We appreciate the summary, and we feel we have appropriately discussed both gene conversion and diversifying selection in the context of the MHC genes. Because we cannot know for sure when and where gene conversion has occurred, we cannot quantify percentages of genes that have been homogenized or diversified.

Duplication models. The phylogenetic overlays or ideograms (Figures 6 and 7) show considerable imperfect multigene duplications, losses, and rearrangements, but the paper's Discussion provides no in-depth consideration of the various multigenic models or mechanisms that can be used to explain the occurrence of such events. How do their duplication models compare to those proposed by others? For example, their text simply says on line 292, 'the proposed series of events is not always consistent with phylogenetic data'. How, why, when? Duplication models for the generation and extension of the human MHC class I genes as duplicons (extended gene or segmental genomic structures) by parsimonious imperfect tandem duplications with deletions and rearrangements in the alpha, beta, and kappa blocks were already formulated in the late 1990s and extended to the rhesus macaque in 2004 based on genomic haplotypic sequences. These studies were based on genomic sequences (genes, pseudogenes, retroelements), dot plot matrix comparisons, and phylogenetic analyses of gene and retroelement sequences using computer programs. It already was noted or proposed in these earlier 1999 studies that (1) the ancestor of HLA-P(90)/-T(16)/W(80) represented an old lineage separate from the other HLA class I genes in the alpha block, (2) HLA-U(21) is a duplicated fragment of HLA-A, (3) HLA-F and HLA-V(75) are among the earliest (progenitor) genes or outgroups within the alpha block, (4) distinct Alu and L1 retroelement sequences adjoining HLA-L(30), and HLA-N genomic segments (duplicons) in the kappa block are closely related to those in the HLA-B and HLA-C in the beta block; suggesting an inverted duplication and transposition of the HLA genes and retroelements between the beta and kappa regions. None of these prior human studies were referenced by Fortier and Pritchard in their paper. How does their human MHC class I gene duplication model (Fig. 6) such as gene duplication numbers and turnovers differ from those previously proposed and described by Kulski et al (1997 JME 45,599), (1999 JME 49,84), (2000 JME 50,510), Dawkins et al (1999 Immunol Rev 167,275), and Gaudieri et al (1999 GR 9,541)? Is this a case of reinventing the wheel?

Figures 6 and 7 are intended to synthesize and reconcile past findings and our own trees, so they do not strictly adhere to the findings of any particular study and cannot fully match all studies. In the supplement, Figure 6 - figure supplement 1 and Figure 7 - figure supplement 1 duly credit all of the past work that went into making these trees. Most previous papers focus on just one aspect of these trees, such as haplotypes within a species, a specific gene or allelic lineage relationship, or the branching pattern of particular gene groups. We believe it was necessary to bring all of these pieces of evidence together. Even among papers with the same focus (to understand the block duplications that generated the current physical layout of the MHC), results differ. For example, Geraghty (1992), Hughes (1995), Kulski (2004)/Kulski (2005), and Shiina (1999) all disagree on the exact branching order of the genes MHC-W, -P, and -T, and of MHC-G, -J, and -K. While the Kulski studies you pointed out were very thorough for their era, they still only relied on data from three species and one haplotype per species. Our work is not intended to replace or discredit these past works, simply build upon them with a larger set of species and sequences. We hope the hypotheses we propose in Figures 6 and 7 can help unify existing research and provide a more easily accessible jumping-off-point for future work.

Results. The results are presented as new findings, whereas most if not all of the results' significance and importance already have been discussed in various other publications. Therefore, the authors might do better to combine the results and discussion into a single section with appropriate citations to previously published findings presented among their results for comparison. Do the trees and subsets differ from previous publications, albeit that they might have fewer comparative examples and samples than the present preprint? Alternatively, the results and discussion could be combined and presented as a review of the field, which would make more sense and be more honest than the current format of essentially rehashing old data.

In starting this project, we found that a large barrier to entry to this field of study is the immense amount of published literature over 30+ years. It is both time-consuming and confusing to read up on the many nuances of the MHC genes, their changing names, and their evolution, making it difficult to start new, innovative projects. We acknowledge that while our results are not entirely novel, the main advantage of our work is that it provides a thorough, comprehensive starting point for others to learn about the MHC quickly and dive into new research. We feel that we have appropriately cited past literature in both the main text, appendices, and supplement, so that readers may dive into a particular area with ease.

Minor corrections:(1) Abstract, line 19: 'modern methods'. Too general. What modern methods?

To keep the abstract brief, the methods are introduced in the main text when each becomes relevant as well as in the methods section.

(2) Abstract, line 25: 'look into [primate] MHC evolution.' The analysis is on the primate MHC genes, not on the entire vertebrate MHC evolution with a gene collection from sharks to humans. The non-primate MHC genes are often differently organised and structurally evolved in comparison to primate MHC.

Thank you! We have added the word “primate” to the abstract (line 25).

(3) Introduction, line 113. 'In a companion paper (Fortier and Pritchard, 2024)' This paper appears to be unpublished. If it's unpublished, it should not be referenced.

This paper is undergoing the eLife editorial process at the same time; it will have a proper citation in the final version.

(4) Figures 1 and 2. Use the term 'gene symbols' (circle, square, triangle, inverted triangle, diamond) or 'gene markers' instead of 'points'. 'Asterisks "within symbols" indicate new information.

Thank you, the word “symbol” is much clearer! We have changed “points” to “symbols” in the captions for Figure 1, Figure 1 - figure supplement 1, Figure 2, and Figure 2 - figure supplement 1. We also changed this in the text (lines 157-158 and 170).

(5) Figures. A variety of colours have been applied for visualisation. However, some coloured texts are so light in colour that they are difficult to read against a white background. Could darker colours or black be used for all or most texts?

With such a large number of genes and species to handle in this work, it was nearly impossible to choose a set of colors that were distinct enough from each other. We decided to prioritize consistency (across this paper, its supplement, and our companion paper) as well as at-a-glance grouping of similar sequences. Unfortunately, this means we had to sacrifice readability on a white background, but readers may turn to the supplement if they need to access specific sequence names.

(6) Results, line 135. '(Fortier and Pritchard, 2024)' This paper appears to be unpublished. If it's unpublished, it should not be referenced.

Repeat of (3). This paper is undergoing the eLife editorial process at the same time; it will have a proper citation in the final version.

(7) Results, lines 152 to 153, 164, 165, etc. 'Points with an asterisk'. Use the term 'gene symbols' (circle, square, triangle, inverted triangle, diamond) or 'gene markers' instead of 'points'. A point is a small dot such as those used in data points for plotting graphs .... The figures are so small that the asterisks in the circles, squares, triangles, etc, look like points (dots) and the points/asterisks terminology that is used is very confusing visually.

Repeat of (4). Thank you, the word “symbol” is much clearer! We have changed “points” to “symbols” in the captions for Figure 1, Figure 1 - figure supplement 1, Figure 2, and Figure 2 - figure supplement 1. We also changed this in the text (lines 157-158 and 170).

(8) Line 178 (BEA, 2024) is not listed alphabetically in the References.

Thank you for catching this! This reference maps to the first bibliography entry, “SUMMARIZING POSTERIOR TREES.” We are unsure how to cite a webpage that has no explicit author within the eLife Overleaf template, so we will consult with the editor.

(9) Lines 188-190. 'NWM MHC-G does not group with ape/OWM MHC-G, instead falling outside of the clade containing ape/OWM MHC-A, -G, -J and -K.' This is not surprising given that MHC-A, -G, -J, and -K are paralogs of each other and that some of them, especially in NWM have diverged over time from the paralogs and/or orthologs and might be closer to one paralog than another and not be an actual ortholog of OWM, apes or humans.

We included this sentence to clarify the relationships between genes and to help describe what is happening in Figure 6. Figure 6 - figure supplement 1 includes all of the references that go into such a statement and Appendix 3 details our reasoning for this and other statements.

(10) Line 249. Gene conversion: This is recombination between two different genes where a portion of the genes are exchanged with one another so that different portions of the gene can group within one or other of the two gene clades. Alternatively, the gene has been annotated incorrectly if the gene does not group within either of the two alternative clades. Another possibility is that one or two nucleotide mutations have occurred without a recombination resulting in a mistaken interpretation or conclusion of a recombination event. What measures are taken to avoid false-positive conclusions? How many MHC gene conversion (recombination) events have occurred according to the authors' estimates? What measures are taken to avoid false-positive conclusions?

All of these possibilities are certainly valid. We used the program GENECONV to infer gene conversion events, but there is considerable uncertainty owing to the ages of the genes and the inevitable point mutations that have occurred post-event. Gene conversion was not the focus of our paper, so we did our best to acknowledge it (and the resulting differences between trees from different exons) without spending too much time diving into it. A list of inferred gene conversion events can be found in Figure 3 - source data 1 and Figure 4 - source data 1.

(11) Lines 284-286. 'The Class I MHC region is further divided into three polymorphic blocks-alpha, beta, and kappa blocks-that each contains MHC genes but are separated by well-conserved non-MHC genes.' The MHC class I region was first designated into conserved polymorphic duplication blocks, alpha and beta by Dawkins et al (1999 Immunol Rev 167,275), and kappa by Kulski et al (2002 Immunol Rev 190,95), and should be acknowledged (cited) accordingly.

Thank you for catching this! We have added these citations (lines 302-303)!

(12) Lines 285-286. 'The majority of the Class I genes are located in the alpha-block, which in humans includes 12 MHC genes and pseudogenes.' This is not strictly correct for many other species, because the majority of class I genes might be in the beta block of new and old-world monkeys, and the authors haven't provided respective counts of duplication numbers to show otherwise. The alpha block in some non-primate mammalian species such as pigs, rats, and mice has no MHC class I genes or only a few. Most MHC class I genes in non-primate mammalian species are found in other regions. For example, see Ando et al (2005 Immunogenetics 57,864) for the pig alpha, beta, and kappa regions in the MHC class I region. There are no pig MHC genes in the alpha block.

Yes, which is exactly why we use the phrase “in humans” in that particular sentence. The arrangement of the MHC in several other primate reference genomes is shown in Figure 1 - figure supplement 2.

(13) Line 297 to 299. 'The alpha-block also contains a large number of repetitive elements and gene fragments belonging to other gene families, and their specific repeating pattern in humans led to the conclusion that the region was formed by successive block duplications (Shiina et al., 1999).' There are different models for successive block duplications in the alpha block and some are more parsimonious based on imperfect multigenic segmental duplications (Kulski et al 1999, 2000) than others (Shiina et al., 1999). In this regard, Kulski et al (1999, 2000) also used duplicated repetitive elements neighbouring MHC genes to support their phylogenetic analyses and multigenic segmental duplication models. For comparison, can the authors indicate how many duplications and deletions they have in their models for each species?

We have added citations to this sentence to show that there are different published models to describe the successive block duplications (line 307). Our models in Figure 6 and Figure 7 are meant to aggregate past work and integrate our own, and thus they were not built strictly by parsimony. References can be found in Figure 6 - figure supplement 1 and Figure 7 - figure supplement 1.

(14) Lines 315-315. 'Ours is the first work to show that MHC-U is actually an MHC-A-related gene fragment.' This sentence should be deleted. Other researchers had already inferred that MHC-U is actually an MHC-A-related gene fragment more than 25 years ago (Kulski et al 1999, 2000) when the MHC-U was originally named MHC-21.

While these works certainly describe MHC-U/MHC-21 as a fragment in the 𝛼-block, any relation to MHC-A was by association only and very few species/haplotypes were examined. So although the idea is not wholly novel, we provide convincing evidence that not only is MHC-U related to MHC-A by sequence, but also that it is a very recent partial duplicate of MHC-A. We show this with Bayesian phylogenetic trees as well as an analysis of haplotypes across many more species than were included in those papers.

(15) Lines 361-362. 'Notably, our work has revealed that MHC-V is an old fragment.' This is not a new finding or hypothesis. Previous phylogenetic analysis and gene duplication modelling had already inferred HLA-V (formerly HLA-75) to be an old fragment (Kulski et al 1999, 2000).

By “old,” we mean older than previous hypotheses suggest. Previous work has proposed that MHC-V and -P were duplicated together, with MHC-V deriving from an MHC-A/H/V ancestral gene and MHC-P deriving from an MHC-W/T/P ancestral gene (Kulski (2005), Shiina (1999)). However, our analysis (Figure 5A) shows that MHC-V sequences form a monophyletic clade outside of the MHC-W/P/T group of genes as well as outside of the MHC-A/B/C/E/F/G/J/K/L group of genes, which is not consistent with MHC-A and -V being closely related. Thus, we conclude that MHC-V split off earlier than the differentiation of these other gene groups and is thus older than previously thought. We explain this in the text as well (lines 317-327) and in Appendix 3.

(16) Line 431-433. 'the Class II genes have been largely stable across the mammals, although we do see some lineage-specific expansions and contractions (Figure 2 and Figure 2-gure Supplement 2).' Please provide one or two references to support this statement. Is 'gure' a typo?

We corrected this typo, thank you! This conclusion is simply drawn from the data presented in Figure 2 and Figure 2 - figure supplement 2. The data itself comes from a variety of sources, which are already included in the supplement as Figure 2 - source data 1.

(17) Line 437. 'We discovered far more "specific" events in Class I, while "broad-scale" events were predominant in Class II.' Please define the difference between 'specific' and 'broad-scale'.

These terms are defined in the previous sentence (lines 466-469).

450-451. 'This shows that classical genes experience more turnover and are more often affected by long-term balancing selection or convergent evolution.' Is balancing selection a form of divergent evolution that is different from convergent evolution? Please explain in more detail how and why balancing selection or convergent evolution affects classical and nonclassical genes differently.

Balancing selection acts to keep alleles at moderate frequencies, preventing any from fixing in the population. In contrast, convergent evolution describes sequences or traits becoming similar over time even though they are not similar by descent. While we cannot know exactly what selective forces have occurred in the past, we observe different patterns in the trees for each type of gene. In Figures 1 and 2, viewers can see at first glance that the nonclassical genes (which are named throughout the text and thoroughly described in Appendix 3) appear to be longer-lived than the classical genes. In addition, lines 204-222 and 475-488 describe topological differences in the *BEAST2* trees of these two types of genes. However, we acknowledge that it could be helpful to have additional, complimentary information about the classical vs. non-classical genes. Thus, we have added a sentence and reference to our companion paper (Fortier and Pritchard, 2025), which focuses on long-term balancing selection and draws further contrast between classical and non-classical genes. In lines 481-484, we added “We further explore the differences between classical and non-classical genes in our companion paper, finding ancient trans-species polymorphism at the classical genes but not at the non-classical genes (Fortier2025b).”

References

Some references in the supplementary materials such as Alvarez (1997), Daza-Vamenta (2004), Rojo (2005), Aarnink (2014), Kulski (2022), and others are missing from the Reference list. Please check that all the references in the text and the supplementary materials are listed correctly and alphabetically.

We will make sure that these all show up properly in the proof.

**Reviewer #3 (Public review):**
Summary:The article provides the most comprehensive overview of primate MHC class I and class II genes to date, combining published data with an exploration of the available genome assemblies in a coherent phylogenetic framework and formulating new hypotheses about the evolution of the primate MHC genomic region.Strengths:I think this is a solid piece of work that will be the reference for years to come, at least until population-scale haplotype-resolved whole-genome resequencing of any mammalian species becomes standard. The work is timely because there is an obvious need to move beyond short amplicon-based polymorphism surveys and classical comparative genomic studies. The paper is data-rich and the approach taken by the authors, i.e. an integrative phylogeny of all MHC genes within a given class across species and the inclusion of often ignored pseudogenes, makes a lot of sense. The focus on primates is a good idea because of the wealth of genomic and, in some cases, functional data, and the relatively densely populated phylogenetic tree facilitates the reconstruction of rapid evolutionary events, providing insights into the mechanisms of MHC evolution. Appendices 1-2 may seem unusual at first glance, but I found them helpful in distilling the information that the authors consider essential, thus reducing the need for the reader to wade through a vast amount of literature. Appendix 3 is an extremely valuable companion in navigating the maze of primate MHC genes and associated terminology.Weaknesses:I have not identified major weaknesses and my comments are mostly requests for clarification and justification of some methodological choices.

Thank you so much for your kind and supportive review!

**Reviewer #1 (Recommendations for the authors):**
(1) Line 151: How is 'extensively studied' defined?

Extensively studied is not a strict definition, but a few organisms clearly stand apart from the rest in terms of how thoroughly their MHC regions have been studied. For example, the macaque is a model organism, and individuals from many different species and populations have had their MHC regions fully sequenced. This is in contrast to the gibbon, for example, in which there is some experimental evidence for the presence of certain genes, but no MHC region has been fully sequenced from these animals.

(2) Can you clarify how 'classical' and 'non-classical' MHC genes are being determined in your analysis?

Classical genes are those whose protein products perform antigen presentation to T cells and are directly involved in adaptive immunity, while non-classical genes are those whose protein products do not do this. For example, these non-classical genes might code for proteins that interact with receptors on Natural Killer cells and influence innate immunity. The roles of these proteins are not necessarily conserved between closely related species, and experimental evidence is needed to evaluate this. However, in the absence of such evidence, wherever possible we have provided our best guess as to the roles of the orthologous genes in other species, presented in Figure 1 - source data 1 and Figure 2 - source data 1. This is based on whatever evidence is available at the moment, sometimes experimental but typically based on dN/dS ratios and other indirect measures.

(3) I find the overall tone of the paper to be very descriptive, and at times meandering and repetitive, with a lot of similar kinds of statements being repeated about gene gain/loss. This is perhaps inevitable because a single question is being asked of each of many subsets of MHC gene types, and even exons within gene types, so there is a lot of repetition in content with a slightly different focus each time. This does not help the reader stay focused or keep track. I found myself wishing for a clearly defined question or hypothesis, or some rate parameter in need of estimation. I would encourage the authors to tighten up their phrasing, or consider streamlining the results with some better signposting to organize ideas within the results.

We totally understand your critique, as we talk about a wide range of specific genes and gene groups in this paper. To improve readability, we have added many more signposting phrases and sentences:

“Aside from MHC-DRB, …” (line 173)

“Now that we had a better picture of the landscape of MHC genes present in different primates, we wanted to understand the genes’ relationships. Treating Class I, Class IIA, and Class IIB separately, ...” (line 179-180)

“We focus first on the Class I genes.” (line 191)

“... for visualization purposes…” (line195)

“We find that sequences do not always assort by locus, as would be expected for a typical gene.” (lines 196-197)

“... rather than being directly orthologous to the ape/OWM MHC-G genes.” (lines 201-202)

“Appendix 3 explains each of these genes in detail, including previous work and findings from this study.“ (lines 202-203)

“... (but not with NWM) …” (line 208)

“While genes such as MHC-F have trees which closely match the overall species tree, other genes show markedly different patterns, …” (lines 212-213)

“Thus, while some MHC-G duplications appear to have occurred prior to speciation events within the NWM, others are species-specific.” (lines 218-219)

“... indicating rapid evolution of many of the Class I genes” (lines 220-221)

“Now turning to the Class II genes, …“ (line 223)

“(see Appendix 2 for details on allele nomenclature) “ (line 238)

“(e.g. MHC-DRB1 or -DRB2)” (line 254)

“... meaning their names reflect previously-observed functional similarity more than evolutionary relatedness.” (lines 257-258)

“(see Appendix 3 for more detail)” (line 311)

“(a 5'-end fragment)” (line 324)

“Therefore, we support past work that has deemed MHC-V an old fragment.” (lines 326-327)

“We next focus on MHC-U, a previously-uncharacterized fragment pseudogene containing only exon 3.” (line 328-329)

“However, it is present on both chimpanzee haplotypes and nearly all human haplotypes, and we know that these haplotypes diverged earlier---in the ancestor of human and gorilla. Therefore, ...” (lines 331-333)

“Ours is the first work to show that MHC-U is actually an MHC-A-related gene fragment and that it likely originated in the human-gorilla ancestor.” (lines 334-336)

“These pieces of evidence suggest that MHC-K and -KL duplicated in the ancestor of the apes.” (lines 341-342)

“Another large group of related pseudogenes in the Class I $\alpha$-block includes MHC-W, -P, and -T (see Appendix 3 for more detail).” (lines 349-350)

“...to form the current physical arrangement” (lines 354)

“Thus, we next focus on the behavior of this subgroup in the trees.” (line 358)

“(see Appendix 3 for further explanation).” (line 369)

“Thus, for the first time we show that there must have been three distinct MHC-W-like genes in the ape/OWM ancestor.” (lines 369-371)

“... and thus not included in the previous analysis. ” (lines 376-377)

“MHC-Y has also been identified in gorillas (Gogo-Y) (Hans et al., 2017), so we anticipate that Gogo-OLI will soon be confirmed. This evidence suggests that the MHC-Y and -OLI-containing haplotype is at least as old as the human-gorilla split. Our study is the first to place MHC-OLI in the overall story of MHC haplotype evolution“ (lines 381-384)

“Appendix 3 explains the pieces of evidence leading to all of these conclusions (and more!) in more detail.” (lines 395-396)

“However, looking at this exon alone does not give us a complete picture.” (lines 410-411)

“...instead of with other ape/OWM sequences, …” (lines 413-414)

“Figure 7 shows plausible steps that might have generated the current haplotypes and patterns of variation that we see in present-day primates. However, some species are poorly represented in the data, so the relationships between their genes and haplotypes are somewhat unclear.” (lines 427-429)

“(and more-diverged)” (line 473)

“(of both classes)” (line 476)

“..., although the classes differ in their rate of evolution.” (line 487-488)

“Including these pseudogenes in our trees helped us construct a new model of $\alpha$-block haplotype evolution. “ (lines 517-518)

(4) Line 480-82: "Notably...." why is this notable? Don't merely state that something is notable, explain what makes it especially worth drawing the reader's attention to: in what way is it particularly significant or surprising?

We have changed the text from “Notably” to “In particular” (line 390) so that readers are expecting us to list some specific findings. Similarly, we changed “Notably” to “Specifically” (line 515).

(5) The end of the discussion is weak: "provide context" is too vague and not a strong statement of something that we learned that we didn't know before, or its importance. This is followed by "This work will provide a jumping-off point for further exploration..." such as? What questions does this paper raise that merit further work?

We have made this paragraph more specific and added some possible future research directions. It now reads “By treating the MHC genes as a gene family and including more data than ever before, this work enhances our understanding of the evolutionary history of this remarkable region. Our extensive set of trees incorporating classical genes, non-classical genes, pseudogenes, gene fragments, and alleles of medical interest across a wide range of species will provide context for future evolutionary, genomic, disease, and immunologic studies. For example, this work provides a jumping-off-point for further exploration of the evolutionary processes affecting different subsets of the gene family and the nuances of immune system function in different species. This study also provides a necessary framework for understanding the evolution of particular allelic lineages within specific MHC genes, which we explore further in our companion paper (Fortier 2025b). Both studies shed light on MHC gene family evolutionary dynamics and bring us closer to understanding the evolutionary tradeoffs involved in MHC disease associations.” (lines 576-586)

**Reviewer #3 (Recommendations for the authors):**
(1) Figure 1 et seq. Classifying genes as having 'classical', 'non-classical' and 'dual' properties is notoriously difficult in non-model organisms due to the lack of relevant information. As you have characterised a number of genes for the first time in this paper and could not rely entirely on published classifications, please indicate the criteria you used for classification.

The roles of these proteins are not necessarily conserved between closely related species, and experimental evidence is needed to evaluate this. However, in the absence of such evidence, wherever possible we have provided our best guess as to the roles of the orthologous genes in other species, presented in Figure 1 - source data 1 and Figure 2 - source data 1. This is based on whatever evidence is available at the moment, sometimes experimental but typically based on dN/dS ratios and other indirect measures.

(2) Line 61 It's important to mention that classical MHC molecules present antigenic peptides to T cells with variable alphabeta T cell receptors, as non-classical MHC molecules may interact with other T cell subsets/types.

Thank you for pointing this out; we have updated the text to make this clearer (lines 63-65). We changed “‘Classical’ MHC molecules perform antigen presentation to T cells---a key part of adaptive immunity---while ‘non-classical’ molecules have niche immune roles.” to “‘Classical’ MHC molecules perform antigen presentation to T cells with variable alphabeta TCRs---a key part of adaptive immunity---while ‘non-classical’ molecules have niche immune roles.”

(3) Perhaps it's worth mentioning in the introduction that you are deliberately excluding highly divergent non-classical MHC molecules such as CD1.

Thank you, it’s worth clarifying exactly what molecules we are discussing. We have added a sentence to the introduction (lines 38-43): “Having originated in the jawed vertebrates, this group of genes is now involved in diverse functions including lipid metabolism, iron uptake regulation, and immune system function (proteins such as zinc-𝛼2-glycoprotein (ZAG), human hemochromatosis protein (HFE), MHC class I chain–related proteins (MICA, MICB), and the CD1 family) (Hansen2007,Kupfermann 1999, Kaufman 2022, Adams 2013). However, here we focus on…”

(4) Line 94-105 This material presents results, it could be moved to the results section as it now somewhat disrupts the flow.

We feel it is important to include a “teaser” of the results in the introduction, which can be slightly more detailed than that in the abstract.

(5) Line 118-131 This opening section of the results sets the stage for the whole presentation and contains important information that I feel needs to be expanded to include an overview and justification of your methodological choices. As the M&M section is at the end of the MS (and contains limited justification), some information on two aspects is needed here for the benefit of the reader. First, as far as I understand, all phylogenetic inferences were based entirely on DNA sequences of individual (in some cases concatenated) exons. It would be useful for the reader to explain why you've chosen to rely on DNA rather than protein sequences, even though some of the genes you include in the phylogenetic analysis are highly divergent. Second, a reader might wonder how the "maximum clade credibility tree" from the Bayesian analysis compares to commonly seen trees with bootstrap support or posterior probability values assigned to particular clades. Personally, I think that the authors' approach to identifying and presenting representative trees is reasonable (although one might wonder why "Maximum clade credibility tree" and not "Maximum credibility tree" https://www.beast2.org/summarizing-posterior-trees/), since they are working with a large number of short, sometimes divergent and sometimes rather similar sequences - in such cases, a requirement for strict clade support could result in trees composed largely of polytomies. However, I feel it's necessary to be explicit about this and to acknowledge that the relationships represented by fully resolved bifurcating representative trees and interpreted in the study may not actually be highly supported in the sense that many readers might expect. In other words, the reader should be aware from the outset of what the phylogenies that are so central to the paper represent.

We chose to rely on DNA rather than protein sequences because convergent evolution is likely to happen in regions that code for extremely important functions such as adaptive and innate immunity. Convergent evolution acts upon proteins while trans-species polymorphism retains ancient nucleotide variation, so studying the DNA sequence can help tease apart convergent evolution from trans-species polymorphism.

As for the “maximum clade credibility tree”, this is a matter of confusing nomenclature. In the online reference guide (https://www.beast2.org/summarizing-posterior-trees/), the tree with the maximum product of the posterior clade probabilities is called the “maximum credibility tree” while the tree that has the maximum sum of posterior clade probabilities is called the “Maximum credibility tree”. The “Maximum credibility tree” (referring to the sum) appears to have only been named in this way in the first version of TreeAnnotator. However, the version of TreeAnnotator that I used lists the options “maximum clade credibility tree” and “maximum sum of clade probabilities”. So the context suggests that the “maximum clade credibility tree” option is actually maximizing the product. This “maximum clade credibility tree” is the setting I used for this project (in TreeAnnotator version 2.6.3).

We agree that readers may not fully grasp what the collapsed trees represent upon first read. We have added a sentence to the beginning of the results (line 188-190) to make this more explicit.

(6) Line 224, you're referring to the DPB1*09 lineage, not the DRB1*09 lineage.

Indeed! We have changed these typos.

(7) Line 409, why "Differences between MHC subfamilies" and not "Differences between MHC classes"?

We chose the word “subfamilies” because we discuss the difference between classical and non-classical genes in addition to differences between Class I and Class II genes.

(8) Line 529-544 This might work better as a table.

We agree! This information is now presented as Table 1.

(9) Line 547 MHC-DRB9 appears out of the blue here - please say why you are singling it out.

Great point! We added a paragraph (lines 614-623) to explain why this was necessary.

(10) Line 550-551 Even though you've screened the hits manually, it would be helpful to outline your criteria for this search.

Thank you! We’ve added a couple of sentences to explain how we did this (lines 607-610).

(11) Line 556-580 please provide nucleotide alignments as supplementary data so that the reader can get an idea of the actual divergence of the sequences that have been aligned together.

Thank you! We’ve added nucleotide alignments as supplementary files.

(12) Line 651-652 Why "Maximum clade credibility tree" and not "Maximum credibility tree"?

Repeat of (5). This is a matter of confusing nomenclature. In the online reference guide (https://www.beast2.org/summarizing-posterior-trees/), the tree with the maximum product of the posterior clade probabilities is called the “maximum credibility tree” while the tree that has the maximum sum of posterior clade probabilities is called the “Maximum credibility tree”. The “Maximum credibility tree” (referring to the sum) appears to have only been named in this way in the first version of TreeAnnotator. However, the version of TreeAnnotator that I used lists the options “maximum clade credibility tree” and “maximum sum of clade probabilities”. So the context suggests that the “maximum clade credibility tree” option is actually maximizing the product. This “maximum clade credibility tree” is the setting I used for this project (in TreeAnnotator version 2.6.3).

(13) In the appendices, links to references do not work as expected.

We will make sure these work properly when we receive the proofs.